# DNA methylation modulates nucleosome retention in sperm and H3K4 methylation deposition in early mouse embryos

Grigorios Fanourgakis [1], Laura Gaspa-Toneu [1,2], Pavel A. Komarov [1,2], Panagiotis Papasaikas [1], Evgeniy A. Ozonov [1], Sebastien A. Smallwood[1] & Antoine H. F. M. Peters [1,2] ✉

In the germ line and during early embryogenesis, DNA methylation (DNAme) undergoes global erasure and re-establishment to support germ cell and embryonic development. While DNAme acquisition during male germ cell development is essential for setting genomic DNA methylation imprints, other intergenerational roles for paternal DNAme in defining embryonic chromatin are unknown. Through conditional gene deletion of the de novo DNA methyltransferases *Dnmt3a* and/or *Dnmt3b*, we observe that DNMT3A primarily safeguards against DNA hypomethylation in undifferentiated spermatogonia, while DNMT3B catalyzes de novo DNAme during spermatogonial differentiation. Failing de novo DNAme in *Dnmt3a/Dnmt3b* double deficient spermatogonia is associated with increased nucleosome occupancy in mature sperm, preferentially at sites with higher CpG content, supporting the model that DNAme modulates nucleosome retention in sperm. To assess the impact of altered sperm chromatin in formatting embryonic chromatin, we measure H3K4me3 occupancy at paternal and maternal alleles in 2-cell embryos using a transposon-based tagging approach. Our data show that reduced DNAme in sperm renders paternal alleles permissive for H3K4me3 establishment in early embryos, independently of possible paternal inheritance of sperm born H3K4me3. Together, this study provides evidence that paternally inherited DNAme directs chromatin formation during early embryonic development.

Inheritance of genetic information from parents to offspring is fundamental to sexual reproduction. Recent studies have suggested alternative modes of inheritance that extend beyond the DNA sequence itself. Such epigenetic modes of inheritance encompass various means of transmission, such as DNA methylation (DNAme) - the presence of a methyl group at the carbon-5 position of cytosine in the CpG dinucleotide context-, post translational modifications (PTMs) on histone proteins, and noncoding RNAs, which have been implicated in the transmission of parental traits and the regulation of gene expression in subsequent generations[1,2].

Global patterns of DNAme undergo multiple waves of acquisition and resetting during the mammalian life cycle. After germ cell specification during embryogenesis, genome-wide DNAme levels are reduced in proliferating primordial germ cells (PGCs), reaching residual levels of ~3-5% upon their arrival in embryonic gonads. DNAme is subsequently restored in a sexually dimorphic manner, with male germ cells (referred to as prospermatogonia) carrying high levels (~80%) by the time of birth[3–6]. Most global DNAme is reestablished in male embryonic germ cells by three de novo cytosine methyltransferases, DNMT3A, DNMT3B and DNMT3C along with their cofactor

[1]Friedrich Miescher Institute for Biomedical Research, Fabrikstrasse 24, 4056 Basel, Switzerland. [2]Faculty of Sciences, University of Basel, 4056 Basel, Switzerland. ✉e-mail: antoine.peters@fmi.ch

DNMT3L[7–10]. Residual DNAme acquisition occurs after birth leading to highly methylated genomes in mature spermatozoa[11–13]. After birth, undifferentiated spermatogonia (SgU) divide mitotically, undergo differentiation forming differentiating spermatogonia (SgD), and subsequently undergo meiosis and spermiogenesis to form highly differentiated haploid spermatozoa. While spermatozoa contain high levels of DNAme throughout their genomes, most regions enriched in CpG dinucleotides (CpG islands, CGIs), generally serving gene regulatory functions, remain unmethylated[13,14].

During the late stages of spermatid development, chromatin undergoes a major reprogramming event with most nucleosomes being replaced by small basic proteins called protamines, causing extensive nuclear compaction[15]. A small fraction of nucleosomes, however, is retained in sperm, up to approximately 2% in mouse and 15% in human[16–18]. Several studies reported nucleosome distributions throughout the sperm genome[19,20] while we and others reported enrichment at sequences enriched in CpG dinucleotides, largely coinciding with promoters and exons of coding and non-coding genes[17,18,20,21]. We further observed an inverse correlation between the degree of nucleosomal retention and the level of DNAme at such sequences[17,18,21]. It is currently unknown whether DNAme serves a regulatory role in nucleosomal remodeling during spermiogenesis. Recently, previously reported nucleosome enrichments profiles of sperm chromatin were challenged to originate largely from cell free chromatin derived from somatic cellular contaminants[22]. Methodological difficulties in preparing open sperm chromatin, however, preclude a decisive conclusion[22,23].

Upon fertilization, a process of extensive epigenetic reprogramming takes place, enabling the establishment of totipotency and the subsequent differentiation of distinct cell lineages. Following the nucleosomal repackaging of the paternal genome with maternally provided histones, global DNAme is promptly and actively removed[24–26]. DNAme patterns at specific genomic regions called imprinting control regions (ICRs) are, however, maintained, driving allele specific gene regulation at later stages of development in a parent-of-origin specific manner. Such imprinted form of gene regulation is the most well-defined example of DNAme based intergenerational epigenetic inheritance[27]. While maternal transmission of histone PTMs such as H3K27me3 is required for repression of maternal loci in embryos' placentae, representing a non-canonical form of genomic imprinting[28], an instructive role of histones and associated PTMs[29] in paternal epigenetic inheritance remains to be identified. To date, reports analyzing histone perturbations in sperm and associating them with altered phenotypes or expression in embryos are primarily correlative rather than causative in nature[15,29–34]. For example, spermatozoa of rats that had been exposed to toxicants during *in utero* development were reported to have altered histone retention sites that were, in part, also present in sperm of subsequent unexposed generations[30]. Spermatogonial specific overexpression of the histone demethylase KDM1A (LSD1) resulted in severely impaired development of offspring. Transmission of phenotypic defects, even via non-transgenic descendant fathers, correlated with increased H3K4me3 but not H3K4me2 occupancy in sperm[29,33]. Restricting folate in diet of male mice, known to modulate DNAme homeostasis[35,36], altered H3K4me3 profiles in sperm, which were correlated with altered gene expression in embryonic progeny[32]. The molecular mechanism and penetrance of such possible modes of epigenetic inheritance remain, however, to be demonstrated.

Given the known antagonistic interplay between the H3K4 and DNA methylation pathways[37–40], perturbations in one chromatin pathway may impinge on other chromatin characteristics. Understanding the extent by which chromatin states in sperm influence epigenetic reprogramming in early embryos is critical to unraveling the mechanisms underlying modes of intergenerational epigenetic inheritance and its implications for gene expression regulation.

Here we investigate DNAme as a possible determinant of chromatin configuration in sperm and as driver of paternal epigenetic inheritance. By studying germ cells single and double conditionally deficient for *Dnmt3a* and/or *Dnmt3b* we identify developmental and genomic context specific contributions of each enzyme to safeguarding and de novo depositing DNAme during spermatogenesis. By exploiting chromatin perturbations specific to germ cells, we identify two roles for DNAme in sperm in (a) modulating nucleosome occupancy and H3K4me3 enrichment at CpG-rich regions in developing sperm and (b) preventing premature acquisition of H3K4me3 specifically on paternally inherited alleles in the early embryo.

## Results

### Germ line *Dnmt3a/Dnmt3b* deficiency induces DNA hypomethylation in sperm

Previous studies explored the role of de novo DNA methyltransferases in the reestablishment of DNAme during fetal life and the generation of spermatogonial stem cells[7–10,41]. Their role during adult spermatogenesis is, however, poorly understood[42,43]. By immunofluorescent staining of seminiferous tubules, we detected prominent signals of DNMT3A and DNMT3B proteins in differentiating spermatogonia marked by cKIT labeling (Fig. 1A), as shown previously[44]. To assess the role of these proteins during adult spermatogenesis, we generated conditional deletion models of *Dnmt3a* (*3aKO*), *Dnmt3b* (*3bKO*) and double *Dnmt3a/Dnmt3b* (*DKO*). Excision of floxed alleles of *Dnmt3a* and/or *Dnmt3b* was driven by the improved *iCre* recombinase transgene under the control of the *Stra8* promoter, which is active in postnatal undifferentiated and differentiating spermatogonia[45]. DNMT3A and DNMT3B proteins were undetectable in the nuclei of spermatogonia from *3aKO* and *3bKO* animals respectively (Fig. 1A), and deletion of one paralogue did not affect protein expression of the other (Supplementary Fig. 1A). By RNA sequencing analysis in spermatogonia, despite measuring similar total RNA levels of both *Dnmt3a* and *Dnmt3b* among *control* (*Ctrl*) and *DKO* samples (Supplementary Fig. 1B), we did not detect reads mapping to the floxed exons of both *Dnmt3a* and *Dnmt3b* genes in *DKO* samples (Supplementary Fig. 1C), suggesting efficient excision. Likewise, by whole genome sequencing, we did not detect any reads mapping to the floxed regions of both *Dnmt3a* and *Dnmt3b* genetic loci in *DKO* sperm (Supplementary Fig. 1D). Altogether, our models show efficient conditional deficiencies of DNMT3A and/or DNMT3B allowing us to investigate their impact on DNAme and the progression of adult spermatogenesis.

Histological analysis of testicular sections showed an overall normal tissue morphology and presence of male germ cells at all developmental stages upon conditional ablation of the DNMT3A and/or DNMT3B proteins (Supplementary Fig. 1E). However, we noticed increased occurrence of seminiferous tubules lacking germ cells in single and double mutants compared to control mice (Supplementary Fig. 1F), suggestive of a partial failure to establish a functional spermatogonial stem cell compartment. Accordingly, we observed ~15%–25% reduction in testicular weight upon removal of either DNMT3B or DNMT3A enzyme respectively which was further aggravated to ~30% upon depletion of both enzymes (Supplementary Fig. 1G). Nevertheless, *DKO* males sired similar numbers of live offspring (Supplementary Fig. 1H) without evident phenotypic abnormalities or health issues until adulthood as compared to *Ctrl* males. These data suggest that DNMT3A/DNMT3B enzymes are largely dispensable in postnatal germ cells for the completion of spermatogenesis and production of competent gametes.

To investigate the impact of *Dnmt3a/Dnmt3b* deficiency on DNAme status of gametes, we performed whole genome Enzymatic Methyl-seq (EM-seq)[46] using FACS sorted sperm from 2 *Ctrl* and 2 *DKO* animals. We assayed the DNAme status of 98.6% and 99.0% of genomic CpGs to a mean combined coverage of 6.0X and 6.5X for the *Ctrl* and *DKO* samples respectively, with replicates exhibiting high correlations

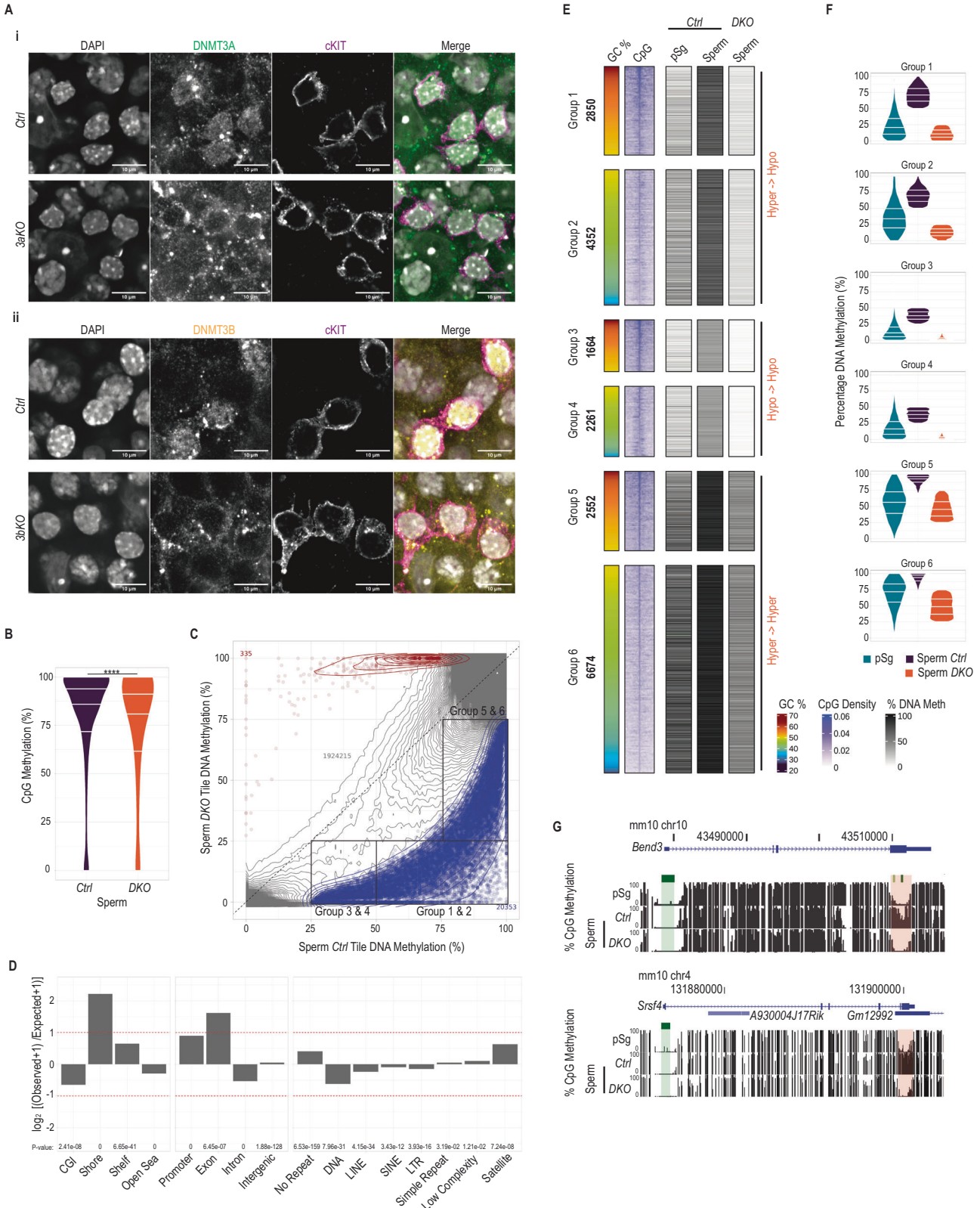

to each other (Supplementary Fig. 2A). Compared to *Ctrl* sperm, the mean CpG methylation level in *DKO* sperm was reduced from 81.7% to 76.5% (Fig. 1B). Reduced methylation was measured at CpGs residing in multiple genomic contexts (Supplementary Fig. 2B, C). DNA methylation at the paternally methylated imprinting control regions (ICRs) of *H19* and *Meg3* were unaffected, while the ICR for *Rasgrf1* exhibited a minor loss of DNA methylation (Supplementary Fig. 2D). These results

contrast with those of previous studies showing extremely low global DNAme levels, including paternally methylated ICRs, upon deleting de novo DNA methyltransferases in fetal germ cells[7,10] or DNAme maintenance components in postnatal germ cells[42].

We calculated DNAme levels in 1,944,903 500 bp non-overlapping genomic tiles which contained at least 5 CpGs and were covered by minimally 25 sequencing reads. When performing differential

**Fig. 1 | Identification of differentially hypomethylated regions in *DKO* sperm.**
**A** Whole mount immunofluorescence staining of (i) DNMT3A in seminiferous tubules of *Ctrl* (*n* = 1) and *3aKO* (*n* = 1) testes, and (ii) DNMT3B in seminiferous tubules of *Ctrl* (*n* = 1) and *3bKO* (*n* = 1) testes. All samples were co-stained for cKIT and DNA was visualized by DAPI. Maximum projections of multiple confocal z-stacks are shown. Scale bars = 10 μm. The remaining cytoplasmic spotty signal in the *Dnmt3a* and *Dnmt3b* single KO samples likely represents background staining.
**B** Violin plot showing the percentage of DNAme of genomic CpGs assessed by EMseq in *Ctrl* (*n* = 2) and *DKO* sperm (*n* = 2) samples. Center lines in violin plots represent median values while upper and lower lines indicate the interquartile range (IQR; from the 25th – 75th percentile). Two-sample, two-sided Wilcoxon test was performed between *Ctrl* and *DKO* groups with ****p*-value < 2.2e-16.
**C** Scatterplot showing percentage DNAme of 500 bp genomic tiles assessed by EMseq in *Ctrl* (*x*-axis) and *DKO* sperm (*y*-axis). Tiles with significantly increased methylation in *DKO* sperm are highlighted in red (HyperDMRs) and those with reduced levels in blue (HypoDMRs). Grey contour lines denote the density of

NonDMR tiles. **D** Bar plots showing $\log_2$ enrichments of features related to CpG Islands, genes and repetitive elements among hypoDMRs tiles compared to the whole genome. *P*-values were calculated by two-sided Fisher's exact test.
**E** Heatmap showing classification of hypoDMRs based on DNAme percentages assessed by EMseq in *Ctrl* (*n* = 2) and *DKO* (*n* = 2) sperm samples, and the GC percentage within hypoDMRs. pSg (*n* = 2) DNAme percentage assessed by PBATseq (Shirane et al., 2020) is shown as an external early germ cell developmental stage reference. The density of CpGs at hypoDMRs (±5 kb from their center) is indicated. Within each group, hypoDMRs are ordered according to decreasing GC percentages. **F** Violin plots showing percentages of DNAme assessed by EMseq of hypoDMRs in pSg, *Ctrl* and *DKO* sperm for each group. Center lines in violin plots represent median values while upper and lower lines indicate the interquartile range (IQR; from the 25th to 75th percentile). **G** UCSC genome browser snapshot of the *Bend3* and *Srsf4* loci, showing tracks for DNAme levels in pSg, *Ctrl* and *DKO* sperm. The promoter CGIs and the hypoDMR are highlighted in green and red respectively.

methylation analysis with stringent thresholds (methylation difference > 25% and FDR < 0.05), we identified 1% of all examined regions (*n* = 20,688/N = 1,944,903) as significantly differentially methylated regions (DMRs) in *DKO* versus *Ctrl* sperm, with the vast majority (20'353; 98.3%) being hypomethylated (hypoDMRs) (Fig. 1C). Tiles in *DKO* sperm showed a wide range in the reduction of DNAme (Fig. 1C). We found that CGI shores and exonic regions were overrepresented among hypoDMRs, while actual CGIs, intronic and repetitive sequences were represented proportionally to all genomic windows examined in this analysis (Fig. 1D). In summary, these data indicate that DNMT3A/DNMT3B activities regulate DNAme dynamics at specific genomic elements during adult spermatogenesis.

To investigate the variable hypomethylation observed in DKO sperm, we compared the sperm methylomes to those of wild-type prospermatogonia (pSg) at postnatal day 1 (PBAT-seq data from ref. 47), representing the starting stage of postnatal germ cell development. We classified the hypoDMRs into 6 groups, based on (a) their sequence composition (GC percentage > 50% for Groups 1,3,5 and <50% for Groups 2,4,6) and (b) their DNAme status in *Ctrl* and *DKO* samples (Fig. 1E).

Group 1 & 2 hypoDMRs displayed low-to-mid levels (~10–50%) of DNAme in pSg which increased to high levels (>50%) in *Ctrl* sperm. In *DKO* sperm, DNAme levels were reduced below levels present in pSg (<20%), as exemplified for the *Bend3* and *Srsf4* loci, suggesting that these are bona fide de novo methylated regions during adult spermatogenesis (Fig. 1C, E, F, and G).

HypoDMRs in groups 3 & 4 displayed only low DNAme (<20%) in pSg, mid-levels of DNAme (25%–50%) in *Ctrl* sperm and extremely low DNAme (<5%) levels in *DKO* sperm, suggesting that DNMT3A/DNMT3B enzymes are also required to sustain DNAme and/or counteract demethylation activities at these regions (Fig. 1C, E, and F).

Finally, groups 5 & 6 displayed mid-to-high DNAme in pSg (~30%–80%) and *Ctrl* sperm (75%–100%). However, despite a significant decrease in DNAme in *DKO* sperm, these regions maintained mid-levels of DNAme (25%–75%), suggesting that DNMT3A/DNMT3B confer de novo activity, while other DNA methyltransferases (such as *Dnmt1* which is highly expressed in spermatogonia (Supplementary Fig. 3D)), may act on these regions as well (Fig. 1E, F). In general, over ~65% of GC-rich regions (groups 1, 3 & 5) are located within a genic feature and over 40% of them reside in a vicinity of a CpG island (Supplementary Fig. 2E, F). In summary, these results identify a function for the DNMT3A and/or DNMT3B enzymes in de novo depositing and preserving DNAme during spermatogenesis.

### *Dnmt3a* and *Dnmt3b* safeguard and de novo catalyze DNAme in spermatogonial stem cells and differentiated cells respectively
We then set out to identify the enzymes responsible for the developmental dynamics of de novo DNAme acquisition during

spermatogenesis. We FACS isolated spermatogonia and sperm from *Ctrl*, *3aKO*, *3bKO* and *DKO* animals based on DNA content and expression of cell surface markers E-cadherin (CDH1/ CD324), alpha6-integrin (ITGA6/ CD49f) and stem cell growth factor receptor (cKIT/ CD117)[48–50] (Supplementary Fig. 3A). Flow cytometry analysis showed that the relative abundances of undifferentiated spermatogonia (SgU), expressing CDH1$^{high}$, ITGA6$^{high}$, cKIT$^{low}$, and of differentiated spermatogonia (SgD), expressing CDH1$^{low}$, ITGA6$^{low}$, cKIT$^{high}$, were similar between *Ctrl* and *DKO* animals (Supplementary Fig. 3B). Immunofluorescence staining of sorted cells recapitulated the histological findings, with higher expression of DNMT3A and DNMT3B detected in SgD compared to SgU (Supplementary Fig. 3C). RNA-seq analysis confirmed the purity of samples based on expression of general and differentiation specific germ cell markers and the absence of somatic cell markers from the FACS sorted germ cells (Supplementary Fig. 3D–F). Interestingly, RNA-seq analysis identified hardly any significantly differentially expressed genes between *DKO* and *Ctrl* spermatogonia (Supplementary Fig. 3G, H), corroborating that loss of DNMT3A/DNMT3B expression does not majorly affect spermatogonial development.

We next performed Reduced Representation Bisulfite Sequencing (RRBS) using DNA extracted from the FACS isolated spermatogonial populations (SgU and SgD) and mature sperm from *Ctrl*, *3aKO*, *3bKO* and *DKO* animals. We assayed DNAme from two biological replicates for each sample for 12.4% to 17.3% of genomic CpGs to a mean combined coverage ranging from 23.4X to 32.1X, which showed high correlation to each other (Supplementary Fig. 4A). Then we calculated DNAme levels in 214,798 500 bp non-overlapping genomic tiles which contained at least 5 CpGs and were covered by minimally 20 sequencing reads and performed differential methylation analysis with stringent thresholds (methylation difference >20% and FDR < 0.05). We identified 4961 (in SgU cells) and 2485 (in SgD cells) *Dnmt3a*-specific tiles that significantly lost DNA methylation in *3aKO* versus *Ctrl*, but not in *3bKO* versus *Ctrl*. Similarly, we identified 16 (in SgU cells) and 830 (in SgD cells) *Dnmt3b*-specific tiles that significantly lost DNA methylation in *3bKO* versus *Ctrl*, but not in *3aKO* versus *Ctrl* (Supplementary Fig. 4B–E). Additionally, *Dnmt3b*-specific tiles were more enriched in intragenic regions, particularly at exons (Supplementary Fig. 4F), and did not display any sequence motif preference, in line with a transcription-coupled recruitment of the enzyme[51]. For *Dnmt3a*-specific tiles, we detected slight overrepresentation of ELK3 and ELF2/4/5 transcription factor motifs, along with hexamers containing the CCGG tetramer, possibly reflecting the MspI biased restriction enzyme sites detected in the RRBS experiment, suggesting that also DNMT3A does not exhibit any genomic feature or sequence motif specificity (Supplementary Fig. 4G).

Due to the lower genomic coverage of the RRBS experiment we detected 1987 of 20,353 hypoDMRs (9.7%) previously identified in *DKO*

versus *Ctrl* sperm by EM-seq (Supplementary Fig. 5A). Focusing on these hypoDMRs in *Ctrl* samples, they gained more DNAme when SgU differentiated to SgD, than during subsequent development from SgD to sperm (Supplementary Fig. 5B). This was particularly significant when analyzing changes at individual hypoDMRs (Supplementary Fig. 5 Ai). For all six groups, these DNAme dynamics during spermatogenesis were well captured within the first two dimensions of a principal component analysis (PCA) (Fig. 2A). We conclude that postnatally, most de novo DNAme occurs during differentiation of spermatogonia, prior to their entry into meiosis and haploid differentiation[4]. Consistently, deletion of both *Dnmt3a* and *Dnmt3b* *(DKO)* resulted in considerable reduction of DNAme in all cell types examined (Fig. 2B–D, Supplementary Fig. 5Aii), abolishing any gains in DNAme during spermatogenesis (Supplementary Fig. 5C). Accordingly, the PCA showed a high similarity between the methylomes of all *DKO* germ cell types while they differed majorly from those of *Ctrl* samples (Fig. 2A).

To identify enzyme specific roles of DNMT3A and DNMT3B at HypoDMRs during spermatogenesis, we studied methylome changes in single mutants. Intriguingly, we measured gains of DNAme in *3aKO* SgD and sperm relative to *3aKO* SgU particularly obvious in GC-rich groups 1, 3 & 5 (Supplementary Fig. 5D), while *Dnmt3b* depleted SgU, SgD and sperm show comparable levels of DNAme (Supplementary Fig. 5E). Indeed, PC analysis showed that the DNAme states in all three *3bKO* cell types were similar to each other, while *3aKO* SgU methylome differed from *3aKO* SgD and sperm methylomes (Fig. 2A). These data argue that de novo DNAme acquisition during spermatogonial differentiation is primarily catalyzed by DNMT3B rather than DNMT3A.

In addition, we noticed that *Dnmt3a* single depletion, similar to *DKO*, led to decreased DNAme at many hypoDMRs in SgU, particularly obvious in groups 1, 2, 5 & 6, which display intermediate to high DNAme in *Ctrl* cells (Fig. 2B and Supplementary Fig. 5Aiii). In contrast, *Dnmt3b* depletion in SgU lead to a marginal decrease of DNAme in groups 1 and 5 (Fig. 2B), with only a minor fraction of hypoDMRs displaying statistically significant reduction of DNAme (Supplementary Fig. 5Aiv). Accordingly, the DNAme status of *3aKO* SgU was largely similar to *DKO* SgU in the PCA, while the methylome of *3bKO* SgU was mostly similar to that of *Ctrl* SgU (Fig. 2A). Hence, these data show that DNMT3A serves a more prominent role than DNMT3B in depositing DNAme in SgU cells, thereby safeguarding the DNAme levels set prior to or during SgU formation.

Nonetheless, a comparison of methylomes between single and double mutant samples supports the notion that the two enzymes serve some overlapping functions during spermatogenesis (Fig. 2C, D and Supplementary Fig. 5Aiii, Aiv). In summary, during postnatal spermatogenesis we identified a major role of DNMT3A to safeguard basal DNAme levels at hypoDMRs in SgU, while DNMT3B and to some extent DNMT3A catalyze de novo DNAme at hypoDMRs in response to differentiation cues in SgD.

## Stable H3K36me3 marking around hypoDMRs in *Ctrl* and *DKO* spermatogonia

Next, we investigated possible mechanisms of DNMT3A/DNMT3B recruitment to chromatin. Previous studies showed that both enzymes are recruited to chromatin by their PWWP domains interacting with nucleosomes that have been di- or trimethylated on H3K36 by the histone methyltransferases NSD1 and NSD2 or SETD2, respectively[47,51–54]. H3K36me3 is co-transcriptionally deposited within transcribed genes by SETD2 interacting with RNA polymerase II[55,56]. Approximately 8% and 40% of hypoDMRs ($N = 1'613$ and $8'027$, respectively) reside close to promoters or within genes respectively, that are abundantly expressed in SgU and SgD cells (Supplementary Fig. 6A). These genes serve basic cellular or developmental processes rather than spermatogenesis-specific functions (Supplementary Fig. 6B). To investigate a possible role of H3K36 methylation in

DNMT3A/DNMT3B recruitment, we generated genome wide profiles of H3K36me3 by ultra-low input native ChIP-seq (ULI-NChIP-seq)[57] for 2 biological replicates of SgU and SgD cells from *Ctrl* and *DKO* animals (Supplementary Fig. 7A). In *Ctrl* samples, H3K36me3 occupancy levels increased or got reduced during spermatogonial differentiation at a comparable number of genomic tiles. HypoDMRs behaved similarly as other genomic tiles (Supplementary Fig. 7Bi, Ci). In SgU and SgD *DKO* samples, 2-fold more regions gained H3K36me3 signal compared to *Ctrl* samples (Supplementary Fig. 7Bii, Biii). HypoDMRs behaved, however, similarly as nonDMR regions (Supplementary Fig. 7Cii, Ciii). Globally, we noticed an increased H3K36me3 signal at the *Dnmt3b*-specific tiles compared to the *Dnmt3a*-specific tiles arguing that primarily DNMT3B may be targeted to gene bodies via an interaction of its PWWP domain with H3K36me3 (Supplementary Fig. 7D, E).

Irrespective of the developmental stage or *Dnmt3a/Dnmt3b* pro- or deficiency, hypoDMRs were generally embedded in regions with moderate H3K36me3 occupancy, as compared to intragenic regions of genes characterized by high to low H3K36me3 levels, matching their high to low expression states (Supplementary Fig. 8A–D). Contrasting typical CpG-rich promoters with extensive H3K36me3 depletion, we measured only a minor depletion of H3K36me3 at the centers of hypoDMR regions in GC-rich groups 1 and 3 which exhibit low DNAme in *Ctrl* SgU. GC-rich group 5 hypoDMRs characterized by higher DNAme in *Ctrl* SgU showed an even more uniform H3K36me3 signal (Supplementary Fig. 8E, F). We conclude that H3K36me3 at and around hypoDMRs may suffice for DNMT3A/DNMT3B recruitment in SgU and SgD cells. It is, however, unlikely to specify DNAme gain within the hypoDMRs during spermatogonial differentiation.

## HypoDMRs convert from H3K4me3 to DNAme positive states during spermatogonial differentiation

Contrary to H3K36 methylation, H3K4me3 is known to inhibit DNMT3 dependent catalysis DNAme by blocking the interaction between DNMT3A's ADD domain with unmodified H3 N-termini[37–40]. To identify possible roles of H3K4me3 in hypoDMRs, we generated genome wide profiles of H3K4me3 by ULI-NChIP-seq[57] for 2 biological replicates of SgU and SgD cells from *Ctrl* and *DKO* animals (Supplementary Fig. 9A). In *Ctrl* SgU, hypoDMRs of mainly groups 1-4 displayed moderate H3K4me3 levels, comparable to levels of non-CGI promoters (Fig. 3A, B, and F, Supplementary Fig. 10A–C). These hypoDMRs had low-to-intermediate DNAme levels in the *Ctrl* SgU samples (Fig. 2A). In contrast, hypoDMRs with high DNAme levels, belonging to groups 5 & 6, were devoid of H3K4me3 in *Ctrl* SgU (Fig. 3A, B). Differential occupancy analysis showed a global reduction of H3K4me3 in *Ctrl* SgD relative to *Ctrl* SgU (Supplementary Fig. 9Bi). Statistical testing demonstrated that GC-rich hypoDMRs showed the highest enrichment among those that had lost H3K4me3 in *Ctrl* SgD versus *Ctrl* SgU (Fig. 3C). More specifically, during the transition from SgU to SgD, H3K4me3 levels became reduced at hypoDMRs gaining DNAme (Fig. 3A, B, and F, Supplementary Fig. 10D), suggesting that H3K4me3 is removed during spermatogonial differentiation prior to DNAme acquisition.

When comparing H3K4me3 occupancy in *DKO* versus *Ctrl* samples in SgU, we observed that GC-poor hypoDMRs were over-represented among tiles with increased H3K4me3 in *DKO* (Fig. 3D, Supplementary Fig. 9Bii, Cii). In contrast, we observed significant enrichment of GC-rich hypoDMRs among tiles with increased H3K4me3 in *DKO* SgD (Fig. 3E, Supplementary Fig. 9Biii, Ciii). The lack of de novo DNAme in *DKO* SgD (relative to *Ctrl* SgD) was evidently associated with increased H3K4me3 levels especially in GC-rich groups 1 & 3 (Fig. 3A, B, and F, Supplementary Fig. 10E). Notably, such changes in H3K4me3 occupancy during development and in response to *Dnmt3a/Dnmt3b* deficiency occur in absence of transcriptional misregulation (Supplementary Fig. 3G, H). In summary, the data indicate that H3K4me3 methyltransferase enzymes (possibly in conjunction

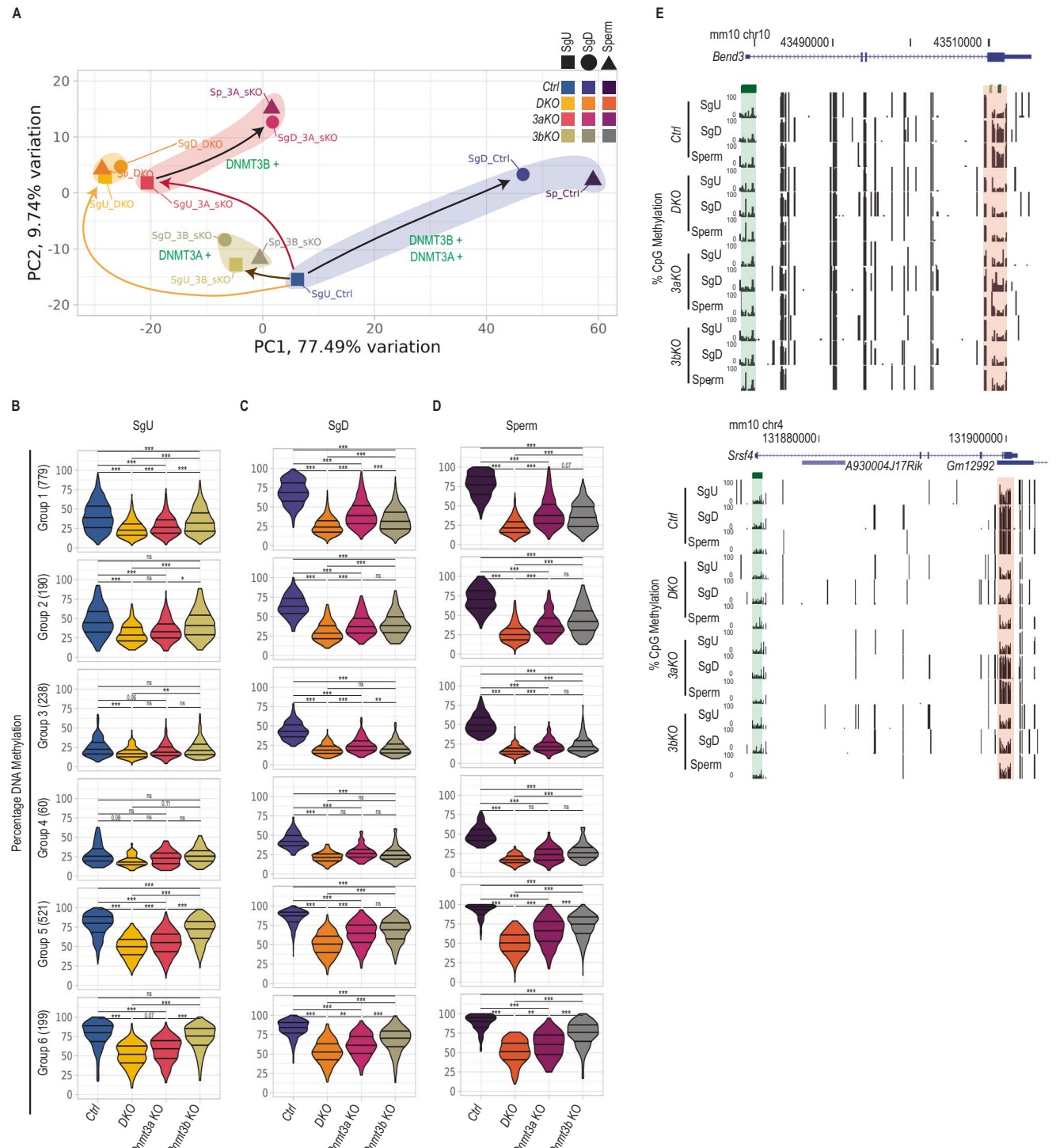

**Fig. 2 | DNAme dynamics of hypoDMRs during spermatogonial differentiation upon *Dnmt3a* and/or *Dnmt3b* deletion. A** Principal component analysis (PCA) of DNAme levels at hypoDMRs in SgU, SgD and sperm samples from *Ctrl, 3aKO, 3bKO* and *DKO* animals. **B–D** Violin plots showing percentages DNAme of hypoDMRs assessed by RRBS in (**B**) undifferentiated spermatogonia (SgU) for *Ctrl* (n = 2), *3aKO* (n = 2), *3bKO* (n = 2) and *DKO* (n = 2) samples, (**C**) differentiated spermatogonia for *Ctrl* (n = 2), *3aKO* (n = 2), *3bKO* (n = 2) and *DKO* (n = 2) samples and (**D**) sperm for *Ctrl* (n = 2), *3aKO* (n = 2), *3bKO* (n = 2) and *DKO* (n = 2) samples. Two-sample, two-sided Wilcoxon tests were performed between indicated groups ns $P > 0.05$, *$P < 0.05$, **$P \leq 0.01$, ***$P \leq 0.001$. Center lines in violin plots represent median values while upper and lower lines indicate the interquartile range (IQR; from the 25th to 75th percentile). **E** UCSC genome browser snapshot of the *Bend3* and *Srsf4* loci, showing tracks for DNAme levels in SgU, SgD and sperm in *Ctrl, 3aKO, 3bKO* and *DKO* samples. The promoter CGIs and the hypoDMR are highlighted in green and red respectively.

with H3K4 demethylase enzymes) versus DNMT3 enzymes mutually antagonize each other in differentiating spermatogonia e.g., by inhibiting DNMT3 catalytic activity versus preventing recruitment of KMT2A, KMT2B and SETD1A/CXXC1 H3K4 methyltransferases to methylated CpG-rich sequences.

## DNAme reduces nucleosome retention at GC-rich regions during spermiogenesis

During spermatid differentiation and nuclear compaction, a small fraction of nucleosomes evades removal and replacement by protamines[15]. We and others reported previously higher nucleosomal

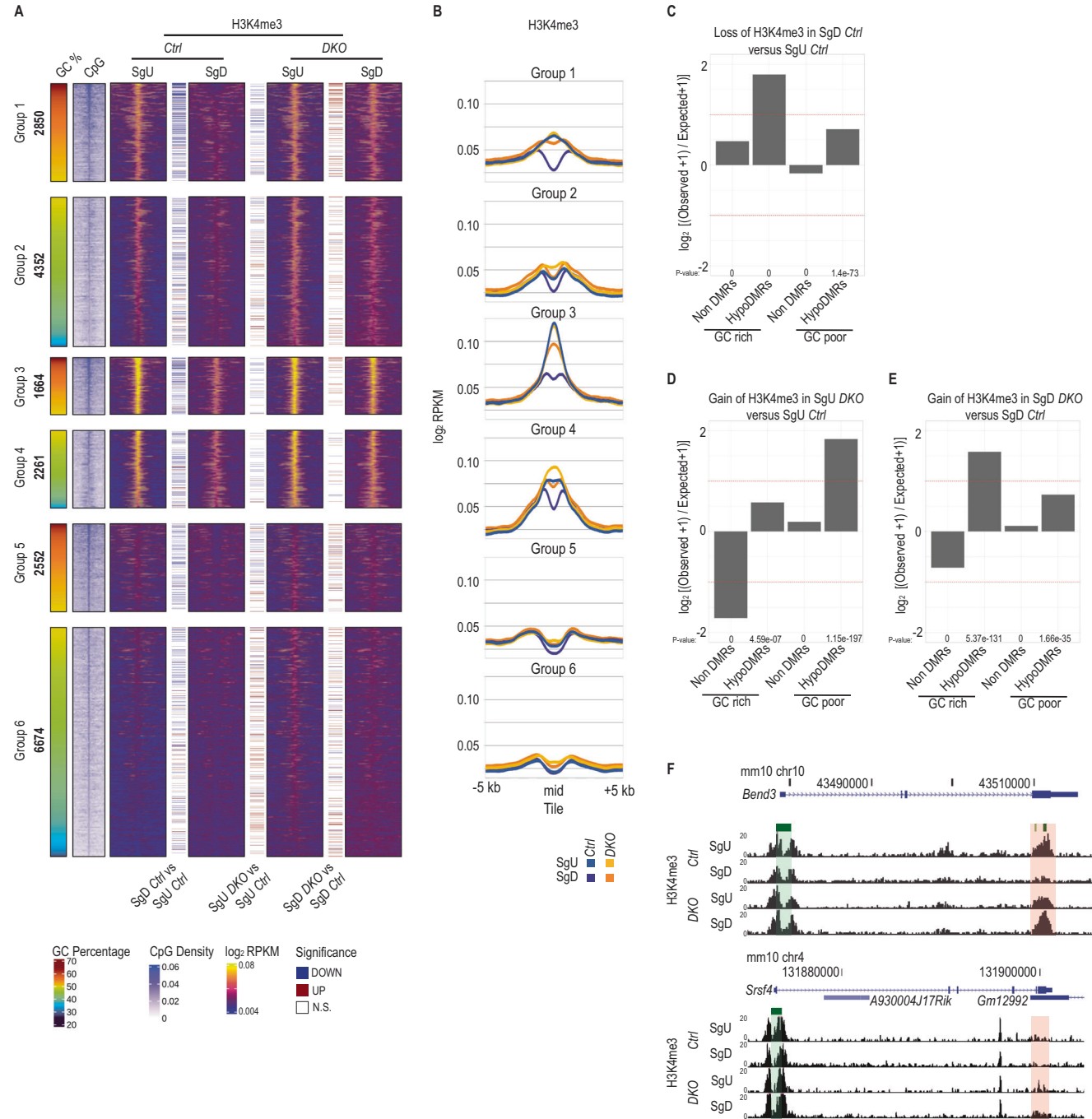

**Fig. 3 | HypoDMRs in *DKO* spermatogonia retain H3K4me3. A** Heatmaps showing H3K4me3 signal at the center of hypoDMRs (±5 kb) assessed by ULI-NChIP-seq in *Ctrl* SgU (*n* = 2), *Ctrl* SgD (*n* = 2), *DKO* SgU (*n* = 2) and *DKO* SgD (*n* = 2). Colored lines between heatmaps denote hypoDMRs with significantly differential H3K4me3 occupancy in the indicated contrasts. The GC percentage and CpG density at hypoDMRs (±5 kb) are indicated. **B** Line plots showing average H3K4me3 signal for each group of hypoDMRs (±5 kb) assessed by ULI-NChIP-seq in *Ctrl* SgU (*n* = 2), *Ctrl* SgD (*n* = 2), *DKO* SgU (*n* = 2) and *DKO* SgD (*n* = 2). **C** Bar plots showing log$_2$ enrichments of GC-rich or GC-poor nonDMR and hypoDMR tiles among tiles with significant H3K4me3 loss in *Ctrl* SgD versus *Ctrl* SgU. *P*-values were calculated by

two-sided Fisher's exact test. **D** Bar plots showing log$_2$ enrichments of GC-rich or GC-poor nonDMR and hypoDMR tiles among tiles with significant H3K4me3 gain in *DKO* SgU versus *Ctrl* SgU. *P*-values were calculated by two-sided Fisher's exact test. **E** Bar plots showing log$_2$ enrichments of GC-rich or GC-poor nonDMR and hypoDMR tiles among tiles with significant H3K4me3 gain in *DKO* SgD versus *Ctrl*. *P*-values were calculated by two-sided Fisher's exact test. **F** UCSC genome browser snapshot of the *Bend3* and *Srsf4* loci, showing tracks for H3K4me3 occupancy in *Ctrl* and *DKO* SgU and SgD. The promoter CGIs and the hypoDMR are highlighted in green and red respectively.

occupancy in sperm chromatin at CpG-rich sequences lacking DNAme[17,18,21]. To assess whether DNAme may restrain nucleosome retention during spermiogenesis, we performed native ChIP-seq[23] with an antibody specific to nucleosomes[58] using 2 biological replicates of *Ctrl* and *DKO* sperm (Supplementary Fig. 11A). We detected 45'125 and

50'354 anti-Nucleosome peaks, covering 1.2% and 0.8% of the genome, with a median length of 508 bp and 295 bp in *Ctrl* and *DKO* sperm, respectively (Supplementary Fig. 11B–D). Relating to the DNAme levels, low DNAme hypoDMRs of groups 3 & 4 displayed nucleosomal occupancy, with more signal detected in GC-rich group 3 in *Ctrl* sperm.

In contrast, the nucleosome signal was low or absent at hypoDMRs belonging to high DNAme groups 1, 2, 5, and 6 (Fig. 4A, B). Genome wide differential enrichment analysis between the two genotypes showed comparable numbers of regions with increased or decreased nucleosome occupancy (Supplementary Fig. 11E). However, GC-rich hypoDMRs, but not nonDMRs were significantly overrepresented among tiles with increased nucleosomal occupancy in *DKO* versus *Ctrl*

sperm samples (Fig. 4C, Supplementary Fig. 11F). Comparing *Ctrl* and *DKO* sperm, more GC-rich group 1 hypoDMRs displayed elevated nucleosome occupancy than GC-poor group 2 hypoDMRs, although both hypoDMR groups exhibiting comparable relative reductions in DNAme (Fig. 4A, B, and D, Supplementary Fig. 11G). Smaller increases in nucleosomal occupancy were detected for GC-rich hypoDMRs in groups 3 & 5, possible due to the presence of nucleosomes at group 3

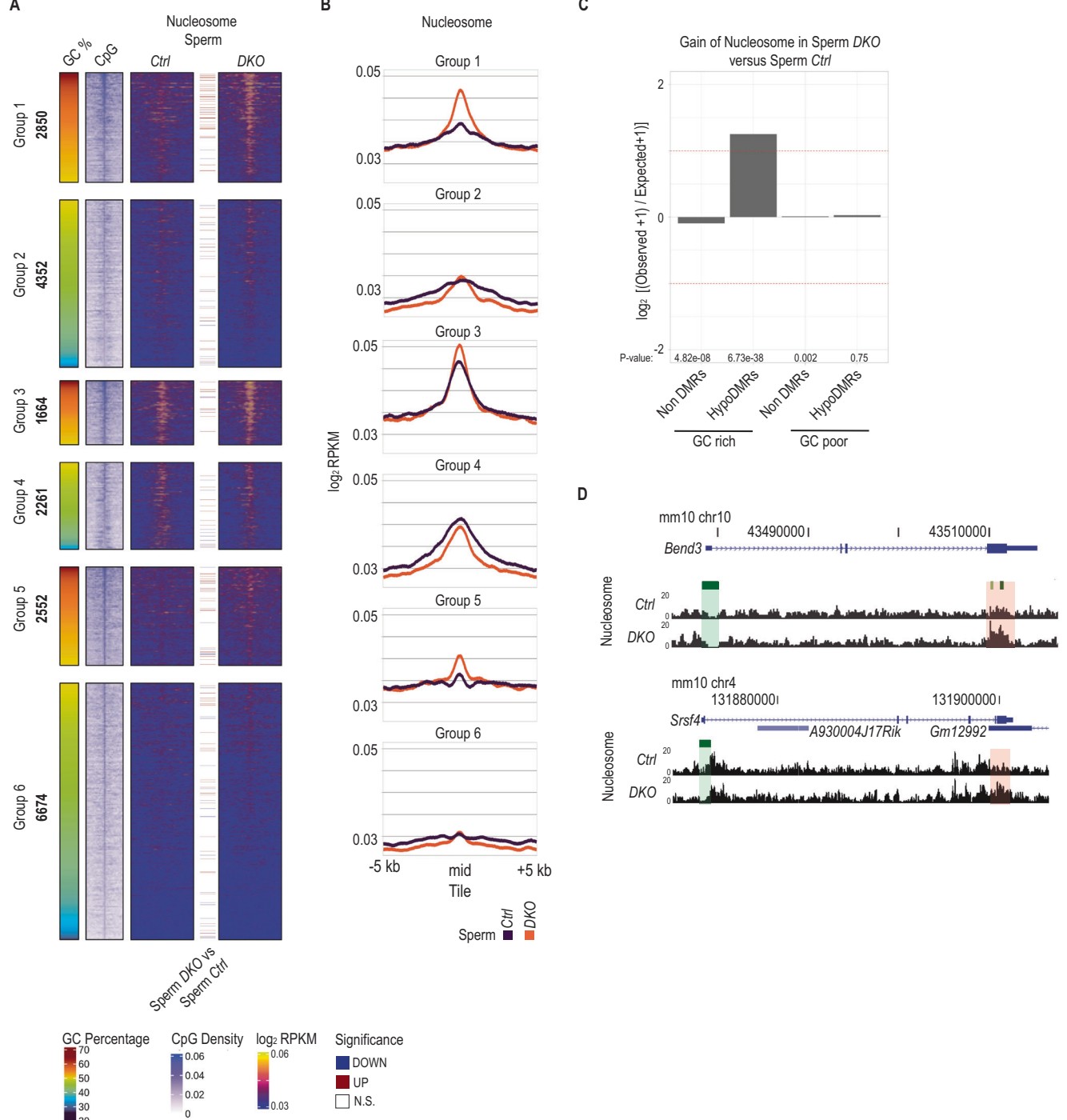

**Fig. 4 | HypoDMRs gain nucleosomes in *DKO* sperm. A** Heatmaps showing nucleosome signal at the center of hypoDMRs (±5 kb) assessed by ChIP-seq in *Ctrl* (*n* = 2) and *DKO* (*n* = 2) sperm samples. Colored lines between heatmaps denote hypoDMRs with significantly differential nucleosome occupancy between indicated samples. The GC percentage and CpG density at hypoDMRs (± 5 kb) are indicated. **B** Line plots showing average nucleosome signal for each group of hypoDMRs (± 5 kb) assessed by ChIP-seq in *Ctrl* (*n* = 2) and *DKO* (*n* = 2) sperm

samples. **C** Bar plots showing log₂ enrichments of GC-rich or GC-poor nonDMR and hypoDMR tiles among tiles with significant nucleosome gain in *DKO* sperm versus *Ctrl*. *P*-values were calculated by two-sided Fisher's exact test. **D** UCSC genome browser snapshot of the *Bend3* and *Srsf4* loci, showing tracks for nucleosome occupancy in *Ctrl* and *DKO* sperm. The promoter CGIs and the hypoDMR are highlighted in green and red respectively.

hypoDMRs already in *Ctrl* samples and due to the remaining of intermediate levels of DNAme at group 5 hypoDMRs (Fig. 4A, B, Supplementary Fig. 11G). GC-poor group 4 & 6 hypoDMRs showed similar nucleosome occupancies between genotypes. Together, these data support a model in which a high proportion of unmethylated CpG dinucleotides within more GC-rich sequences enables nucleosome retention during spermiogenesis.

## Increased H3K4me3 deposition at hypoDMRs in *DKO* sperm

Given the antagonistic relationship between H3K4me3 and DNAme states in spermatogonia (Fig. 3) and other cell types[59–61], we generated genome wide profiles of H3K4me3 by ChIP-seq[23] for 2 biological replicates of *Ctrl* and *DKO* sperm (Supplementary Fig. 12A). We detected 26'654 and 29'166 anti-H3K4me3 peaks, covering 1.5% and 1.7% of the genome, with a median length of 950 bp and 953 bp in *Ctrl* and *DKO* sperm respectively (Supplementary Fig. 12B–D). In *Ctrl* sperm, low DNAme hypoDMRs of groups 3 & 4 with some nucleosomal occupancy (Fig. 4) also displayed signal for H3K4me3. The H3K4me3 signal was low or absent at hypoDMRs belonging to high DNAme groups 1,2,5,6 (Fig. 5A, B). Genome wide differential enrichment analysis showed comparable numbers of regions with increased or decreased H3K4me3 occupancy (Supplementary Fig. 12E). Nonetheless, GC-rich but also GC-poor hypoDMRs were significantly overrepresented among tiles with increased H3K4me3 occupancy in *DKO* versus *Ctrl* sperm samples (Fig. 5C, Supplementary Fig. 12F). Particularly group 1 hypoDMRs, but also group 3 and 5 hypoDMRs, displayed increased H3K4me3 enrichment in *DKO* sperm compared to *Ctrl* sperm (Fig. 5A, B, and D). Notably, ~30% (*N* = 154) of regions with increased nucleosome signal displayed significantly increased H3K4me3 (Supplementary Fig. 12G). These regions showed the greatest reduction of DNAme and the highest gain of nucleosome signal in *DKO* sperm (Supplementary Fig. 12H, I). Inversely, 87% (*N* = 1'077) of H3K4me3 enriched regions displayed no change in nucleosome occupancy (Supplementary Fig. 12G). The latter likely reflects a higher immunoprecipitation efficiency by anti-H3K4me3 over anti-Nucleosome antibodies, also in line with a wider distribution of anti-H3K4me3 versus anti-Nucleosome signals around the center of hypoDMRs (Figs. 4A, B and 5A, B). Increased H3K4me3 occupancy measured in *DKO* sperm may result from enhanced chromatin recruitment of H3K4 methyltransferase(s) via unmethylated CpG dinucleotides[62,63] in spermatogonia.

## Sperm born DNAme prevents H3K4me3 deposition at hypoDMRs in mouse early embryos

To investigate whether DNAme in sperm affects chromatin establishment in embryos, shortly after fertilization, we developed a highly sensitive chromatin profiling approach named Antibody TArgeted Tagmentation Assay combined with sequencing (ATATA-seq). This assay uses strategies similarly used in CUT&Tag[64] and Stacc-seq[65]. It differs, however, (i) in the type of chimeric protein used, (ii) the high salt buffer conditions used during the targeting of antibody-ZZ-Tn5 transposomes to chromatin to limit the targeting to open chromatin, and (iii) the absence of washing steps. Specifically, we generated a Tn5 chimeric protein consisting of two Z domains of staphylococcal protein A fused to the Tn5 transposase (ZZ-Tn5). Next, the fusion protein is loaded with Tn5 sequencing adapters and incubated with antibodies. Finally, the "antibody-ZZ-Tn5 assembled and active transposomes" are applied to permeabilized nuclei to recognize antibody targeted regions and to insert sequencing adapters into the DNA surrounding the chromatin target sites.

We used ATATA-seq to profile H3K4me3 in triplicate pools of 50 early 2-cell stage genetically hybrid embryos, generated by fertilizing JF1/Ms background oocytes with *Ctrl* or *DKO* sperm on a C57BL/6 background allowing parental allelic discrimination based on the presence of single nucleotide polymorphisms (SNPs). The biological

replicates of ATATA-seq correlate better with anti-H3K4me3 ChIP seq of 2-cell stage embryos rather than ICM datasets from Liu et al.[66], validating the sensitivity of our approach (Supplementary Fig. 13A). In *Ctrl* sperm derived embryos H3K4me3 occupancy was predominantly detected at hypoDMRs in groups 3 & 4 and moderately in groups 1,2,5 (Fig. 6A, B). H3K4me3 occupancy at the maternal alleles was higher compared to the paternal alleles (Fig. 6C–F). Genome wide differential enrichment analysis between *Ctrl* or *DKO* sperm derived embryos showed comparable numbers of regions with increased or decreased H3K4me3 occupancy, with most of the differences arising at paternal alleles rather than at maternal alleles (Supplementary Fig. 13B). However, GC-rich hypoDMRs were significantly overrepresented among tiles with increased H3K4me3 occupancy in *DKO* versus *Ctrl* sperm-derived embryos (Fig. 6Gi). Importantly, when splitting the maternal and paternal alleles similar enrichment was observed for nonDMRs among tiles with increased H3K4me3, however more than 2 fold enrichment of GC-rich hypoDMRs was evident only for the paternal allele (Fig. 6Gii, Giii, Supplementary Fig. 13C).

Examination of total (non-allelic discrimination) ATATA-seq signal at hypoDMRs revealed an increased H3K4me3 signal primarily at all GC-rich groups in *DKO* sperm-derived embryos (Fig. 6A, B, and H). Upon assignment of the parental origin of ATATA-seq reads, we observed that H3K4me3 signals in the maternal genome were largely unchanged in regions examined between embryos generated by sperm of *Ctrl* or *DKO* males (Fig. 6C, D, H). In striking contrast, the paternal genome demonstrated differential enrichments comparable to those observed in the non-allelic data. E.g., paternal alleles from embryos fathered by *DKO* sperm displayed increased H3K4me3 signals at hypoDMRs in GC-rich groups 1,3,5 and moderate increases at group 2,4,6 hypoDMRs, compared to those from *Ctrl* sperm-derived embryos, which had only low occupancy levels (Fig. 6E, F, H). The magnitude of H3K4me3 changes in GC-rich hypoDMRs in *DKO* sired embryos associated to the extent of DNAme loss (Supplementary Fig. 13D). These observations indicate that low DNAme in sperm renders paternal alleles in early embryos permissive for H3K4me3 deposition.

## Hypo-DNAme rather than H3K4me3 occupancy in sperm is associated with H3K4me3 deposition in paternal embryonic chromatin

We next asked whether differential enrichment of H3K4me3 at hypoDMRs in *DKO* sperm leads to changes in H3K4me3 deposition at paternal alleles in 2-cell stage embryos. To address this question, we grouped all hypoDMRs based on increased (log$_2$ FC > 0.5), unaltered (-0.5 <log$_2$ FC < 0.5) or reduced (log$_2$ FC < -0.5) H3K4me3 occupancies in *DKO* versus *Ctrl* sperm and their GC content (GC-poor <50% GC content <GC-rich). Then we quantified corresponding differential H3K4me3 occupancies in embryos (Fig. 7A, B). The group of GC-rich hypoDMRs, which gained H3K4me3 in *DKO* sperm, showed increased H3K4me3 in *DKO* sperm-derived embryos (specific to the paternal allele) (Fig. 7A, B). Here, we cannot distinguish whether the changes in embryonic chromatin are inherited, or de novo deposited. The groups of GC-rich hypoDMRs which remained unaltered or had lost H3K4me3 in *DKO* sperm showed, however, comparable increases in H3K4me3 in *DKO* sperm-derived embryos (specific to the paternal allele), strongly pointing towards elevated de novo H3K4me3 deposition in the embryo in absence of paternal DNAme (Fig. 7A, B). All groups of GC-poor hypoDMRs, irrespective of the differential enrichment of H3K4me3 in sperm, displayed unaltered H3K4me3 levels in *DKO* sperm-derived embryos compared to *Ctrl* sperm-derived embryos (Fig. 7A, B). Analyzing single GC-rich hypoDMRs we observed that only a minor fraction of hypoDMRs showed gain of H3K4me3 in both *DKO* sperm and *DKO* sperm-derived embryos (Fig. 7C). This indicates that the gain of H3K4me3 in *DKO* sperm does not directly specify gain of H3K4me3 deposition in embryos. Instead, our data suggest that the presence of DNA methylation in sperm mostly restricts H3K4me3

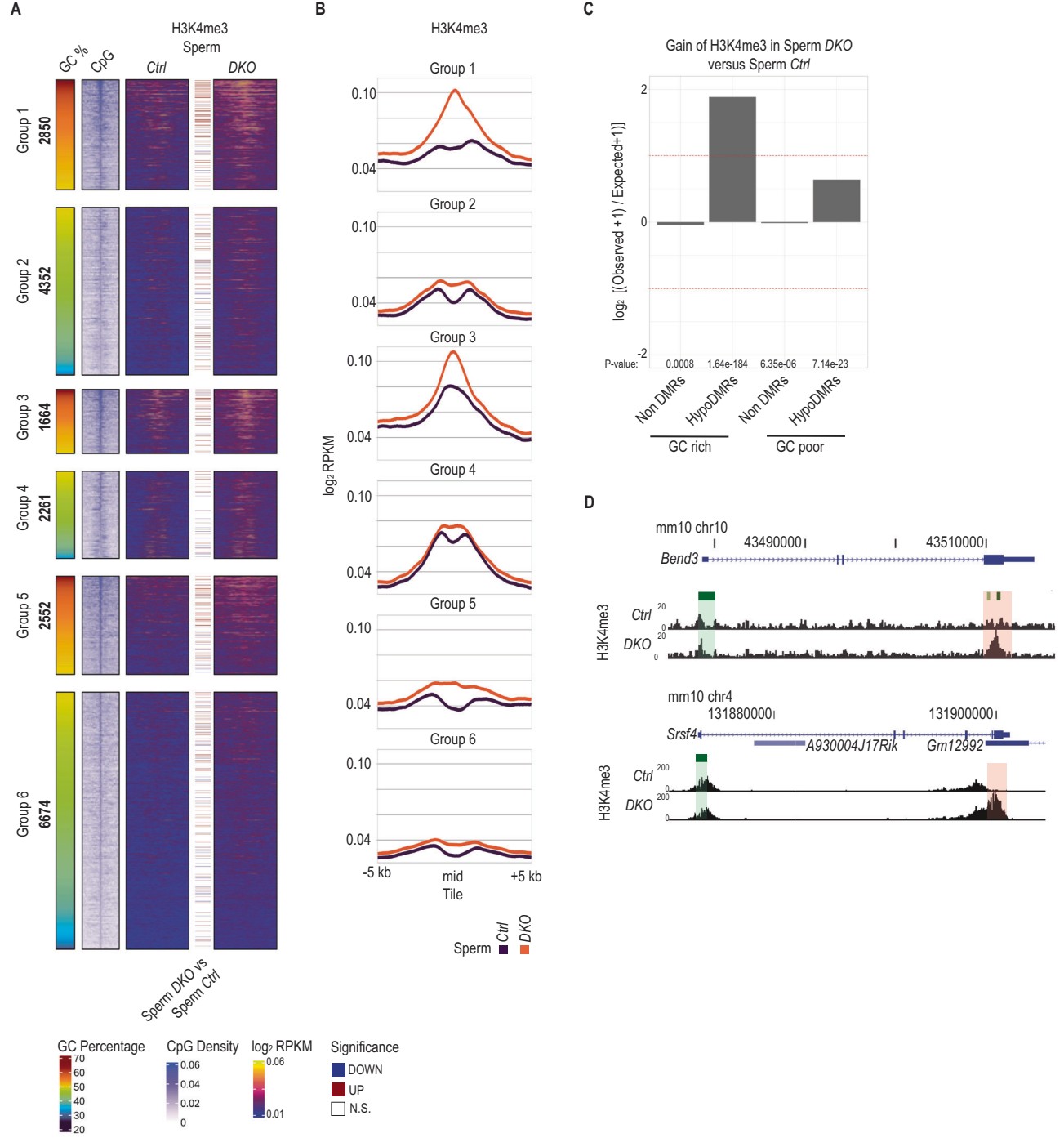

**Fig. 5 | HypoDMRs gain H3K4me3 in *DKO* sperm. A** Heatmaps showing H3K4me3 signal at the center of hypoDMRs ( ± 5 kb) assessed by ChIP-seq in *Ctrl* (*n* = 2) and *DKO* (*n* = 2) sperm samples. Colored lines between heatmaps denote hypoDMRs with significantly differential nucleosome occupancy between indicated samples. The GC percentage and CpG density at hypoDMRs ( ± 5 kb) are indicated. **B** Line plots showing average H3K4me3 signal for each group of hypoDMRs (±5 kb) assessed by ChIP-seq in *Ctrl* (*n* = 2) and *DKO* (*n* = 2) sperm

samples. **C** Bar plots showing log₂ enrichments of GC-rich or GC-poor nonDMR and hypoDMR tiles among tiles with significant H3K4me3 gain in *DKO* sperm versus *Ctrl* sperm. *P*-values were calculated by two-sided Fisher's exact test. **D** UCSC genome browser snapshot of the *Bend3* and *Srsf4* loci, showing tracks for H3K4me3 occupancy in *Ctrl* and *DKO* sperm. The promoter CGIs and the hypoDMR are highlighted in green and red respectively.

deposition at GC-rich regions in the paternal chromatin of embryos, rather than H3K4me3 in sperm promoting its deposition in embryos.

### Absence of altered transcription around hypoDMRs in *DKO*-sperm-sired early embryos

As described above *DKO* males sired similar numbers of healthy offspring without evident phenotypic abnormalities as compared to *Ctrl*

males (Supplementary Fig. 1G). Nonetheless, to identify possible functional consequences of an altered paternal DNA methylome on transcription in early embryos, we performed RNA-sequencing analysis of single early 4-cell stage embryos after the occurrence of zygotic genome activation (*n* = 21 for *Ctrl* and *n* = 20 for *DKO* sperm derived embryos). Following developmental pseudo-time calculations, we selected age-matched embryos to perform differential gene

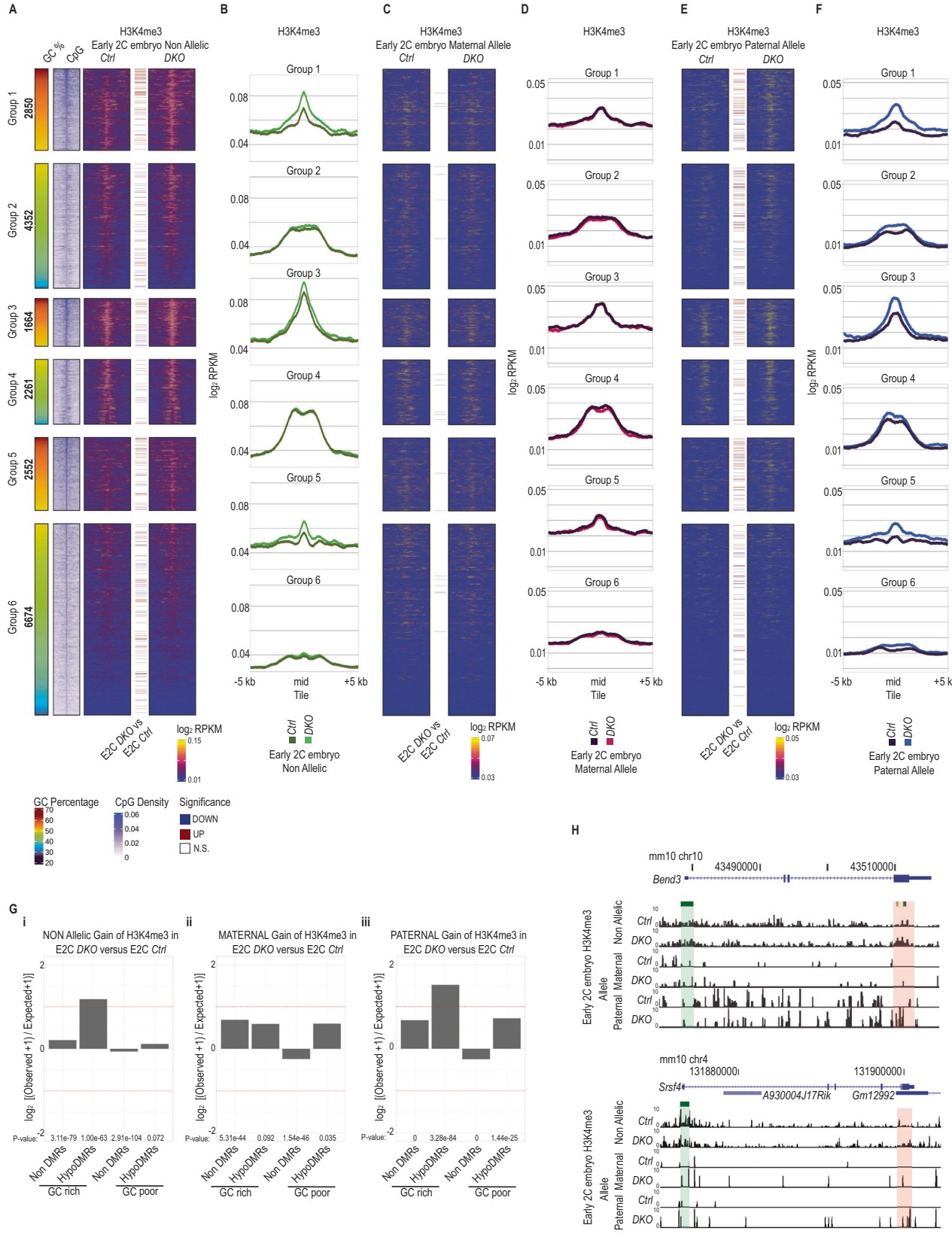

expression analysis ($n=7$ for *Ctrl* and $n=7$ for *DKO* sperm derived embryos) (Supplementary Fig. 14A). We identified only a handful of differentially expressed genes, which notably didn't overlap with genes containing hypoDMRs (Supplementary Fig. 14B–D).

To identify non-gene related transcriptional changes we quantified Smart-Seq2 reads around hypoDMRs and stratified them according to their expression levels in *Ctrl* embryos into 4 groups. We detected a trend towards increased transcription in lowly expressed regions in *DKO* sperm-derived embryos compared to *Ctrl*. However, this trend was evident in both paternal and maternal alleles suggesting that these transcriptomic alterations may be indirect consequences of the altered DNA methylome in *DKO* sperm (Supplementary Fig. 14E).

**Fig. 6 | HypoDMRs acquire H3K4me3 in early mouse embryos generated by *DKO* sperm. A** Heatmaps showing H3K4me3 signal at the center of hypoDMRs (±5 kb) assessed by ATATA-seq in early 2-cell (E2C) embryos generated by *Ctrl* (*n* = 3) or *DKO* (*n* = 3) sperm. Colored lines between heatmaps denote hypoDMRs with significantly differential H3K4me3 occupancy between indicated samples. The GC percentage and CpG density at hypoDMRs (±5 kb) are indicated. **B** Line plots showing average H3K4me3 signal for each group of hypoDMRs (±5 kb) assessed by ATATA-seq in early 2-cell (E2C) embryos generated by *Ctrl* (*n* = 3) or *DKO* (n = 3) sperm. **C–E** Heatmaps showing H3K4me3 signal of maternal (**C**) or paternal (**E**) alleles at the center of hypoDMRs ( ± 5 kb) assessed by ATATA-seq in early 2-cell (E2C) embryos generated by *Ctrl* (n = 3) or *DKO* (n = 3) sperm. Colored lines between heatmaps denote hypoDMRs with significantly differential H3K4me3 occupancy between indicated samples. The GC percentage and CpG density at hypoDMRs (±5 kb) are indicated. **D–F** Line plots showing average H3K4me3 signal at maternal (**D**) or paternal (**F**) alleles for each group of hypoDMRs (±5 kb) assessed by ATATA-seq in early 2-cell (E2C) embryos generated by *Ctrl* (*n* = 3) or *DKO* (*n* = 3) sperm. **G** Bar plots showing log$_2$ enrichments of GC-rich or GC-poor nonDMR and hypoDMR tiles among tiles with significant H3K4me3 gain at (i) all alleles, (ii) maternal alleles and (iii) paternal alleles in early 2-cell (E2C) embryos generated by *Ctrl* or *DKO* sperm. *P*-values were calculated by two-sided Fisher's exact test. **H** UCSC genome browser snapshot of the *Bend3* and *Srsf4* loci, showing tracks for H3K4me3 occupancy non-allelically, or at maternal or paternal alleles in *Ctrl* and *DKO* sperm. The promoter CGIs and the hypoDMR are highlighted in green and red respectively.

---

The absence of evidence for direct transcriptional mis-regulation may be due to that the majority of hypoDMRs reside in genomic regions unlikely to serve regulatory functions. For those that could confer regulation, the cofactors necessary to drive their expression may not be present at the 4-cell stage, when we performed our transcriptional profiling.

## Discussion

### Molecular cues of intergenerational epigenetic inheritance

Our study demonstrates that the DNAme state in sperm impacts on the establishment of chromatin states at the paternal genome in early embryos. Sperm borne DNAme at GC-rich regions (Fig. 1F) constrains de novo deposition of H3K4me3 at paternal alleles in early embryos (Fig. 6E, F and 7D), while its absence at such regions is permissive. Interestingly, global DNAme present in sperm undergoes rapid removal upon fertilization in zygote embryos, except at ICRs and certain repetitive element families[67]. Reanalysis of publicly available methylome data shows that most hypoDMR sequences identified in this study undergo such rapid DNAme removal, when paternally inherited (Supplementary Fig. 15A–C)[25]. In contrast, maternally speaking, the same hypoDMRs methylated in mature oocytes undergo only protracted, presumably replication-related, loss of DNAme during pre-implantation development. Intriguingly, GC-rich hypoDMRs gain H3K4me3 in wild type early embryos, concurrently to DNAme removal (Supplementary Fig. 15D, E)[66]. Given that the extensive erasure of DNAme from the paternal genome occurs prior to replication in one-cell embryos[68,69], we hypothesize that the de novo H3K4me3 deposition at hypoDMRs in paternal chromatin of *DKO* sperm-derived embryos occurs even prior to or concurrent with the wave of global DNA demethylation, possibly through precocious recruitment of H3K4 methyltransferases to CpG-rich unmethylated chromatin. Most other CpG-rich sequences, frequently found at gene promoters that remain largely unmethylated in sperm may undergo a similar process, mechanistically and timing wise. Intriguingly, H3K4 methylation catalyzed on maternally provided H3.3 histones that become incorporated into the paternal genome is essential for the onset of minor ZGA in the paternal pronucleus and subsequent preimplantation development[26]. In contrast to the distinct narrow peaks of H3K4me3 on promoters in sperm, paternal alleles in zygotes exhibit large domains of H3K4me3[70,71], arguing for de novo activity of the KMT2A, KMT2B and/ or SETD1B/CXXC1 enzymes recognizing unmethylated GC-rich regions via their CXXC domains, possibly followed by spreading *in cis*[72–75]. Analogously to paternal transmission, it will be important to determine whether low levels of DNAme in oocyte genome primes H3K4me3 deposition not only in the oocyte[72] but also on maternal chromatin in the embryo.

A multitude of studies investigated the role of H3K4me3 in diet-induced phenotype models of epigenetic inheritance and suggested that paternal H3K4me3 is transmitted to the embryo and influences gene expression and development[32,76]. While we detected differential H3K4me3 enrichments at a subset of GC-rich hypoDMRs in sperm, we did not observe a direct correlation of such differential enrichments to

differential H3K4me3 enrichments observed at many more hypoDMRs in two-cell embryos (Fig. 7A, B). Thus, our data argues that H3K4me3 is de novo deposited in early zygotes in response to the absence of sperm borne DNAme, rather than instructed by sperm-borne H3K4me3. Some hypoDMRs with clear gains in H3K4me3 in *DKO* sperm exhibited gains of H3K4me3 signal in *DKO* sperm-derived early embryos. Hence, we cannot exclude the possibility that high H3K4me3 occupancy in sperm may potentially enhance the deposition of H3K4me3 by e.g. the maternally provided SETD1B/CXXC1 enzyme complex reading unmethylated CpG dinucleotides and H3K4me3 via the CXXC and PHD domains of the CXXC1 protein, respectively[77–80]. PHD-finger domain specific maternal loss-of-function studies of *Setd1b/Cxxc1*, or alternatively of *Kmt2a* or *Kmt2b* may contribute to deciphering the contribution of paternal H3K4me3-bearing nucleosomes for paternal inheritance of a H3K4me3 state. Irrespectively, our data support the model that the absence of DNAme serves as an effective intergenerational signal, enabling early H3K4me3 deposition in the next generation.

Over the past years, an increasing body of studies has investigated changes of sperm DNAme in mediating paternal epigenetic inheritance of acquired phenotypes[81,82]. Molecular modes of inheritance remain, however, unclear. For example, changes in sperm DNAme upon paternal cigarette smoking were not recapitulated in offspring pre-frontal cortex tissues[83]. In another study, aged male mice exhibited local hypoDNAme in sperm at sites enriched for REST/NRSF binding motif. These males sired progeny with aberrant neuronal gene expression in forebrains during their embryonic development, which was linked to REST/NRSF motifs. At adulthood, the progeny displayed abnormal vocal communication patterns[84]. Building upon our results providing a proof of concept directly linking changes in DNAme at specific CGIs in sperm to an altered H3K4me3 chromatin state in early embryos, the differential impact of REST/NRSF occupancy on DNAme and H3K4me3 in paternal transmission may be an interesting avenue to pursue further[85,86].

### Dynamics of DNAme and interplay with chromatin modifications

We show that the de novo DNMT3B and DNMT3A enzymes fulfill distinct non-redundant functions in de novo DNAme and even in safeguarding DNAme levels during adult spermatogenesis, respectively. Our FACS sorting protocol enabled us to isolate undifferentiated spermatogonia (SgU), including self-renewing spermatogonial stem cells (SSC), and differentiated spermatogonia (SgD). Surprisingly in SgU, hypoDMRs with mid-to-high levels of DNAme were more sensitive to depletion of DNMT3A than DNMT3B (Supplementary Fig. 5B), suggesting that DNMT3A is required to safeguard DNAme. DNMT3A may possibly counteract TET activity in SgU[87] and/or augment inefficient maintenance by DNMT1. Based on previous studies[3–5,88], global de novo DNAme in male germ cells appears to be largely completed by postnatal day 0.5 (P0.5). Conditional depletion of DNMT3A using *Cre* drivers active during PGC development (e.g. *Prdm1-Cre* or *Tnap-Cre*) resulted in differentiation and genomic imprinting defects of

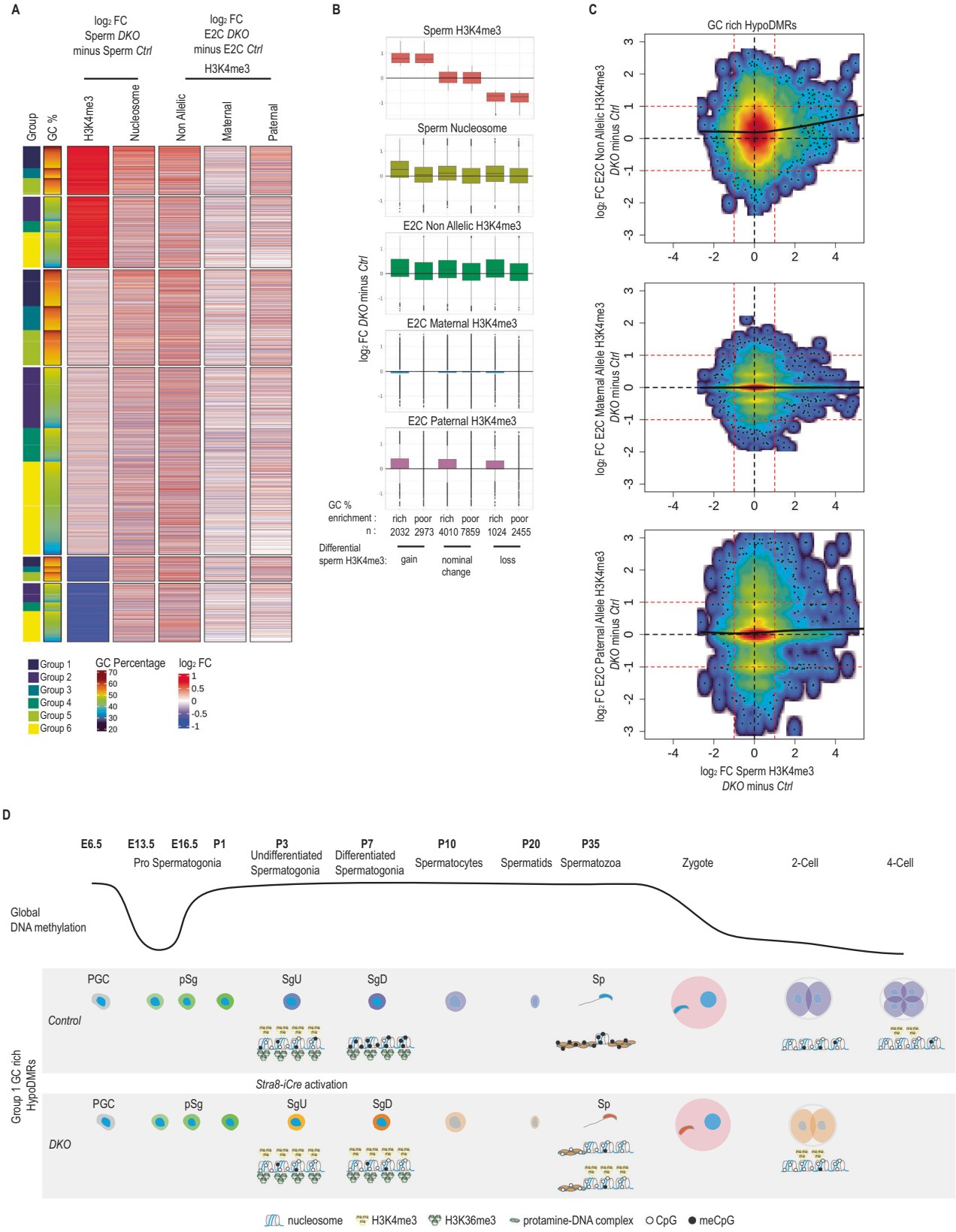

spermatogonia after birth. In these studies, *Dnmt3a* deficiency resulted in a greatly hypomethylated genome of spermatogonia[7,10]. Contrasting PGC-expressed Cre-drivers, expression of the *Stra8-iCre* transgene was detected by postnatal day 3 (P3) but not at E18.5[45]. Hence, we used *Stra8-iCre* to conditionally delete *Dnmt3a* and *Dnmt3b* during postnatal spermatogenesis, a developmental period after

which global DNAme patterns had already been established. Given that global DNAme levels were not affected in our model, we interpret that DNAme was genome-wide normally established in prospermatogonia and subsequently properly maintained by DNMT1 in spermatogonial stem cells[43]. Nonetheless, we cannot entirely exclude the possibility of precocious expression of the CRE recombinase in some perinatal

**Fig. 7 | Comparison of intergenerational H3K4me3 changes at hypoDMRs.**
**A** Heatmap showing the differential H3K4me3 and nucleosome occupancy between *DKO* and *Ctrl* sperm samples and differential H3K4me3 in E2C embryos generated by *DKO* and *Ctrl* sperm for Group 1 hypoDMRs. Group 1 hypoDMRs were clustered in 3 subclusters according to H3K4me3 differential enrichments between *DKO* and *Ctrl* sperm samples. **B** Boxplots showing the H3K4me3 $\log_2$ fold changes between *DKO* and *Ctrl* sperm or E2C embryos generated by *DKO* and *Ctrl* sperm for the 3 subclusters of Group 1 hypoDMRs. The boxes extend from the first quartile (Q1) to the third quartile (Q3) of the data distribution, with a line in the box representing the median. Whiskers indicate variability outside Q1 and Q3, extending to the minimum and maximum values within 1.5 times the interquartile range (IQR), while points outside this range represent outliers. **C** Scatterplots showing

$\log_2$ fold change of H3K4me3 signal between *DKO* minus *Ctrl* sperm (x-axis) versus $\log_2$ fold change of H3K4me3 signal between early 2-cell (E2C) embryos generated by *Ctrl* or *DKO* sperm at all alleles (top), maternal alleles (middle) and paternal alleles (bottom) in (y-axis) for the GC-rich hypoDMRs (Groups 1, 3, 5). Black lines represent LOESS weighted linearly regressed data. **D** Schematic representation of genome wide (upper solid line) and Group1 HypoDMRs DNA methylation and chromatin dynamics during murine male germline development and early embryogenesis. Undifferentiated spermatogonia (SgU), differentiated spermatogonia (SgD) and sperm first appear during juvenile period (P1 to P35), but are present throughout adulthood. PGC: primordial germ cells, pSg: prospermatogonia, Sp: Spermatozoa.

prospermatogonia, impairing the completion of most de novo DNA methylation in such cells. Indeed, conversion of some prospermatogonia into spermatogonial stem cells may be impaired in our model[7], potentially explaining the increased presence of seminiferous tubules devoid of germ cells and reduced testis weights that we had observed for several *3aKO*, *3bKO* and many *DKO* males (Supplementary Fig. 1E–G).

In contrast, DNAme levels at hypoDMRs in DNMT3B depleted SgUs resembled those in *Ctrl* SgUs suggesting minimal contribution of this enzyme in undifferentiated spermatogonia. However, the DNAme profile at hypoDMRs in DNMT3B depleted SgDs resembled those measured in *Ctrl* and DNMT3B-depleted SgU cells, suggesting that the developmental gain of DNAme at hypoDMRs during spermatogonial differentiation is defective (Fig. 2B, Supplementary Fig. 5E). In addition, when SgUs depleted for DNMT3A differentiate to SgD we observed de novo gains of DNAme which are mediated by DNMT3B, albeit towards altered levels due to the abnormal initial DNAme status existing in DNMT3A-depleted SgU. Previously it was shown that DNMT3B preferentially located at gene bodies of transcribed genes in ESCs[51], possibly through interactions between its PWWP domain and the H3K36me3 modification, deposited by the RNA polymerase II interacting histone methyltransferase SETD2. Like in ESCs, it is plausible that in male germ cells, genic DNA methylation is also deposited by DNMT3B as we found that the *Dnmt3b*-specific tiles were more enriched in H3K36me3 signal compared to the *Dnmt3a*-specific tiles (Supplementary Fig. 7D, E). Nevertheless, genes that contain hypoDMRs were transcriptionally active both at SgU and SgD stages (Supplementary Fig. 6A), displaying already mid-high levels of H3K36me3 in SgUs (Supplementary Fig. 8A), posing the question why high DNAme levels are established only in SgD. The lowly DNA methylated hypoDMRs are embedded in broader H3K36me3 domains in both cell types, however they are highly enriched in H3K4me3 at the SgU stage (Fig. 3A). We postulate a constant tug of war for recruitment of DNMT3 enzymes and subsequent establishment of DNAme via the H3K36me3-rich flanks of hypoDMRs and a repulsion of DNMT3 enzymes from the H3K4me3-rich center of hypoDMRs at SgU stage, which is resolved by the SgD stage with the establishment of high DNAme (Fig. 7D). Given that *Dnmt3c* is rather low expressed in undifferentiated or differentiated spermatogonia (Supplementary Fig. 3D) compared to prospermatogonia around E16.5[9], and that DNMT3C lacks a PWWP domain and has been implicated in methylating young retrotransposon elements[9], its role in postnatal DNA methylation of gene body associated and H3K36me3 embedded hypoDMRs may be limited. In female mice, demethylation of H3K4 by LSD2 (KDM1B) is critical for establishing the DNAme imprints during oogenesis, pointing out that H3K4 demethylases are genetically upstream of DNAme[89]. H3K4 demethylases such as LSD2 and KDM5B are recruited to gene bodies by NPAC and MRG15 respectively - two putative H3K36me3 readers[90,91]. In oocytes, an interplay between H3K4me3, H3K36me3 and DNAme was observed where loss of H3K36me3 led to failure in establishing the correct DNA methylome, and invasion of H3K4me3 and H3K27me3 into former H3K36me3 territories[56]. Additionally, LSD1

(KDM1A) binds in vitro to DNMT1 and DNMT3B[92] and functionally impact on DNAme levels in mouse ESCs[93]. In analogy to oocytes, we propose that H3K4 demethylases may reprogram H3K4me3 at hypoDMRs during spermatogonial differentiation. Intriguingly, H3K4me3 occupancy persisted at hypoDMRs in SgD in the absence of the DNMT3A/DNMT3B DNAme machinery, suggesting that these enzymes may regulate recruitment of H3K4 demethylases and/or that the absence of DNAme enables H3K4 methyltransferases recruitment and H3K4me3 deposition (Fig. 3A). Examining the effect of abnormal H3K4me3 reprogramming on DNAme in the male germ line may, however, be challenging as loss of Lsd1 or Kmt2b in spermatogonia results in a developmental arrest[94,95].

### Sperm chromatin and intergenerational epigenetic inheritance

We previously inferred from MNase protection and ChIP assays for histones and their modifications the genomic localization of nucleosomes/histones in mature mouse and human sperm[18,21]. We observed an inverse correlation between high DNAme and nucleosome occupancy at CpG-rich sequences[21]. Subsequent MNase studies reported the presence of nucleosomes in intergenic regions and repetitive elements[19,96]. Experimental conditions aggravated by a complex chromatin structure in sperm nuclei are likely at the core of these inconsistencies. Among others, they may originate from variability in pretreatments, degrees of protamine reductions and extractions, enzymatic digestions required to solubilize sperm chromatin, and from antibody pulldown efficiencies[15,23,97]. Moreover, given the low levels of histones in sperm, purity of spermatozoa samples is critical[18,22,23]. Consequently, the distributions and the relative abundances of nucleosomes in mammalian sperm genomes remain unclear.

Here, by FACS sorting highly pure sperm cell samples, as indicated by our DNAme measurements at maternal ICRs, and comparing nucleosome ChIP assays on different genetic backgrounds of wild type and *Dnmt3a/Dnmt3b* deficient sperm allowed us to examine the impact of DNAme status on nucleosome occupancy at various genomic sequence composition contexts in a comparative and germ cell specific manner. As before[21], we measured that normally methylated GC-rich regions exhibited low nucleosome signal in *Ctrl* sperm. In *Dnmt3a/Dnmt3b* mutant spermatozoa, the same regions that had become hypomethylated had increased nucleosome signals (Fig. 7D). We measured only marginal increases in nucleosome signals at GC-poor regions that had become hypomethylated in the mutant or in GC-rich regions that showed only small reduction of DNAme. Together, these data support the model that indeed the combination of GC-richness and low DNAme promote nucleosome retention in sperm (Fig. 4A).

The ChIP-seq methods employed in this study examine relative nucleosome enrichments in large bulk populations of sperm limiting our understanding of the heterogeneity among spermatozoa that would contain nucleosomes at a given genomic region. Future studies examining sperm chromatin variability at a single cell resolution will be required to provide more quantitative insights to the extent by which the presence or absence of DNAme would modulate nucleosome eviction during spermiogenesis. Importantly, beyond modulating

sperm chromatin, this study provides a molecular paradigm of how the DNA sequence and its methylation status in mature male gametes define the establishment of the chromatin landscape at the paternal genome in the early embryo in terms of H3K4me3 deposition.

## Methods

### Mice and early embryos collection

Animal housing, handling and procedures of mice conformed to the Swiss Animal Protection Ordinance (protocol numbers: 2569, 2670, 3183) and are compliant with the FMI institutional guidelines. Mice were housed in Type II long cages containing aspen bedding on IVC racks in rooms with 15–20 air changes per hour and a 12 h light/dark cycle. Temperature was maintained within 20–24 °C with a relative humidity within 45–65%. Food and water were provided ad libitum. Mice of 129SvJae × C57BL/6 hybrid background bearing *Dnmt3a*[flox] (*Dnmt3a*[tm3.1Enl]) and *Dnmt3b*[flox] (*Dnmt3b*[tm5.1Enl]) alleles[98] were provided by Dr. En Li. Mice of C57BL/6 background bearing the iCre expressing transgene under the control of *Stra8* promoter (*Tg(Stra8-iCre)1Reb*)[45] were obtained from The Jackson Laboratory (RRID:IMSR_JAX: 017490). The JF1/MsJ (Japanese fancy mouse 1) inbred strain[99] was purchased from The Jackson Laboratory (RRID:IMSR_JAX:003720). C57BL/6JRj mice were purchased from Janvier Labs. To generate the *Dnmt3a*; *Dnmt3b*; *Stra8-iCre* conditional mice, we first intercrossed mice from the single *Dnmt3a*[fl/fl] and *Dnmt3b*[fl/fl] colonies to generated double *Dnmt3a*[fl/fl]; *Dnmt3b*[fl/fl] homozygote females which were then crossed with males from *Stra8-iCre* colony to generate *Dnmt3a*[fl/+]; *Dnmt3b*[fl/+]; *Stra8-iCre* male mice. *Dnmt3a*[fl/+]; *Dnmt3b*[fl/+]; *Stra8-iCre* males were backcrossed to C57BL/6JRj females to create *Dnmt3a*[-/+]; *Dnmt3b*[-/+]; *Stra8-iCre* males. *Dnmt3a*[-/+]; *Dnmt3b*[-/+]; *Stra8-iCre* males were then mated with *Dnmt3a*[fl/fl]; *Dnmt3b*[fl/fl] homozygote females to generate double or single conditional mutants *Dnmt3a*[-/fl]; *Dnmt3b*[-/fl]; *Stra8-iCre* (DKO), *Dnmt3a*[-/fl]; *Dnmt3b*[+/fl]; *Stra8-iCre*, (3aKO), *Dnmt3a*[+/fl]; *Dnmt3b*[-/fl]; *Stra8-iCre* (3bKO), and double heterozygote *Dnmt3a*[+/fl]; *Dnmt3b*[+/fl]; *Stra8-iCre* (HET) or wildtype *Dnmt3a*[+/fl]; *Dnmt3b*[+/fl] (Ctrl) male experimental mice. Single conditional mutant mice for one Dnmt3 paralogue are heterozygous for the other. Testis samples were collected from 3–12 month-old animals.

To obtain early embryos, 7–10 week-old JF1 female mice were injected intraperitoneally with HyperOva (Cosmo Bio, KYD-010-EX-X5) 63 hours pre collection and human chorionic gonadotropin (hCG, Chorulon MSD) 15 hours pre collection. Oviducts were dissected and oocytes from 2 animals were collected in 30 µl drops of HTF medium (EmbryoMax® Human Tubal Fluid Millipore, MR-070-D). Sperm was collected from 2 cauda epididymides in 50 µl drops of HTF medium. 2–5 µl of sperm sample was transferred to the oocyte drops and incubated for 3 hours at 37 °C. Zygotes with visible pronuclei washed 3 times in HTF medium and transferred to KSOM medium (Millipore, MR-106-D) and cultured in low oxygen chamber filled with the mixture gas (5% O2/ 5% CO2). Early 2 cell stage embryos were collected 22 hours post fertilization. Early 4 cell stage embryos were collected 50–55 hours post fertilization.

### Fluorescence Activated Cell Sorting (FACS) of spermatogonia and sperm

Testicular cell suspension was prepared by incubation of seminiferous tubules in 200 U/ml Collagenase type I (Worthington Biochemical LS004196), 5 µg/ml DNAse I (Roche 10104159001), and 0.05%Trypsin (Gibco 25200056) in GBSS (Sigma G9779) as described in ref. 100. Cells were stained with 1/200 anti-CD117/c-kit PE (eBioscience 12-1171-83), anti-CD324/E-Cadherin eFluor 660 (eBioscience 50-3249-82) and anti-CD49f/Integrin alpha 6 PE-Cyanine7 (eBioscience 25-0459-82) for 1 hour at 32 °C with constant shaking and protected from light. After washing cells were incubated with 20 µg/ml with Hoechst 33342 (Thermo Fischer Scientific H3570) for 1 hour at 32 °C with constant shaking and protected from light and then with 30 nM DRAQ7 (Biostatus DR71000) for 5 min, on bench. The stained cells were strained

through a 40 µm nylon filter into 5 ml polypropylene tubes and were sorted on a BD FACSAria III cell sorter fitted with a 70 µm nozzle (Becton Dickinson) using a 375 nm laser to excite Hoechst 33342, a 561 nm laser to excite PE and PE-Cy7 and a 633 nm laser to excite DRAQ7 and eFluor600. Cells were first gated for FSC and SSC to exclude debris. Then the live cells were selected based on the absence of DRAQ7 signal detected using 755LP, 780/60BP. Then we gated for cells positive for Hoechst 33342 emission which was detected using a 670LP (Hoechst-Red) and 450/20 BP (Hoechst-Blue). We gated for the Hoechst-Red[low] and Hoechst-Blue[mid] population which is enriched for 2 N spermatogonia. Fluorescence for PE was detected using a 582/15BP filter, for PE-Cy7 was detected using 735LP,780/60BP filter and for eFluor660 using a 660/20BP filter. Undifferentiated spermatogonia were sorted as CD324[high], CD49f[high], CD117[low] and differentiated spermatogonia were sorted as CD324[low], CD49f[low], CD117[high].

For sperm cell sorting, mouse cauda epididymides were dissected into a Petri dish and fat patches were removed with forceps and scissors. Each epididymis was punctured with a needle and carefully squeezed into 100 ul of PBS with the help of two forceps. Sperm cell suspension was transferred into an Eppendorf tube and was allowed to liquefy at room temperature for 5 min. To break sperm tails, cells were briefly sonicated with a Brandson Tip digital sonicator with 10% amplitude and 3 cycles of 0.5 sec ON / 2 sec OFF. Cells were stained with 2 ul/ml Hoechst 33342 (H3570, ThermoFisher) for 1 h at 25 °C with constant shaking and protected from light. Cells were filtered through a 40 µm Nylon filter into 5 ml polypropylene tubes prior to sorting into PBS with a BD FACS Aria III instrument. Cells were first gated for FSC and SSC to exclude debris. Then sperm cells were sorted according to their 1 C DNA content.

### Reduced Representation Bisulfite Sequencing (RRBS)

Pellets of 1 million sperm cells were incubated in 200 µl of DNA extraction buffer (20 mM Tris-HCl pH 7.5, 300 mM NaCl, 2% SDS, 10 mM EDTA, 200 µg/ml Proteinase K) for 16 hours at 55 °C. An equal volume of phenol:chloroform:isoamyl alcohol (25:24:1) solution was added to the samples. After a vigorous vortexing for 30 s and centrifugation at 16,000 g for 10 min, the upper aqueous phase was transferred to new tubes. 0.05 µg/µl glycogen, 1/10 volume of 3 M sodium acetate and 2.5 volumes of 100% ethanol were added, and the samples were incubated for 2 h at −80 °C. DNA precipitated by centrifugation at 16,000 g for 30 min at 4 °C. DNA pellet was washed once with 70% ethanol and resuspended in 20 µl. DNA was further purified using 2 volumes of Ampure XP beads (Beckman Coulter). DNA was extracted from 10,000 undifferentiated or differentiated spermatogonia cell samples by QIAamp DNA Micro Kit (Qiagen # 56304) according to the manufacturer's instructions. We prepared RRBS libraries using 10-20 ng of DNA and following the previously published method[101]. Briefly, DNA was digested with 0.9 µl MspI Fast Digest (Thermo Scientific FD0544) in 1X Tango buffer in a volume of 18 µl for 6 h at 37 °C. DNA ends were repaired by adding 1 µl Klenow Fragment, exo− (Thermo Scientific EP0421) and 1 µl of nucleotide end-repair mix (1 mM dATP; 0.1 mM dGTP; 0.1 mM dCTP prepared in 2X Tango buffer) and incubated for 45 min at 37 °C, followed by enzyme heat inactivation at 75 °C for 15 min. Then 6.25 nM of 5mC sequencing adapter (NEB E7535S) was ligated with 1 µl T4 DNA Ligase, HC (Thermo Scientific EL0013) and 0.5 µl ATP solution (Thermo Scientific R0441) for 16 hours at room temperature. The looped adaptor ligated DNA fragments were subjected to USER (NEB M5508) digestion at 37 °C for 20 min and used directly for bisulfite treatment using the Imprint DNA Modification Kit (Sigma MOD50) according to the manufacturer's 2-step modification procedure. The modified DNA was amplified with 13 cycles of PCR using Kapa U+ polymerase (Roche 07959052001) and indexed primers (NEB) and purified with 1.8X Ampure XP beads. Libraries were sequenced on a HiSeq 2500 sequencing platform generating 50 bp single end reads, per the manufacturer's recommendations.

## Whole genome methylation sequencing

Genomic DNA (100 ng per reaction) was sheared to an average fragment size of 400 bp using the Covaris S220 instrument. The sheared DNA was used as input for the NEBNext Enzymatic Methyl-seq (NEB E7120) following the manufacturer's instructions except for doubling the reaction incubation times. The libraries were amplified for 5 cycles and sequenced in 2 × 75 bp paired-end mode with NextSeq 500 sequencing technology (Illumina), per the manufacturer's recommendations.

## Chromatin Immunoprecipitation (ChIP)

To profile histone modifications in spermatogonial cells we followed the previously published ultra-low input native ChIP-seq (ULI-NChIP-seq) protocol[57] with minor modifications. Briefly 5000-7500 spermatogonia were FACS-sorted directly in 20 µl nuclear isolation buffer (Sigma-Aldrich NUC101) in Eppendorf® LoBind microcentrifuge tubes. The nuclei were centrifuged at 500 g for 5 min at 4 °C in a swing bucket centrifuge and the volume of the sample was reduced to 10 µl. The sample was snap frozen in liquid nitrogen and stored at −80 °C freezer until further processing. The sample was thawed on ice and 1 µl of TS buffer (5% Triton X-100, 5% Sodium deoxycholate) was added for 10 min to ensure permeabilization of nuclei. 40 µl of MNAse digestion buffer (3.33 gelUnits/µl Micrococcal Nuclease (NEB M0247S), 1.25X Micrococcal Nuclease buffer, 6.25% PEG 6000, 0.85 mM DTT) was added to the sample for 10 min at 25 °C to fragment the chromatin. Reaction was stopped by addition of 5 µl 100 uM EDTA and the sample was placed on ice. The sample was diluted by addition of 145 µl ChIP buffer (20 mM Tris-HCl pH 8.0, 2 mM EDTA, 150 mM NaCl, 0.1% Triton X-100, 1X EDTA-free protease inhibitor cocktail). To ensure release of chromatin fragments, the sample was sonicated using the Diagenode Bioruptor Plus® for 3 cycles of 5 s ON and 30 s OFF at low output. Then chromatin was pre-cleared with 10 µl of 1:1 protein A:protein G Dynabeads (Thermo Fisher Scientific, 10001D and 10003D). The precleared chromatin sample was incubated with 0.5 µg of anti-H3K4me3 (Millipore 17-614) or 0.75 µg anti-H3K36me3 (Cell Signaling #4909) conjugated with 10 µl of 1:1 protein A: protein G Dynabeads overnight at 4 °C on a rotator. Chromatin-Antibody-Dynabeads complexes were washed once with 500 µl of ChIP buffer, 3 times with 1 ml of Low Salt wash buffer (20 mM Tris-HCl (pH 8.0), 0.05% SDS, 1% Triton X-100, 2 mM EDTA and 150 mM NaCl) and 3 times with 1 ml High Salt buffer (20 mM Tris-HCl (pH 8.0), 0.05% SDS, 1% Triton X-100, 2 mM EDTA and 250 mM NaCl). Chromatin was eluted in 40 µl of Elution buffer (100 mM NaHCO3 and 1% SDS) for 90 min at 65 °C. DNA from eluted material was purified by 2X volumes of Ampure XP DNA purification beads (Beckman Coulter, A63881) and resuspended in 25 µl nuclease free H2O.

To profile histone modifications in spermatozoa, we followed the previously published native ChIP-seq protocol[23] with modifications as follows. Spermatozoa (500,000 per reaction) were FACS sorted in Eppendorf® LoBind microcentrifuge tubes. The cells were pelleted at 6000 g for 5 min at 4 °C in a benchtop centrifuge. The pellet was snap frozen in liquid nitrogen and stored at −80 °C freezer until further processing. The pellet was thawed on ice and resuspended in 500 µl PBS. 50 mM DTT was added for 2 hours at 21 °C to reduce disulfide bonds of protamine. 100mM N-Ethylmaleinimid was added for 30 min at 21 °C to modify the cysteine residues. The cells were pelleted at 6000 g for 5 min at 4 °C in a benchtop centrifuge and washed once with 600 µl PBS. Cells were resuspended in 50 µl Lysis buffer (20 mM Tris-HCl pH 7.5, 60 mM KCl, 5 mM MgCl2, 0.1 mM EGTA, 300 mM Sucrose, 0.5 mM DTT, 0.25 % Igepal, 0.5% sodium deoxycholate, 1X EDTA-free protease inhibitor cocktail) and incubated for 30 min at 21 °C. 50 µl of MNAse digestion buffer (5 gelUnits/µl Micrococcal Nuclease (NEB M0247S), 2X Micrococcal Nuclease buffer) was added to the sample for 15 min at 37 °C to fragment the chromatin. Reaction was stopped by addition of 5 µl 500 uM EDTA and the sample was centrifuged at 16000 g for 10 min at 4 °C in a benchtop centrifuge. The

supernatant chromatin was pre-cleared with 10 µl of 1:1 protein A: protein G Dynabeads (Thermo Fisher Scientific, 10001D and 10003D). The precleared chromatin sample was incubated with 0.5 µg of anti-H3K4me3 (Millipore 17-614) or 1 µg anti-nucleosome[58] (provided by J. van der Vlag) or 1 µg anti-H3.3 (Cosmo Bio CE-040B) or 1 µg anti-H3.1/2/t[102] (provided by J. van der Vlag) conjugated with 10 µl of 1:1 protein A:protein G Dynabeads overnight at 4 °C on a rotator. Chromatin-Antibody-Dynabeads complexes were washed once with 500 µl of ChIP buffer, 3 times with 1 ml of Low Salt wash buffer (20 mM Tris-HCl (pH 8.0), 0.05% SDS, 1% Triton X-100, 2 mM EDTA and 150 mM NaCl) and 3 times with 1 ml High Salt buffer (20 mM Tris-HCl (pH 8.0), 0.05% SDS, 1% Triton X-100, 2 mM EDTA and 250 mM NaCl). Chromatin was eluted in 40 µl of Elution buffer (100 mM NaHCO3 and 1% SDS) for 90 min at 65 °C. DNA from eluted material was purified by phenol-chloroform, ethanol-precipitation and resuspended in 25 µl nuclease free H2O.

For library construction the ChIP purified DNA was used as input for the NEBNext® Ultra™ II DNA Library Prep Kit for Illumina® (NEB E7645L) with modifications of the manufacturer instruction as we assembled half volume reactions, and we doubled reaction incubation times. For adaptor ligation we used 1:15 adaptor dilution. We amplified the ligated fragments for 12 cycles using single indexed primers. Libraries were pooled and sequenced in 50 bp single-end mode with HiSeq 2500 sequencing technology (Illumina), per the manufacturer's recommendations.

## Antibody Targeted Tagmentation (ATATA)

ATATA-seq was based on the Cut&Tag strategy[64]. We fused N terminally the Tn5 transposase to two tandem IgG binding Z domains to specifically target the ZZ-Tn5 to antibody bound regions. Early 2 cell embryos (50 embryos per reaction) were incubated in 30 ul LIB buffer (20 mM HEPES pH 7.5, 300 mM NaCl, 0.5 mM Spermidine, 0.05% Digitonin, 0.05% Triton X-100, 1X EDTA-free protease inhibitor cocktail) on ice for 10 min for nuclei permeabilization. Active transposomes were generated by incubating ZZ-Tn5 (8 µM) with Mosaic End double-stranded (MEDS) oligonucleotides (12 µM) for 30 min at 37 °C. Active transposomes (0.4 µM) were incubated with anti-H3K4me3 antibody or normal rabbit IgG (1 µM) in LIB buffer for 30 min at 4 °C with continuous mixing. Antibody- ZZ-transposomes complexes (final concentration 0.1 µM) were added to the embryo samples and incubated for 1 hour at 4 °C with continuous mixing to allow targeting to the chromatin. Antibody- ZZ-transposomes were activated by addition of TAB buffer (20 mM HEPES, 5%PEG6000, 7.5 mM MgCl2, 300 mM NaCl) and incubated for 1 hour at 37 °C with continuous mixing. The tagmentation reaction was stopped with addition of Stop buffer (0.2% SDS, 10 mM EDTA, 100ug/ml ProteinaseK) and incubated at 65 °C for 15 min. DNA was purified by phenol-chloroform, ethanol precipitation and libraries were amplified for 16 cycles. The libraries were purified by 0.5X-1.5X volumes left-right side of Ampure XP DNA purification beads and sequenced in 50-bp single-end mode with HiSeq 2500 sequencing technology (Illumina), per the manufacturer's recommendations.

## RNA sequencing

For 4-cell stage embryos RNA-seq libraries were prepared following the previously published Smart-Seq2 method[103]. Each 4-cell stage embryo was collected at 50–55 h after IVF, was washed in PBS + 0,02% PVA, transferred into a well of a 96-well plate containing SS2 Lysis buffer (0,09% Triton-X 100, 0,5U/µl SUPERAseIN (Life Technologies AM2696), 2.5 mM oligodT primer (Microsynth), 2.5 mM dNTP mix (Promega), ERCC RNA Spike-In Mix (1:3.2 × 10^7 Thermo Fischer scientific 4456740)), snap frozen on dry ice and then stored in −80 °C until further use. The sample was incubated at 72 °C for 3 min and immediately put back on ice. Then SS2 RT mix (10 U/µl SuperScript II reverse transcriptase (Thermo Fischer 18064014), 0.25 U/µl SUPERAseIN, 1X SuperScript II first-strand buffer, 5 mM DT, 1 M Betaine (Sigma B0300-1VL), 6 mM MgCl2, 1 mM TSO (Exiqon)) was added to the sample and

incubated for 90 min at 42 °C, then 10 cycles of 2 min at 50 °C and 2 min at 42 °C and finally for 15 min at 70 °C. Sample was preamplified for 16 cycles using 1X KAPA HiFi HotStart ReadyMix (KAPA Biosystems KK2602) and 0.1 µM ISPCR primers (Microsynth). DNA was purified using 1X volumes of Ampure XP DNA purification beads. 1 ng of pre-amplified DNA was added to Tn5 tagmentation mix (1x TAPS-DMF buffer, homemade Tn5 (1:1,200)) in total volume 20 µl and incubated at 55 °C for 7 min. Then the reaction was stopped by adding 5 µl of 0.2% SDS and kept at 25 °C for 7 min. Adapter-ligated fragment amplification was done using Nextera XT index kit (Illumina) in a total volume 50 µl (1x Phusion HF Buffer, 2 µl of Phusion High Fidelity DNA Polymerase (Thermo Fischer, F530L), dNTP mix (0.3 mM each) (Promega)) with 10 cycles of PCR. The library was purified by 1X volumes of Ampure XP DNA purification beads and resuspended in 25 µl nuclease free H2O. Sequencing was performed on an Illumina HiSeq 2500 machine with single-end 50-bp read length (Illumina), per the manufacturer's recommendations.

RNA from Spermatogonia (10.000 per sample) was extracted with Single Cell RNA Purification Kit (Norgen, 51800) and libraries were prepared using the Ovation® SoLo RNA-Seq Library Preparation Kit (Nugen, 0501-32) following the manufacturer's instructions. Libraries were sequenced in 2 × 38 bp paired-end mode with NextSeq 500 sequencing technology (Illumina), per the manufacturer's recommendations.

## Whole mount seminiferous tubules immunohistochemistry

Seminiferous tubules were gently detangled and washed 2-3 times with PBS, to remove the interstitial cells. Then, the tubules were fixed with 4% PFA in PBS for 1 hour at 4 °C. After fixation, the tubules were washed 3 times in PBS and then 3 times in PBS containing 0.04% Tween-20 (PBST) in a cell strainer, each wash for 10 min. Then, the tubules were dehydrated through a graded series of 25%, 50%, 75% methanol containing PBST and 100% methanol at 4 °C for 7 min, respectively. The samples were stored at −80 °C until later usage. At the time of observation, the samples were rehydrated through a graded series of 75%, 50%, 25% methanol containing PBST for 7 min at 4 °C and then washed 3 times with PBST for 10 min each. Rehydrated tubules were blocked in PBST containing 4% Normal Donkey Serum (Abcam ab7475) for 1 hour. After blocking, the tubules were incubated with primary antibodies against cKit (1:1000 R&D systems AF1356), DNMT3A (1:1000 Imgenex IMG-268A) and DNMT3B (1:1000 Imgenex IMG-184A) for 3 hours at room temperature. Afterwards, the tubules were washed 3 times for 10 min in blocking buffer followed by incubation with species specific secondary Alexafluor-conjugated antibodies (1:1000, ThermoFischer Scientific) for 2 hours. Finally, tubules were incubated in 0.001 mg/ml DAPI solution (Sigma D9542) for 10 min and then washed 3 times with PBST for 10 min. For imaging, tubules were oriented in PBST between a microscope slide and a coverslip separated by a 0.12 mm thick SecureSeal™ Imaging Spacers (Grace Bio Labs). Images were obtained using spinning disk confocal scanning unit Yokogawa CSU W1 Dual T2 with 100x/1.4 oil immersion objective. Representative regions were selected using Fiji software[104].

## Histopathology

Fresh whole testis samples were fixed in 5 ml of Bouin's solution (Sigma HT10132) for at least 48–72 hours and then stored in 70% ethanol until further processing. The fixed samples were embedded in paraffin using an automated tissue processing center (TPC 15 Duo, Medite) with standard settings. Sectioning was done at 3 um thickness using the automatic microtome (HM355S, Thermo Fisher Scientific). Sections were mounted onto Superfrost Plus Adhesion Microscope Slides (J1800AMNZ, Thermo Fisher Scientific) and dried at 37 °C overnight. For staining, sections were deparaffinized by incubating in xylene solution (Sigma 534056) 2 times for 5 min and rehydrated in a series of decreasing concentrations of ethanol (2 × 100%, 95%, 70%, 3 min each) to deionized water. Rehydrated tissues were immersed in Periodic Acid

Solution (Sigma, 395132) for 5 min at RT, rinsed several times in deionized water, immersed in Schiff's Reagent (Sigma 3952016) for 15 min at RT and then rinsed in tap water for 5 min. Samples were counterstained with Mayer's Hematoxylin Solution (MHS32, Sigma) for 2 min and rinsed in tap water for 5 min. Finally, samples were dehydrated in a series of increasing concentrations of ethanol (70%, 95%, 2 × 100%, 3 min each), cleared in xylenes solution (2 x 3 min) and mounted with PermountTM mounting media (ThernmoFischer, SP15-100). Images were acquired using motorized automated slide Scanner Zeiss Axioscan Z1 with 40x air objective and analyzed with ZEN blue software (version 2.3, Zeiss). At least 4 testis sections (technical replicates) per mouse were scored manually for the presence of degenerative seminiferous tubules. Statistical comparisons were performed using a mixed-effects logistic regression model, accounting for both biological and technical replicates.

## Computational Analysis of Sequencing data

Computational analyses were performed using publicly available bioconductor software run in Rstudio (R version 4.2.1). Mouse genome BSgenome.Mmusculus.UCSC.mm10 was tiled in 500 bp non overlapping tiles using GenomicFeatures (version 1.56.0). Tiles that overlap with blacklisted regions from[105] were removed from the subsequent analysis. Tiles containing less than 4 CpGs were excluded from the analysis as non-informative. For annotation of genomic features, coordinates of genes (including exons and introns) were extracted from TxDb.Mmusculus.UCSC.mm10.knownGene (version 3.10.0) and the promoter regions were defined as genes TSS +/- 1kb (Supplementary Fig. 2B). CpG islands and repetitive element coordinates were downloaded from the UCSC "CpG Islands" and "RepeatMasker" tables respectively. Gametic DMR coordinates were obtained from[106].

EMseq (this study) and PBAT (from ref. [47]) reads were first processed using TrimGalore (version 0.6.2) to trim adaptor and low-quality reads with settings (−stringency 3). EMseq and PBAT trimmed reads were then aligned to the mouse genome build mm10 using the qAlign function from the QuasR package and Bismark package, respectively. Methylated and unmethylated read counts for each CpG were extracted from the BAM files using the function qMeth from the QuasR package. Tiles covered with less than 25 reads on their CpGs were discarded from downstream analysis. The summed read count of methylated and unmethylated CpGs was calculated for each retained tile. The resulting read count data were processed by edgeR to identify differentially methylated tiles between experimental groups as described previously[107]. Tiles with methylation percentage change >25% or <−25% and false discovery rate-adjusted FDR < 0.05 were considered to be significantly changed. The percentage of methylation was calculated as the fraction of methylated CpG read counts to the total CpG read counts for tile.

RRBS reads were first processed using TrimGalore (version 0.6.2) to trim adaptor and low-quality reads with settings (−rrbs, −stringency 3). Trimmed reads were then aligned to the mouse genome build mm10 using the qAlign function from the QuasR package. Methylated and unmethylated read counts for each CpG were extracted from the BAM files using the function qMeth from the QuasR package. CpGs covered with less than 5 reads were discarded from downstream analysis. In addition, tiles covered with less than 20 reads on their CpGs were discarded from downstream analysis. The summed read count of methylated and unmethylated CpGs was calculated for each retained tile. The resulting read count data were processed by edgeR to identify differentially methylated tiles among experimental groups. Tiles with methylation percentage change >20% or <−20% and false discovery rate-adjusted FDR < 0.05 were considered as significantly changed. The percentage of methylation was calculated as the fraction of methylated CpG read counts to the total CpG read counts for tile.

ChIP-seq and ATATA reads were first processed using TrimGalore (version 0.6.2) to trim adaptor and low-quality reads with

settings (−stringency 3). Trimmed reads were then aligned to the mouse genome build mm10 using STAR (version 2.5.0a) with settings (−alignIntronMin 1 −alignIntronMax 1 −alignEndsType EndToEnd −alignMatesGapMax 1000 −outFilterMatchNminOverLread 0.85). ATATA samples for hybrid JF1/MsJ x C57BL/6 JRj 2-cell embryos were separately aligned to C57BL/6J and JF1/MsJ genomes obtained by incorporating JF1 single-nucleotide polymorphisms (SNPs) into reference mm10 genome using previously published SNP table from[108]. Reads were categorized as maternal (JF1/MsJ), paternal (C57BL/6JRj) or undefined based on minimal number of mismatches in alignments to both genomes. Aligned reads were deduplicated using SAMtools (version 1.10) with standard settings. Uniquely mapped reads were counted on genomic regions using the function qCount (mapqMin = 255 L) from QuasR package (version 1.36.0). log$_2$RPKM values were calculated for each tile and normalized between biological samples of mutant and control using the function normalizeBetweenArrays (method = "cyclicloess",cyclic.method = "fast") from the package limma (version 3.52.2). For allelic analysis, total number of maternal and paternal reads was used as library size for calculating RPKM values. Normalized log$_2$RPKM values were transformed back to read counts and were processed by edgeR to identify differential enrichment of histone modification on tiles among experimental groups and calculate log$_2$ fold changes with standard parameters (prior.count = 1). Statistical significance cutoffs for a tile to be considered as differentially enriched were set as follows: log$_2$FC > 1 or < -1 and P-value < 0.05. ChIPseq peaks were identified using MACS3 with standard settings for broad peak search [macs3 callpeak −broad −nolambda -g mm −keep-dup all]. For downstream analysis we considered peaks with q-value < 0.01, and lengths lower than the 99th quantile. For heatmap plots the uniquely mapped reads were quantified 5 kb upstream and downstream of the middle of each genomic tile in 51 bp bins using the qProfile function from QuasR package. Log$_2$RPKM values of each bin were calculated and normalized as above. We performed smoothening by taking the mean value of 10 bins upstream and downstream of a given bin. The data were plotted using the package ComplexHeatmap (version 2.12.1) or ggplot2 (version 3.3.6).

Nugen Ovation Solo RNA-seq reads were first processed using TrimGalore (version 0.6.2) to trim adaptor and low-quality reads with settings (−clip_R1 3, −stringency 3). Trimmed reads were then aligned to the mouse genome build mm10 using STAR (version 2.5.0a) with settings (−outFilterMismatchNmax 6). Duplicated reads were marked/removed using nudup.py (version 2.2) with settings (-s 8, -l8). Uniquely mapped reads were counted on genes (TxDb.Mmusculus.UCSC.mm10.knownGene) using the function qCount (mapqMin = 255 L) from QuasR package (version 1.36.0). log$_2$RPKM values for each gene were calculated. The resulting read count data were processed by edgeR to identify differentially expressed genes among experimental groups. Genes with CPM < 1 were excluded from the analysis. Genes with log$_2$FC > 1 or < -1 and false discovery rate-adjusted FDR < 0.05 were considered to be significantly changed.

Differential expression analysis for early 4-cell mouse embryos RNA-Seq data was done using generalized linear model (GLM) with basis functions for natural splines included into the model to regress out expression differences that might be explained by possible developmental delay. First, we performed pseudotime ordering of knock-out and Ctrl embryos using in-house RNA-seq data for several time points of pre-implantation development as a benchmark. Pseudotime for each embryo was estimated using R package SCORPIUS (v1.0.8)[109]. Next, we constructed a model matrix for GLM considering genotypes as covariate SmartSeq2 RNA-seq reads were first processed using TrimGalore (version 0.6.2) to trim adaptor and low-quality reads with settings (−stringency 3). Trimmed reads were then aligned to the mouse genome build mm10 using STAR (version 2.5.0a) with settings (−outFilterMismatchNmax 6). SmartSeq2 RNA-seq samples for hybrid JF1/MsJ x C57BL/6 JRj 4-cell embryos were separately aligned to C57BL/6J and JF1/MsJ genomes obtained by incorporating JF1 single-nucleotide polymorphisms (SNPs) into reference mm10 genome using previously published SNP table from[108]. Reads were categorized as maternal (JF1/MsJ), paternal (C57BL/6JRj) or undefined based on minimal number of mismatches in alignments to both genomes. Aligned reads were deduplicated using SAMtools (version 1.10) with standard settings. Uniquely mapped reads were counted on genes (TxDb.Mmusculus.UCSC.mm10.ensGene) and tiles using the function qCount (mapqMin = 255 L) from QuasR package (version 1.36.0). log$_2$RPKM values for each gene or tile were calculated.

Differential expression analysis for early 4-cell mouse embryos RNA-Seq data was done using generalized linear model (GLM) with basis functions for natural splines with 3 components generated by ns function in R package splines (version 3.5.1) using pseudotime as knots to regress out effects of possible developmental delays. More explicitly, design matrix for GLM was generated using model.matrix function with formula ~ 0 + genotype + ns(PsT,3). To control possible overfitting by splines, the same model was fit for samples with randomly permuted pseudotime estimates. Expression changes and FDR were calculated for difference between paternal knock-out and control using log-likelihood test and Benjamini-Hochberg method for multiple testing correction. Differential expression analysis for allele specific expression was done similarly, taking read counts for respective allele, and normalizing by total number of allelic reads. In addition, uniquely mapped reads were quantified 5 kb upstream and downstream of the middle of each genomic tile in 51 bp bins using the qProfile function from QuasR package. Since genomic tiles do not have any particular orientation, we have artificially oriented the tiles to exhibit higher amount of RNA tags on their right side bins. We performed smoothening by taking the mean value of 10 bins upstream and downstream of a given bin.

Bigwig files for all sequencing experiments were generated using qExportWig function from QuasR package. For principal component analysis in Supplementary Fig. 5A the package PCAtools (version 2.8.0) was used. Clustering of tiles in Fig. 7A was performed with hclust and dist functions from the stats package (version 4.2.1) using the "Ward.D" method and "euclidian" distance. Gene ontology analysis was performed using the Mouse GO slim subset and the package topGO (version 2.48.0).

Analysis of enrichments of GC-poor/GC-rich hypoDMR/nonDMR tiles among tiles with significant changes in particular chromatin mark, e.g. gain or loss of nucleosome ChIP enrichment, was done by constructing 2 × 2 contingency tables with tile counts belonging/not belonging to particular class of tiles, e.g. GC-poor hypoDMR, versus having/not having significant change in particular chromatin mark and running Fisher's exact test using R function "fisher.test". log$_2$-enrichment values were calculated by dividing observed to expected tile counts calculated using R function "chisq.test" (after adding pseudocount 1) and taking log$_2$, i.e. $\log_2(\text{Observed} + 1)/(\text{Expected}+1)$.

We performed both known transcriptional motif and kmer ($k = 6$) enrichment analysis for the 500nt genomic regions that displayed methylation loss (>20%) specifically in Dnmt3a KO, Dnmt3b KO or DKO context in the different stages. In order to assess differential enrichment in these regions we defined a set of control regions encompassing the set 500nt genomic windows that display methylation loss in any assessed context and stage. Selection of these regions aimed to control for sequence biases associated with all regions that can undergo methylation loss in a manner that is not specific to the genetic context. Both motif and kmer enrichment analysis were carried out using the bioconductor monaLisa package using the calcBinnedMotifEnrR and calcBinnedKmerEnr functions respectively. For the known transcriptional motif analysis, we used the Jaspar 2020 database collection of vertebrate transcription factor position weight matrices. Adjusted enrichment p-values for both motif and kmer analyses were calculated using the Benjamini & Hochberg correction method.

**Reporting summary**

Further information on research design is available in the Nature Portfolio Reporting Summary linked to this article.

## Data availability

The raw sequencing data generated in this study have been deposited in the Gene Expression Omnibus database under accession code GSE229246. Publicly available datasets used in this study can be found at Gene Expression Omnibus database with the following accession IDs: GSE148150[47], GSE56697[25] and GSE73952[66]. The raw FACS generated in this study have been deposited in the FLOWRepository database under accession code FR-FCM-Z8EL. The raw image data files generated in this study have been deposited in the BioImage Archive database under accession code S-BIAD1457.The Source Data files are available at general repository Figshare (https://doi.org/10.6084/m9.figshare.27316803).

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

## Acknowledgements

We thank Y.K. Kawamura for expert advice on embryo culture experiments, M.B. Stadler for advice on computational data analysis and interpretation, H. Kohler for cell sorting, J. Keusch for support in Tn5-ZZ purification, N. Accart-Gris for histology support, L. Plantard for microscopy and imaging support, members of the animal facility for support, E. Li for *Dnmt3a* and *Dnmt3b* floxed mouse lines and J. van der Vlag for providing antibodies. We thank J. Brind'Amour and M. Lorincz for training on using the ULI-NChIP protocol. We acknowledge group members and particularly M. Gill for feedback on the manuscript. This research was supported by the Novartis Research Foundation, the Swiss National Science Foundation (31003A-172873) and the European Research Council (ERC) under the European Union's Horizon 2020 research and innovation programme (grant agreement ERC-AdG no. 695288 – Totipotency).

## Author contributions

G.F. and A.H.F.M.P. conceived the study, designed the experiments, interpreted the data, and wrote the manuscript with input from all authors. G.F. performed genomic experiments and analyzed the data with the help of E.A.O.. G.F. and S.A.S. developed the ATATA-seq methodology. L.G.T. and G.F. performed ChIP-seq experiment of sperm samples. P.A.K. and G.F. performed imaging experiments. P.P. performed motif search analysis.

## Competing interests

The authors declare no competing interests.
