## [Peer Review file · Nature Communications]

DNA methylation modulates nucleosome retention in sperm and H3K4 methylation deposition in early mouse embryos.

Corresponding Author: Professor Antoine Peters

Version 0:

Reviewer comments:

Reviewer #1

(Remarks to the Author)

During spermatogenesis, most histones are replaced by protamine and the genomic DNA becomes highly methylated at cytosine bases. However, sperm chromatin retains a small number of nucleosomes enriched for H3K4me3 at their DNA hypomethylated promoters, a phenomenon known as "histone retention" or "nucleosome retention" in spermatozoa. Although the exact function and mechanism of histone retention remain elusive, this phenomenon has received attention as a potential route for epigenetic transgenerational inheritance from fathers. This study demonstrated that conditional de novo DNA methyltransferase (DNMT3A and DNMT3B) deletion during early postnatal spermatogenesis resulted in slightly reduced DNA methylation levels and increased nucleosome occupancy, suggesting that paternal (sperm-derived) DNA methylation could influence nucleosome formation extent. Since artificial epigenomic modifier alteration could severely impair sperm formation or offspring development during spermatogenesis, studying how such changes affect the offsprings is challenging. For instance, conditional DNMT3A/3B deletion during fetal spermatogenesis [13] or spermatogonial overexpression of the histone demethylase KDM1A [33] result in severe consequences. However, this study provides a conceptual link between sperm-inherited DNA methylation and H3K4me3 deposition on the paternal genome in early embryos.

Please find below my questions and suggestions concerning the progress of this study.

1. Does DNMT3A exert a DNAm maintenance function?

The authors suggested that DNMT3A contributes to maintaining DNA methylation during spermatogenesis, as indicated by the methylome analysis: DNA methylation patterns did not dramatically change in undifferentiated spermatogonia (or sperms) lacking only DNMT3B compared to the control (Fig. 2 and Supplementary Figs. 4 and 5). However, "DNA methylation maintenance" is generally considered to involve hemimethylated cytosine (CpG) recognition and methylation, a function attributed solely to DNMT1 among DNA methyltransferases. The authors explained the potential molecular effector mechanism of the DNMT3A-mediated de novo methylation against TET-mediated demethylation or DNMT3A interaction with DNMT1 activity. While these possibilities are acceptable, I think that calling such a function "DNA methylation maintenance" is confusing. Therefore, I recommend changing this expression, e.g., to "safeguard against hypomethylation."

2. When did the Cre/loxP-mediated deletion begin?

The authors used the Stra8-Cre/loxP system to generate Dnmt3a- and Dnmt3b-deficient adult male germ cells. It is widely accepted that genome-wide DNA hypomethylation in sperm occurs predominantly before birth, specifically between embryonic day 16 (E16) and postnatal day 0.5 (P0.5) [7]. In contrast, Stra8 (Cre) is reportedly downregulated during the embryonic stages but emerges after birth [45]. However, could the conditional DNMT deletions be observed prenatally at all? Could this potentially explain the cases of mild oligospermia? A previous study demonstrated that embryonic DNMT3A/3B deletion leads to severe azoospermia [13]. If Stra8-Cre expression is initiated in a part of pre-spermatogonia, these cells might fail to properly develop into normal spermatogonia. To shed light on this phenomenon, the authors should compare the timing of de novo methylation completion and Stra8 expression onset during spermatogenesis, leveraging previously published data. If a potential overlap exists between de novo methylation and Stra8 initiation, their potential impact, including their association with oligozoospermia, should be discussed.

3. Is the epigenetic change maintained through subsequent generations?

Are sperm-derived epimutations reset during epigenomic reprogramming during periimplantation or germ cell development?

For example, can a hypo-DMR at the Bend3 locus be remethylated in the sperm of DKO offsprings? In a previous study, Belmonte et al. demonstrated that altered methylation through DNA methylation editing was transmitted across multiple generations, despite the temporal DNA methylation mark removal in blastocysts and PGCs (Takahashi Y., et al., 2023, Cell). This mechanism is unknown but interesting. Transgenerational tracing of artificial epigenetic changes would be a good model for studying paternal epimutation inheritance.

4. Did nucleosome occupancy increase at the genomic level?

The authors focused on HypoDMR (a hypomethylated region in the DKO sperm) and observed an H3K4me3- and nucleosome signal-related expansion. I think this is as good as it is. However, would it also be possible to quantify the number of peaks, their length distribution (e.g., depicted in a violin plot), and the aggregate coverage of all peaks (e.g., as a percentage of the genome) for each peak (ChIP-seq and ATATA-seq), then comparing them across samples using e.g., a general peak-calling program such as MACS2?

Minor points:

Page 8, line 269

Change: "Approximately 8% and 40% of hypoDMRs~" to: "Approximately 8% and 40% of hypoDMRs (N = 1,613 and 8,027, respectively) ~"

Page 9, line 302

Change: "These hypoDMRs had low to intermediate levels of DNAm in Ctrl samples (Figure 2A)." These hypoDMRs had low-to-intermediate DNAm levels in the Ctrl sperm (Figure 1A)."

Page 11, line 365

Change: "~30% of regions with increased nucleosome signal" to: "~30% (N = 154) of the regions with increased nucleosome signal".

Page 11, line 368

Change: "Inversely, 87% of H3K4me3 enriched regions" to: "Inversely, 87% (N = 1,077) of the H3K4me3-enriched regions."

Page 12, line 322

Do "methyltransferases" refer to DNA methyltransferases (DNMTs) or histone methyltransferases (HMTs)?

Figure 2

HypoDMR numbers in groups 1–6 should be indicated (as shown in Supplemental Figure 4).

Supplementary Figure 15D

HypoDMR numbers in groups 1–6 should be indicated (as shown in Supplemental Figure 15A or B).

Reviewer #2

(Remarks to the Author)

Remarks to the authors

Genome-wide de novo DNA methylation (DNAm) in the male germline occurs primarily in embryonic and prenatal prospermatogonia before birth. At these stages, de novo DNA methyltransferases DNMT3A, DNMT3B, and DNMT3C and their cofactor, DNMT3L play a pivotal role in establishing male-specific patterns of DNAm including at imprinted DMRs. However, some regions are devoid of de novo DNAm during the genome-wide wave of de novo DNAm in the embryonic stages. In this study, Fanourgakis and Peters et al. focused on de novo DNAm that occurs in postnatal germ cells and uncovered the novel role of this DNAm in spermatogenesis and pre-implantation development. Using Stra8-iCre, they generated conditional KO alleles for either DNMT3A or DNMT3B or both enzymes to deplete these proteins in the postnatal stages. Surprisingly, these mutant males are fertile in stark contrast with those that depleted these genes in the earlier stages of male germline development. Genome-wide DNAm analysis using EM-seq and RRBS methods in the mutant male germ cells identified regions depending on DNMT3A, DNMT3B, or both and differential dependency of these enzymes in the specific cell types in spermatogenesis. ChIP-seq analysis identified a subset of CpG-rich regions losing DNAm that show the retention of nucleosome and gain of H3K4me3 in the double KO (DKO) mutant spermatozoa. Pre-implantation embryos derived from the DKO spermatozoa gain more H3K4me3 at their paternal alleles and they claimed that the loss of DNAm in DKO sperm presumably caused this gain. Intriguingly, however, these mutant embryos can develop to term and the progenies show no obvious health issues. This is nice work with comprehensive datasets that advance the understanding of the role of de novo DNAm in postnatal male germ cells on the chromatin states in the early embryos. To strengthen their striking observations, I have some suggestions that could be considered in the revised version of the manuscript.

Major point

1. To address whether increased H3K4me3 or reduction of DNAm in DKO sperm promotes H3K4me3 deposition in 2-cell stage embryos, the authors focused on Group 1 of their category in Figure 7. A more solid conclusion could be extracted by using all regions (irrespective of the groups) that gain H3K4me3 in DKO spermatozoa. Although the authors concluded that the gain of H3K4me3 was not correlated with expression changes of the associated genes in the DKO-derived 4-cell stage embryos, integrating all affected regions would also capture more genes that may show misregulation in the embryos. Another concern for this figure is that it does not look clear to me how the authors distinguished H3K4me3 that are inherited from the DKO sperm genome and that are "prematurely" deposited in the paternal allele of DKO-derived early 2-cell stage embryos. Related to this concern, DNAm status at the paternal allele in the WT 1-cell and early 2-cell stage embryos could also be useful if sperm-derived DNAm at these regions blocks deposition of H3K4me3 in the WT. It would also be useful to check H3K4me3 at later stages of pre-implantation development in WT (4-cell for example) as H3K4me3 is truly gained "prematurely" in the DKO-derived 2-cell stage embryos. Another suggestion for this figure is that it would be difficult to follow the authors' claims by heatmap and boxplot formats. Scatterplot formats could also be used to see the impact of H3K4me3

gain in DKO spermatozoa on H3K4me3 at the paternal allele of the DKO-derived 2-cell embryos.

2. The authors claimed that DNMT3A serves a DNAm maintenance function in undifferentiated spermatogonia. As undifferentiated spermatogonia has both self-renewal and differentiating activities, it would be difficult to distinguish these activities in the mitotically active cell types. It also remains unclear to me which data supports the role of DNMT3A for the maintenance of DNAm rather than de novo DNAm. Related to the role of DNMT3A in the postnatal male germ cells, it is interesting that some regions are hypomethylated only in Dnmt3a KO. Could specific DNA motifs such as TET1/TET2 be enriched in these Dnmt3a-dependent regions? As the authors discussed this point in the manuscript, extracting DNA motifs in Dnmt3a-dependent regions would help understand the possible mechanism for DNAm at these regions.

3. While the authors claimed that H3K36me3 is unlikely to specify DNAm gain during spermatogonial differentiation, H3K36me3 ChIP-seq data from SgU DKO r1 looks quite different from other samples. It remains unclear whether the authors included this data in their analysis to conclude. If this data was included, substituting with another replicate data would help if their claim can be supported.

Minor point

1. Lines 141-144: While Dnmt3a cKO, Dnmt3b cKO, and Dnmt3a/3b cDKO males are fertile but have smaller testes compared to their WT counterparts. What would be the plausible reasons that these mutant males have smaller testes? Wouldn't it be also possible that early or efficient deletion of Dnmt3a/3b genes in a subset of germ cells by Stra8-iCre could lead to the early loss of these cells and the survivors are selected populations that show mild or modest impacts by the deletion of Dnmt3a/Dnmt3b.

2. Line 157: They claimed that DNAm at paternally methylated imprinting control regions (ICRs) was, however, unaffected. DNAm at the Rasgrf1 DMR looks affected as highlighted by the red rectangle. This sentence can be reworded.

3. Line 145-146: They mentioned that male mice derived from DKO males do not show any evident phenotypic abnormalities or health issues until adulthood. Are these DKO-derived males fertile? I'm curious about whether the aberrant H3K4me3 in DKO spermatozoa and DKO-derived pre-implantation embryos would impact the fertility of the next generation. To be clear, I'm not asking for additional experiments to answer this question.

Error

Line 129: which is active in postnatal undifferentiated and differentiating. spermatogonia may be missing from the phrase.

Reviewer #3

(Remarks to the Author)

In this manuscript, Fanourgakis et al. investigated the role of DNA methylation in shaping chromatin configuration in sperm and its impact on early embryonic development. The authors employed conditional deletion models targeting DNMT3A and DNMT3B to elucidate their roles in adult spermatogenesis and DNA methylation dynamics. They demonstrate that DNMT3A and DNMT3B enzymes are dispensable in postnatal germ cell development and production of competent gametes. However, they identified novel functions for these enzymes in de novo DNA methylation and H3K4me3 deposition. The authors delineate the temporal dynamics of DNA methylation establishment, suggesting that the majority of de novo DNA methylation occurs during spermatogonial differentiation before entry into meiosis. They further characterize the roles of DNMT3A and DNMT3B in regulating DNA methylation at specific genomic elements, with DNMT3A predominantly involved in establishing basal DNA methylation levels in SgU, while DNMT3B mainly catalyzing de novo DNA methylation in response to differentiation cues in SgD. Additionally, the study sheds light on the relationship between DNA methylation and chromatin organization during spermiogenesis. Their data support a model wherein unmethylated CpGs within GC-rich sequences facilitate nucleosome retention, thereby influencing chromatin architecture in sperm. Lastly, the authors demonstrate that low DNA methylation levels in sperm render paternal alleles permissive for H3K4me3 deposition that can be detected early embryos. However, there is no apparent transcriptional effect associated with the observed changes in methylation and H3K4me3.

Overall, the findings presented in this manuscript advance our understanding of DNA methylation dynamics and the intricate interplay between DNA methylation and chromatin organization in adult spermatogenesis. However, further clarifications are necessary for the completion of the manuscript.

Comments:

The authors mention that the reduction in DNA methylation in the here-generated DKOs is lower compared to previous studies. In the discussion they mention that this could be attributed to the timepoint of the conditional deletions used in the different studies. I think it is important to emphasize this more in the text and further discuss why Stra8 was used over a different Cre driver. Also, what are the durations between DKO generation and material collection in this study? Could the difference to previous studies also stem from a shorter interval between cKO and measurement?

Could the authors explain the reduced testicular weight in absence of observed transcriptional changes?

Lines 258-261: The authors suggest that DNMT3A and DNMT3B have partially different functions in different developmental contexts across adult spermatogenesis. Are there genomic loci specifically de novo methylated by DNMT3A and DNMT3B

within the same developmental context?

Line 352: The title of the section can be misleading. While some regions with increased nucleosome signal display significantly increased H3K4me3, most H3K4me3 enriched regions displayed no change in nucleosome occupancy in DKO sperm.

Figure 1e: This is a very complex figure with groups that are used further in the manuscript. I recommend simplifying the description in the text to make it clearer. I had to read it three times until I could follow what is happening in Fig1e, and I am still not sure I could completely understand what the authors did there.

Figure 5: The H3K4me3 heatmaps (5A) do not match the line plots (5B). According to the figure legends, the H3K4me3 signal is assessed by ChIP-seq in Figure 5A and by ULI-NchIP-seq in Figure 5B. According to the Methods section regarding ChIP, histone modifications in spermatogonial cells were profiled using ULI-NchIP-seq (L710-712), while in spermatozoa using native ChIP-seq (L737-L738). The authors should address these discrepancies.

The authors state that Groups 1,3,5 display increased H3K4me3 enrichment in DKO sperm. While Group 1 and 3 show decreased DNA methylation levels in DKO sperm compared to the Ctrl, Group 5 retains substantial levels of DNA methylation. Can the authors explain how the levels of DNA methylation of Group 5 makes these loci permissive to H3K4me3?

L398-402: How can the authors explain the differential enrichment between Ctrl and DKO sperm-derived embryos at maternal alleles?

Are the hypoDMRs that gain H3K4me3 in DKO sperm overlapping with those that gain H3K4me3 signal in DKO sperm-derived early embryos?

How do the authors explain changes in H3K36me3 without apparent changes in transcription (Figs S3H vs S7B), assuming K36me3 deposition is coupled to elongating Pol2?

Figure 6e-f. In the text the authors mention "increased H3K4me3 signals at hypoDMRs in GC-rich groups 1,3,5 and moderate increases at group 2,4,6 hypoDMRs, compared to those from Ctrl sperm-derived embryos". However, the changes observed are very low and I wonder how representative these measurements are.

Reviewer #4

(Remarks to the Author)

Reviewer #5

(Remarks to the Author)

In the mouse germline, there is extensive DNA methylation reprogramming during embryogenesis. The reestablishment of DNA methylation during spermatogenesis is absolutely crucial for i) keeping transposable elements at bay and ii) maintaining spermatogonial stem cell plasticity (PMID: 35410378). The DNA methylation gains between primordial germ cells and spermatogonia is dramatic. In this study, Fanourgakis and colleagues attempted to dissect the more minor DNA methylation dynamics between mitotic, undifferentiated spermatogonia and post-meiotic, mature sperm. To do so, they utilized a conditional knockout approach to remove either or both of the de novo DNA methyltransferases, DNMT3A and 3B, in undifferentiated spermatogonia.

There are a number of aspects of this study that should be commended. Firstly, the techniques undertaken in the study are challenging, and were carefully executed. This includes sorting pure cell types from heterogenous tissue, and performing low input chromatin analysis methods. Secondly, there were some novel findings that will be of note to the mammalian developmental epigenetics community. For example, despite the fact it is clearly expressed, a role for DNMT3B in spermatogenesis has been completely lacking, as far as I understand. Here, the authors demonstrate that DNMT3B is responsible for the de novo DNA methylation that occurs between spermatogonia and sperm; DNMT3A, has long been known to be required for fertility, plays more of a maintenance role during these stages. Secondly, the others showed that the DNA methylation program in this window helps shape the nucleosome/H3K4me3 patterns in mature sperm. In theory, this could have implications for intergenerational epigenetic inheritance.

On the less positive side, in my assessment our knowledge advancement is incremental. Firstly, there is a relatively tiny amount of genuine de novo methylation that occurs between undifferentiated and differentiated spermatogonia. Moreover, it is not clear what the biological importance of this DNA methylation is. This is reflected by virtually no transcriptional effects—let alone a developmental or fertility effects—in the conditional mutants. The authors used screen shots of only one single gene (Bend3, which coincidentally or not is a chromatin regulator), and the region of interest does not have an obvious regulatory role. I think the study is fully formed, and I would not recommend heavy, major experiments to somehow unravel a

more important role; I tend to think it is what it is. Below I have added some specific comments:

- Overall I found the data presentation, while thorough, difficult to digest and also repetitive. I wonder if, for example, the key results for the nucleosome and H3K4me3 retention could be presented on one figure for the affected groups without all the heat maps/line plots.
- Fig2A: I found Supp 5b a more useful representation of this data, and would recommend swapping these (ie, put Supp 5b in primary, and Fig 2A in supp). If you want to emphasize the specific de novo role from Dnmt3B, perhaps the PCA plot in supp 5A could also be put in the primary. This is clearer than the violin plots in any case, in my opinion.
- For the H3K36me3 analysis, I think it would be worthwhile to interrogate the role the histone mark plays for recruiting the individual Dnmt3 enzymes. Mouse data in other systems points to a general relationship between H3K36me3 and DNMT3B (eg, PMID: 25607372), with H3K36me2 more linked with DNMT3A (eg, PMID: 31485078). It should be possible to overlay H3K36me3 data with the RRBS for the single KOs, no? This would also help distinguish this study from previous work showing H3K36me2 more important from the transition from PGC to prospermatogonia (PMID: 32929285).
- I am not comfortable with calling ATATA-seq an entirely new method. To me it is a modified/optimized CUT&Tag, which has already been an established technique for several years. There are dozens (or more) variations of ChIP, but they are not given completely different names (to wit, ULI-NChIP used in this paper). I found the abstract a bit misleading when I actually read the description of ATATA-seq in the results section.
- Is there a potential role for DNMT3C? This was not mentioned.
- A model for male germline DNA methylation dynamics would be useful, even for specialists in DNA methylation that are not focused on spermatogenesis (eg, Fig 1a in PMID: 35410378).
- Fig 1A: what does the remaining signal in 3A and 3B KO represent? It appears cytoplasmic? It's background?
- Several IF images are missing in Supp 1A (this was probably some error during the upload process)
- Supp 1B is not properly referenced in text (this plot does not show the floxed exon specifically).
- Supp 1A: despite what's written in the text, there appears to be a fair amount of variation between replicates. How can this be explained.
- Supp 2D i: DNA methylation data missing
- Supp 3D mentioned out of sequence (line 189)
- Supp 5A: On PCA plot, why does Dnmt3b KO sperm look closer to SgU than SgD?
- Could other examples of screen shots be used besides Bend3? It seemed odd only to use example repeatedly, and the dynamically methylated region was not obviously important for gene regulation.

Version 1:

Reviewer comments:

Reviewer #1

(Remarks to the Author)

Dear Authors,

I have carefully reviewed your responses to my comments and the revised manuscript. I am pleased to see that you have thoroughly addressed the points raised, especially the clarifications regarding DNMT3A's role in safeguarding DNA methylation and the additional data provided to support the hypotheses on nucleosome occupancy and epigenetic changes.

I find the revisions to be satisfactory, and the additional analyses strengthen the conceptual framework and conclusions of the study. I have no further major comments or concerns.

Therefore, I recommend that this manuscript be accepted for publication in its current form.

Sincerely,
Reviewer #1

Reviewer #2

(Remarks to the Author)

The authors have thoroughly addressed my concerns, and I have no further comments on the revised version of the

manuscript.

Reviewer #3

(Remarks to the Author)

The authors have fully addressed all questions raised in the previous review. The manuscript greatly improved.

Reviewer #4

(Remarks to the Author)

Reviewer #5

(Remarks to the Author)

I appreciate the thorough response provided to the other reviewers and myself. As I stated in my original review, I think this is a scientifically sound study that will be well-received and cited by epigenetics researchers, especially those focused on DNA methylation and mammalian development. The authors appropriately responded to my comments, and I think the manuscript is improved as a result. I am particularly pleased with the new H3K36me3 analysis. I have no reservations about this paper's publication.

REVIEWER COMMENTS

Reviewer #1 (Remarks to the Author):

During spermatogenesis, most histones are replaced by protamine and the genomic DNA becomes highly methylated at cytosine bases. However, sperm chromatin retains a small number of nucleosomes enriched for H3K4me3 at their DNA hypomethylated promoters, a phenomenon known as "histone retention" or "nucleosome retention" in spermatozoa. Although the exact function and mechanism of histone retention remain elusive, this phenomenon has received attention as a potential route for epigenetic transgenerational inheritance from fathers. This study demonstrated that conditional de novo DNA methyltransferase (DNMT3A and asDNMT3B) deletion during early postnatal spermatogenesis resulted in slightly reduced DNA methylation levels and increased nucleosome occupancy, suggesting that paternal (sperm-derived) DNA methylation could influence nucleosome formation extent. Since artificial epigenomic modifier alteration could severely impair sperm formation or offspring development during spermatogenesis, studying how such changes affect the offsprings is challenging. For instance, conditional DNMT3A/3B deletion during fetal spermatogenesis [13] or spermatogonial overexpression of the histone demethylase KDM1A [33] result in severe consequences. However, this study provides a conceptual link between sperm-inherited DNA methylation and H3K4me3 deposition on the paternal genome in early embryos.

We thank the reviewer for acknowledging our efforts to circumvent the adverse impact of modifying extensively the epigenome of germ cells on sperm development and fertility and the merit of our minimally invasive mouse model to provide a conceptual link between sperm-inherited DNA methylation and H3K4me3 deposition on the paternal genome in early embryos. We find the suggestions of the reviewer constructive and have tried to address and implement them in our manuscript.

Please find below my questions and suggestions concerning the progress of this study.

Major points:

1. Does DNMT3A exert a DNAm maintenance function?

The authors suggested that DNMT3A contributes to maintaining DNA methylation during spermatogenesis, as indicated by the methylome analysis: DNA methylation patterns did not dramatically change in undifferentiated spermatogonia (or sperms) lacking only DNMT3B compared to the control (Fig. 2 and Supplementary Figs. 4 and 5). However, "DNA methylation maintenance" is generally considered to involve hemimethylated cytosine (CpG) recognition and methylation, a function attributed solely to DNMT1 among DNA methyltransferases. The authors explained the potential molecular effector mechanism of the DNMT3A-mediated de novo methylation against TET-mediated demethylation or DNMT3A interaction with DNMT1 activity. While these possibilities are acceptable, I think that calling such a function "DNA methylation maintenance" is confusing. Therefore, I recommend changing this expression, e.g., to "safeguard against hypomethylation."

Thank you for this thoughtful comment. Indeed, we do not wish to imply that DNMT3A exhibits typical DNA maintenance activity as attributed to DNMT1 for methylating hemi-methylated CpGs after DNA replication. We appreciate the reviewer's suggestion and agree that referring to DNMT3A's role as "DNA methylation maintenance" could be misleading. Therefore, we have revised the text as follows to clarify that DNMT3A's role is more about safeguarding against hypomethylation rather than traditional "catalytic maintenance" methylation.

In "Abstract" section page 1 line 23 to:

"We observed that DNMT3A primarily safeguards against DNA hypomethylation in undifferentiated spermatogonia while DNMT3B catalyzes *de novo* DNAm during spermatogonial differentiation."

In Introduction section page 4 line 111 to:

"By studying germ cells single and double conditionally deficient for Dnmt3a and/or Dnmt3b we identified developmental and genomic context specific contributions of each enzyme to safeguarding and *de novo* depositing DNAm during spermatogenesis."

In Results section page 6 line 205 to:

"*Dnmt3a* and *Dnmt3b* safeguard and *de novo* catalyze DNAm in spermatogonial stem cells and differentiated cells respectively."

In Results section page 8 line 274 to:

"Hence, these data show that DNMT3A serves a more prominent role than DNMT3B in depositing DNAm in SgU cells, thereby safeguarding the DNAm levels set prior to SgU formation."

In Results section page 8 line 279 to:

"In summary, during postnatal spermatogenesis we identified a major but not exclusive function of DNMT3A to safeguard basal DNAm levels at hypoDMRs in SgU, while DNMT3B mostly catalyzes *de novo* DNAm at hypoDMRs in response to differentiation cues in SgD."

In Discussion section page 16 line 563 to:

"We show that the *de novo* DNMT3B and DNMT3A enzymes fulfill distinct non-redundant functions in *de novo* DNAm and even in DNAm safeguarding during adult spermatogenesis, respectively."

In Discussion section page 16 line 567 to:

"Surprisingly in SgU, hypoDMRs with mid-to-high levels of DNAm were more sensitive to depletion of DNMT3A than DNMT3B (Supplementary Figure 5B), suggesting that DNMT3A is required to safeguard DNAm. DNMT3A may possibly counteract TET activity in SgU (87) and/or augment inefficient maintenance by DNMT1."

2. When did the Cre/loxP-mediated deletion begin?

The authors used the Stra8-Cre/loxP system to generate Dnmt3a- and Dnmt3b-deficient adult male germ cells. It is widely accepted that genome-wide DNA hypomethylation in sperm occurs predominantly before birth, specifically between embryonic day 16 (E16) and postnatal day 0.5 (P0.5) [7]. In contrast, Stra8 (Cre) is reportedly downregulated during the embryonic stages but emerges after birth [45]. However, could the conditional DNMT deletions be observed prenatally at all? Could this potentially explain the cases of mild oligospermia? A previous study demonstrated that embryonic DNMT3A/3B deletion leads to severe azoospermia [13]. If Stra8-Cre expression is initiated in a part of pre-spermatogonia, these cells might fail to properly develop into normal spermatogonia. To shed light on this phenomenon, the authors should compare the timing of *de novo* methylation completion and Stra8 expression onset during spermatogenesis, leveraging previously published data. If a potential overlap exists between *de novo* methylation and Stra8 initiation, their potential impact, including their association with

oligozoospermia, should be discussed.

Thank you for the comment. Hereby we compare public data on the timing of *Stra8-iCre* expression and *de novo* DNA methylation in male germ line. According to Sadate-Ngatchou *et al.* (2008), the *Stra8-iCre* transgene was constructed by fusing the -1400/+7 *Stra8* promoter fragment (+1 indicates the start of transcription of *Stra8*) to the improved *Cre* coding sequence. Interestingly this promoter fragment does not faithfully recapitulate the expression of endogenous *Stra8* as the authors observed that female embryos at E14.5 and E 16.5 did not show any *iCre* mRNA expression when endogenous *Stra8* is transcribed. Male embryos at E14.5 and E16.5 did not express the transgene at detectable level by RT-PCR and gel electrophoresis. When testes were examined for *iCre* transcripts at E18.5, P3, and P7 expression was first detected at P3 and became much stronger by P7 as assessed by RT-PCR and electrophoresis. However, Sadate-Ngatchou *et al.* didn't assess expression of transgene in P1 and P2 to investigate whether the transgene's expression could be detected between E18.5 and P3.

According to Seisenberger *et al.* (2012) (and also findings supported by Molaro *et al.* (2014)), male and female primordial germ cells (PGCs) reach the nadir of average genome wide mCpG at E13.5 (14% and 7% respectively), as assessed by BS-seq, with certain repetitive elements families resisting total demethylation. In female germline, low levels of methylation at E13.5 persist to E16.5, while male E16.5 prospermatogonia show evidence of robust *de novo* methylation with an increase to about 50% methylation (as assessed by BS-seq). In an orthogonal study by Kobayashi *et al.* (2013), the authors examined DNA methylation by PBAT and found global mCpG levels at 3.8% and 3.3% in male and female E13.5 PGCs respectively. Additionally, in male E16.5 prospermatogonia DNA methylation increased 31.8%, being, however, much smaller than in mature sperm cells (89.4%), indicating that further *de novo* DNA methylation occurs during ensuing development. Kubo *et al.* (2015) investigated the subsequent stages of DNA methylome by PBAT-seq and observed global mCpG methylation level increase from 30.1 % in male E16.5 prospermatogonia to 76.1 % in P0.5 prospermatogonia. Levels did not change much further in P7.5 Kit⁻ and Kit⁺ spermatogonia (76.6 % and 76.4 %, respectively). The final methylation level in adult spermatozoa was measured at 79.1 %. They identified 2413 differentially methylated regions DMRs which gain DNA methylation from P0.5 PSGs to spermatogonia. According to the authors, many of these DMRs were mapped to the partially methylated domains (PMDs) and they argue that because the genes in PMDs are mostly silent through the stages, the observed methylation changes did not seem to be relevant to gene regulation, but might rather reflect the ongoing *de novo* methylation of PMDs.

Based on public datasets, it seems that the completion of most genome-wide *de novo* DNA methylation is completed 1.5 days prior to the start of expression of the *Stra8-iCre* transgene. Nonetheless, given that the duration of this developmental time window is rather short, we cannot exclude a scenario in which *Stra8-iCre* expression would start precociously in a subset of prospermatogonia, leading to a failure to complete the wave of *de novo* DNA methylation, possibly causing a failure in the appearance spermatogonial stem cells normally maintaining spermatogenesis. Implication of such scenario in our model are discussed below:

According to Yoshida *et al.* (2006) the first wave of spermatogenesis in mouse initiates directly from prospermatogonia, without passing through the *Ngn3*-expressing stage (considered as the spermatogonia stem cell (SSC) compartment). The SSC compartment is established during

the early postnatal development of testes. According to Dura *et al.* (2022), *Dnmt3a* mutant germ cells (lacking *Dnmt3a* from E9.5 PGC and thus had severely abrogated subsequent *de novo* global DNA methylation) progress only through the first wave of spermatogenesis. The homeostasis, plasticity and differentiation potential of SSCs were reported to be compromised. In our models we considered that SSCs are normally formed and would exhibit normal global DNA methylation patterns (with only hypoDMRs being affected). However, we cannot exclude that a precocious expression of *Stra8-iCre* and the associated excision of *Dnmt3a/3b* floxed alleles may occur in a subset of prospermatogonia, affecting further the DNA methylation levels, in early postnatal testis and impede the proper establishment/plasticity of the progeny SSCs. Thus, a plausible hypothesis could be specifying fewer numbers of differentiation potent SSCs will still support the development of a functional testis tissue, exhibiting all germ cell developmental stages and being fertile yet with a reduced number of spermatogonial clone derived populations. To test this hypothesis, we quantified and statistically tested the presence of germ cell free seminiferous tubules in the testis of the mutant and control animals as shown in the Supplementary Figure 1F. We have additionally updated the previous Supplementary Figure 1E to show a wider representative view of testicular histology.

The above results are added to the Result section Page 4 Line 145:

“However, we noticed increased occurrence of seminiferous tubules lacking germ cells in single and double mutants compared to control mice (Supplementary Figure 1F), suggestive of a partial failure to establish a functional spermatogonial stem cell compartment. Accordingly, we observed ~15%-25% reduction in testicular weight upon removal of either DNMT3B or DNMT3A enzyme respectively which was further aggravated to ~30% upon depletion of both enzymes (Supplementary Figure 1G).”

We also discuss this more extensively in the Discussion section Page 16 Line 571:

“Based on previous studies (6-8,88), global *de novo* DNAm in male germ cells appears to be largely completed by postnatal day 0.5 (P0.5). Conditional depletion of DNMT3A using Cre drivers active during PGC development (e.g. *Prdm1-Cre* or *Tnap-Cre*) resulted in differentiation and genomic imprinting defects of spermatogonia after birth. In these studies, *Dnmt3a* deficiency resulted in a greatly hypomethylated genome of spermatogonia (10,13). Contrasting PGC-expressed Cre-drivers, expression of the *Stra8-iCre* transgene was detected by postnatal day 3 (P3) but not at E18.5 (45). Hence, we used *Stra8-iCre* to conditionally delete *Dnmt3a* and *Dnmt3b* during postnatal spermatogenesis, a developmental period after which global DNAm patterns had already been established. Given that global DNAm levels were not affected in our model, we interpret that DNAm was globally normally established in prospermatogonia and subsequently properly maintained by DNMT1 in spermatogonial stem cells (43). Nonetheless, we cannot entirely exclude the possibility of precocious expression of the CRE recombinase in some perinatal prospermatogonia, impairing the completion of most *de novo* DNA methylation. In turn, conversion of some prospermatogonia into spermatogonial stem cells may be impaired in our model (10), potentially explaining the increased presence of seminiferous tubules devoid of germ cells and reduced testis weights that we had observed for several *3aKO*, *3bKO* and many *DKO* males (Supplementary Figures 1E, 1F, 1G).”

3. Is the epigenetic change maintained through subsequent generations?

Are sperm-derived epimutations reset during epigenomic reprogramming during peri implantation or germ cell development? For example, can a hypo-DMR at the *Bend3* locus be remethylated in the sperm of *DKO* offsprings? In a previous study, Belmonte *et al.* demonstrated that altered methylation through DNA methylation editing was transmitted across multiple

generations, despite the temporal DNA methylation mark removal in blastocysts and PGCs (Takahashi Y., et al., 2023, Cell). This mechanism is unknown but interesting. Transgenerational tracing of artificial epigenetic changes would be a good model for studying paternal epimutation inheritance.

We appreciate the reviewer's interest in the potential implication for transgenerational effects. Takahashi *et al.* (2023) studied 2 promoter proximal CGIs which are normally hypomethylated in somatic and male germ cells. In their model, they specifically hypermethylated the *Ldlr* CGI up to ~50% and the *Ankrd26* CGI to ~100% in ES cells, which were later used to generate animals. They found that DNAm at the *Ankrd26* CGI was only partially reprogrammed in PGCs while the *Ldlr* CGI completely lost the DNA methylation. Subsequently, the sperm methylome analysis revealed that *Ldlr* CGI exhibited low levels, while in the majority of sperm *Ankrd26* CGI retained high DNA methylation levels with a minority of sperm showing DNA methylation reprogramming at *Ankrd26* CGI. They observed that pre-implantation embryos exhibited low DNA methylation at *Ldlr* CGI, while the *Ankrd26* CGI resisted also embryonic DNA methylation reprogramming. Interestingly the *Ldlr* CGI regained DNA methylation in somatic cells. In addition, the increased DNA methylation at *Ankrd26* and *Ldlr* promoters in somatic cells led to lower transcriptional output of these genes and altered metabolic traits.

Contrary to their model we induced nonspecific hypomethylation in sperm at regions which normally exhibit DNA methylation reprogramming in embryos (Supplementary Figure 15 A). In a way, we artificially reset the DNA methylation of these regions in the *DKO* sperm before the phase of embryonic reprogramming. To address the reviewers' question regarding a possible impact of *Dnmt3a/Dnmt3b* germ line deficiency on DNAm intergenerational inheritance, we profiled DNAm levels in embryonic offspring with a *Dnmt3a*^{+/-}; *Dnmt3b*^{+/-} genotype sired either by *Ctrl* and *DKO* males. As shown in **Revision Figure 1**, the hypoDMRs regions exhibited equally high DNA methylation levels in E9.5 embryos generated either by *Ctrl* or *DKO* males. This indicates that regardless of the DNA methylation status of sperm, remethylation of hypoDMRs in somatic cells takes place normally.

As these hypoDMR regions seem to play only marginal roles in the gene regulation (at least in spermatogonia and early embryos) and the absence of phenotypic abnormalities or gross developmental/health issues in *DKO*-derived F1 males until adulthood we were discouraged from further investigating the methylation status of the *DKO*-derived F1 males' sperm, their fertility and potential transgenerational effects. Such a study would involve crossing *DKO* and control animals, waiting for the F1 generation to reach reproductive maturity, and then performing fertility tests and DNA methylation analysis. It would last at least 6 to 8 months, and we consider it to be beyond the scope of this manuscript. Here, we focused on the intergenerational molecular dissection of sperm borne DNA methylation differences on chromatin regulation.

E9.5 embryos
Dnmt3a^{+/-}; *Dnmt3b*^{+/-}

Revision Figure 1. Heatmap showing the DNAm percentage of hypoDMRs assessed by RRBS in E9.5 embryos derived by crossing wild type females with either *Ctrl* males (*Dnmt3a*^{f/+}; *Dnmt3b*^{f/+}; *Stra8-iCre*) or *DKO* males (*Dnmt3a*^{f/-}; *Dnmt3b*^{f/-}; *Stra8-iCre*). The data represent the aggregate of 5 E9.5 embryos derived from each cross with genotypes *Dnmt3a*^{f/+}; *Dnmt3b*^{f/+}; *Stra8-iCre*. The density of CpGs at hypoDMRs (± 5 kb from their center) is indicated. Grouping of hypoDMRs was defined in Figure 1E. Within each group, hypoDMRs are ordered according to decreasing GC percentages. Violin plots showing the distribution of the DNA methylation values in each group for the *Ctrl* and the *DKO* sperm derived E9.5 embryos.

4. Did nucleosome occupancy increase at the genomic level?

The authors focused on HypoDMR (a hypomethylated region in the DKO sperm) and observed an H3K4me3- and nucleosome signal-related expansion. I think this is as good as it is.

However, would it also be possible to quantify the number of peaks, their length distribution (e.g., depicted in a violin plot), and the aggregate coverage of all peaks (e.g., as a percentage of the genome) for each peak (ChIP-seq and ATATA-seq), then comparing them across samples using e.g., a general peak-calling program such as MACS2?

Thank you for your comment and suggestions. We appreciate your interest in further quantifying nucleosome and H3K4me3 occupancies at the global genomic level. To address your questions, we have performed the following analyses:

Peak Calling:

We used MACS3, a widely used peak-calling program, with standard settings for broad peak search [macs3 callpeak --broad --nolambda -g mm --keep-dup all] to identify peaks from our ChIP-seq data. For downstream analysis we considered peaks with qvalue < 0.01, and lengths lower than the 99th quantile.

Peak Quantification:

Number of Peaks: We quantified the number of peaks identified in each sample to assess global changes in nucleosome and H3K4me3 occupancy.

Aggregate Coverage: We calculated the aggregate coverage of all identified peaks as a percentage of the genome. This will provide an overview of how much of the genome is covered by H3K4me3-enriched or nucleosome-enriched regions in each sample.

Length Distribution: We analyzed the length distribution of the identified peaks and depicted this information using violin plots to visualize the variability and distribution of peak lengths across samples.

We incorporated these results in the Supplementary Figures 11B-11D (for Nucleosome peaks in Sperm) and Supplementary Figures 12B-12D (for H3K4me3 peaks in Sperm)

We have revised the Results section to describe the new findings page 10 line 357:

We detected 45'125 and 50'354 anti-Nucleosome peaks, covering 1.2% and 0.8% of the genome, with a median length of 508 bp and 295 bp in *Ctrl* and DKO sperm, respectively (Supplementary Figure 11B, 11C, 11D).

and page 11 line 384:

We detected 26'654 and 29'166 anti-H3K4me3 peaks, covering 1.5% and 1.7% of the genome, with a median length of 950 bp and 953 bp in *Ctrl* and DKO sperm respectively (Supplementary Figure 11B, 11C, 11D).

Minor points:

1. Page 8, line 269 – now page 9 line 290

Change: “Approximately 8% and 40% of hypoDMRs~” to: “Approximately 8% and 40% of hypoDMRs (N = 1,613 and 8,027, respectively) ~”

Thank you for pointing out to refer to the absolute number of hypoDMRs in the text as well.

2. Page 9, line 302 – now page 10 line 326

Change: “These hypoDMRs had low to intermediate levels of DNAm in Ctrl samples (Figure 2A).” These hypoDMRs had low-to-intermediate DNAm levels in the Ctrl sperm (Figure 1A).”

We implemented the following change of text: “These hypoDMRs had low-to-intermediate DNAm levels in the *Ctrl* SgU samples (Figure 2A)”

3. Page 11, line 365 – now line 395

Change: “~30% of regions with increased nucleosome signal” to: “~30% (N = 154) of the regions with increased nucleosome signal”.

We implemented the suggested text edit: “~30% (N = 154) of the regions with significantly increased nucleosome signal”.

4. Page 11, line 368 – now line 397

Change: “Inversely, 87% of H3K4me3 enriched regions” to: “Inversely, 87% (N = 1,077) of the H3K4me3-enriched regions.”

We implement the change of text: “Inversely, 87% (N = 1,077) of H3K4me3 enriched regions displayed no significant change in nucleosome occupancy”.

5. Page 12, line 322 – now page 12 line 398

Do “methyltransferases” refer to DNA methyltransferases (DNMTs) or histone methyltransferases (HMTs)?

At this text segment the “methyltransferases” and the “demethylases” refer to HMT enzymes that have catalytic activity to H3K4. To increase clarity, we changed the text to:

“In summary, the data indicate that H3K4me3 methyltransferase enzymes (possibly in conjunction with H3K4 demethylase enzymes) versus DNMT3 enzymes mutually antagonize each other in differentiating spermatogonia e.g., by inhibiting DNMT3 catalytic activity versus preventing recruitment of KMT2A, KMT2B and SETD1A/CXXC1 H3K4 methyltransferases to methylated CpG-rich sequences.”

6. Figure 2

HypoDMR numbers in groups 1–6 should be indicated (as shown in Supplemental Figure 4).

We added the number of HypoDMRs in each group.

7. Supplementary Figure 15D

HypoDMR numbers in groups 1–6 should be indicated (as shown in Supplemental Figure 15A or B).

We have updated the Supplementary Figure 15D and added the number of HypoDMRs in each group.

Reviewer #2 (Remarks to the Author):

Remarks to the authors

Genome-wide de novo DNA methylation (DNAm) in the male germline occurs primarily in embryonic and prenatal prospermatogonia before birth. At these stages, de novo DNA methyltransferases DNMT3A, DNMT3B, and DNMT3C and their cofactor, DNMT3L play a pivotal role in establishing male-specific patterns of DNAm including at imprinted DMRs. However, some regions are devoid of de novo DNAm during the genome-wide wave of de novo DNAm in the embryonic stages. In this study, Fanourgakis and Peters et al. focused on de novo DNAm that occurs in postnatal germ cells and uncovered the novel role of this DNAm in spermatogenesis and pre-implantation development. Using *Stra8-iCre*, they generated conditional KO alleles for either DNMT3A or DNMT3B or both enzymes to deplete these proteins in the postnatal stages. Surprisingly, these mutant males are fertile in stark contrast with those that depleted these genes in the earlier stages of male germline development. Genome-wide DNAm analysis using EM-seq and RRBS methods in the mutant male germ cells identified regions depending on DNMT3A, DNMT3B, or both and differential dependency of these enzymes in the specific cell types in spermatogenesis. ChIP-seq analysis identified a subset of CpG-rich regions losing DNAm that show the retention of nucleosome and gain of H3K4me3 in the double KO (DKO) mutant spermatozoa. Pre-implantation embryos derived from the DKO spermatozoa gain more H3K4me3 at their paternal alleles and they claimed that the loss of DNAm in DKO sperm presumably caused this gain. Intriguingly, however, these mutant embryos can develop to term and the progenies show no obvious health issues. This is nice work with comprehensive datasets that advance the understanding of the role of de novo DNAm in postnatal male germ cells on the chromatin states in the early embryos. To strengthen their striking observations, I have some suggestions that could be considered in the revised version of the manuscript.

Thank you for your thoughtful and complete summary of our work. We appreciate your recognition of the comprehensiveness of our datasets and the significance of our findings in understanding the role of DNAm in male germ cell chromatin organization and its impact on chromatin states in early embryos. We are grateful for your suggestions to strengthen our observations and have carefully considered them during the revision process of our manuscript.

Major points:

1. **A)** To address whether increased H3K4me3 or reduction of DNAm in DKO sperm promotes H3K4me3 deposition in 2-cell stage embryos, the authors focused on Group 1 of their category in Figure 7. A more solid conclusion could be extracted by using all regions (irrespective of the groups) that gain H3K4me3 in DKO spermatozoa. **B)** Although the authors concluded that the gain of H3K4me3 was not correlated with expression changes of the associated genes in the DKO-derived 4-cell stage embryos, integrating all affected regions would also capture more genes that may show misregulation in the embryos. **C)** Another concern for this figure is that it does not look clear to me how the authors distinguished H3K4me3 that are inherited from the DKO sperm genome and that are “prematurely” deposited in the paternal allele of DKO-derived

early 2-cell stage embryos. D) Related to this concern, DNAm status at the paternal allele in the WT 1-cell and early 2-cell stage embryos could also be useful if sperm-derived DNAm at these regions blocks deposition of H3K4me3 in the WT. E) It would also be useful to check H3K4me3 at later stages of pre-implantation development in WT (4-cell for example) as H3K4me3 is truly gained “prematurely” in the DKO-derived 2-cell stage embryos. F) Another suggestion for this figure is that it would be difficult to follow the authors' claims by heatmap and boxplot formats. Scatterplot formats could also be used to see the impact of H3K4me3 gain in DKO spermatozoa on H3K4me3 at the paternal allele of the DKO-derived 2-cell embryos.

Thank you for your suggestion. Below we address the comments of the reviewer in a point by point manner, yet in a revised order, to facilitate a better understanding of the matter:

C) “Another concern for this figure is that it does not look clear to me how the authors distinguished H3K4me3 that are inherited from the DKO sperm genome and that are “prematurely” deposited in the paternal allele of DKO-derived early 2-cell stage embryos.”

To address whether changes of H3K4me3 are inherited, or *de novo* deposited at a hypoDMR region, we haven't used any biochemical tracking of this modification, but we rather compared the differential enrichment (DE) of this modification in sperm and embryo samples. Of note, we cannot compare directly the enrichment values of the H3K4me3 in *Ctrl* sperm and *Ctrl* sperm derived embryos as they were generated using totally different protocols. However, using the mutant samples to calculate the differential enrichment (DE) of H3K4me3 in Sperm and embryos and comparing the extent of DE may provide some indirect evidence of H3K4me3 inheritance.

For example, we can examine 3 scenarios: i) when a hypoDMR exhibits gain of H3K4me3 in *DKO* sperm ($\log_2 \text{FC} > 0.5$) and gain of H3K4me3 in *DKO*-sperm derived embryos ($\log_2 \text{FC} > 0.5$), we cannot conclude whether the gain of this modification in the embryo is inherited or has been *de novo* deposited. ii) Contrary, when a hypoDMR exhibits gain of H3K4me3 in *DKO* sperm ($\log_2 \text{FC} > 0.5$), but we measure no change or loss of H3K4me3 in *DKO*-sperm derived embryos ($\log_2 \text{FC} < 0.5$), we argue that there is no inheritance of the gain of this modification at this given region. iii) In addition, when a hypoDMR does not show gain of H3K4me3 in *DKO* sperm ($\log_2 \text{FC} < 0.5$) yet displays gain of H3K4me3 in *DKO*-sperm derived embryos ($\log_2 \text{FC} > 0.5$), we argue that the gain of this modification is *de novo* deposited in embryos and is not inherited. This logic is summarized in the table below:

Sperm	Embryo	Argument
H3K4me3 Gain ($\log_2 \text{FC} > 0.5$)	H3K4me3 Gain ($\log_2 \text{FC} > 0.5$)	Either inheritance or de novo deposition
H3K4me3 Gain ($\log_2 \text{FC} > 0.5$)	H3K4me3 no change or loss ($\log_2 \text{FC} < 0.5$)	No inheritance
H3K4me3 no change or loss ($\log_2 \text{FC} < 0.5$)	H3K4me3 Gain ($\log_2 \text{FC} > 0.5$)	De novo deposition

A) “To address whether increased H3K4me3 or reduction of DNAm in DKO sperm promotes H3K4me3 deposition in 2-cell stage embryos, the authors focused on Group 1 of their category

in Figure 7. A more solid conclusion could be extracted by using all regions (irrespective of the groups) that gain H3K4me3 in DKO spermatozoa.”

F) “Another suggestion for this figure is that it would be difficult to follow the authors' claims by heatmap and boxplot formats. Scatterplot formats could also be used to see the impact of H3K4me3 gain in DKO spermatozoa on H3K4me3 at the paternal allele of the DKO-derived 2-cell embryos.”

Based on the logic above, we have now stratified all HypoDMRs (not only those of Group 1) based on the log₂ FC in *DKO* vs *Ctrl* sperm into gain (log₂FC >0.5), no change (-0.5 < log₂ FC < 0.5) and loss (log₂ FC < -0.5) as well as whether they are GC rich or GC poor (the cutoff was set at 50% GC content) and then looked at the distribution of the log₂ FC in the embryos. We present this data in Figure 7A and 7B as heatmap and boxplots. Following the reviewer's suggestion to capture the variability in the changes of H3K4me3 in sperm and embryos we present also the data in scatterplots in Figure 7C.

We have revised the result section page 13 line 455 (current numbering) as follows:

“We next asked whether differential enrichment of H3K4me3 at hypoDMRs in DKO sperm leads to changes in H3K4me3 deposition at paternal alleles in 2-cell stage embryos. To address this question, we grouped all hypoDMRs based on increased (log₂ FC > 0.5), unaltered (-0.5 < log₂ FC < 0.5) or reduced (log₂ FC < -0.5) H3K4me3 occupancies in *DKO* versus *Ctrl* sperm and their GC content (GC-poor < 50% GC content < GC-rich). Then we quantified corresponding differential H3K4me3 occupancies in embryos (Figure 7A, 7B). The group of GC-rich hypoDMRs, which gained H3K4me3 in *DKO* sperm, showed increased H3K4me3 in *DKO* sperm-derived embryos (specific to the paternal allele) (Figure 7A, 7B). Here, we cannot distinguish whether the changes in embryonic chromatin are inherited, or *de novo* deposited. The groups of GC-rich hypoDMRs which remained unaltered or had lost H3K4me3 in *DKO* sperm showed, however, comparable increases in H3K4me3 in *DKO* sperm-derived embryos (specific to the paternal allele), strongly pointing towards elevated *de novo* H3K4me3 deposition in the embryo in absence of paternal DNAm (Figure 7A, 7B). All groups of GC-poor hypoDMRs, irrespective of the differential enrichment of H3K4me3 in sperm, displayed unaltered H3K4me3 levels in *DKO* sperm-derived embryos compared to *Ctrl* sperm-derived embryos (Figure 7A, 7B).”

D) “Related to this concern, DNAm status at the paternal allele in the WT 1-cell and early 2-cell stage embryos could also be useful if sperm-derived DNAm at these regions blocks deposition of H3K4me3 in the WT”

We reanalyzed the DNA methylation of hypoDMRs in publicly available data from Wang *et al.* (2014) in WT oocytes and embryos at 2 cell and 4 cell stages both in non-allelically and allelicly discrimination fashion in Supplementary Figure 15A-15C. Similar to WT sperm in WT oocytes the hypoDMRs show mid to high levels of DNA methylation. In addition, our analysis shows that most hypoDMR sequences identified in this study undergo rapid DNAm removal at the paternal allele by the 2 cell stage which is further reduced in 4 cell stage. Essentially, in the *DKO* sperm the DNA methylation of the hypoDMRs is artificially reset before the phase of embryonic reprogramming.

Interestingly, at the WT maternal allele, the DNA methylation reprogramming at the GC rich hypoDMRs (Groups 1, 3, 5) progresses slower compared to the WT paternal alleles.

E) “It would also be useful to check H3K4me3 at later stages of pre-implantation development in WT (4-cell for example) as H3K4me3 is truly gained “prematurely” in the DKO-derived 2-cell stage embryos.”

We reanalyzed the H3K4me3 methylation of hypoDMRs in publicly available data from Liu *et al.* (2016) in WT oocytes and embryos at 2 cell, 4 cell, 8 cell and blastocyst (ICM) stages. The data is shown in Supplementary Figure 15D-15E.

B) “Although the authors concluded that the gain of H3K4me3 was not correlated with expression changes of the associated genes in the DKO-derived 4-cell stage embryos, integrating all affected regions would also capture more genes that may show misregulation in the embryos.”

We appreciate the suggestion of the reviewer to integrate all regions to capture genes with expression changes, however, we have detected thousands of H3K4me3 DE regions in embryos [many of them possibly due to secondary biological effects or technical variability] while the extent of differential gene expression is limited to a few dozens of genes. Regardless of the gene expression we also compared the RNAseq reads in the vicinity of the hypoDMRs (Supplementary Figure 14E) and found a marginal increase in the *DKO* derived embryos in both maternal and paternal alleles, arguing against a direct specific role of hypoDMRs in transcriptome regulation. It is plausible that DNAm and/or H3K4me3 levels at the hypoDMRs regions do not impact on transcriptional output at this developmental stage.

2. The authors claimed that DNMT3A serves a DNAm maintenance function in undifferentiated spermatogonia. As undifferentiated spermatogonia has both self-renewal and differentiating activities, it would be difficult to distinguish these activities in the mitotically active cell types. It also remains unclear to me which data supports the role of DNMT3A for the maintenance of DNAm rather than de novo DNAm. Related to the role of DNMT3A in the postnatal male germ cells, it is interesting that some regions are hypomethylated only in *Dnmt3a* KO. Could specific DNA motifs such as TET1/TET2 be enriched in these *Dnmt3a*-dependent regions? As the authors discussed this point in the manuscript, extracting DNA motifs in *Dnmt3a*-dependent regions would help understand the possible mechanism for DNAm at these regions.

Thank you for your suggestion. In accordance also to Reviewer’s 1 Major Comment 1, we agree that referring to DNMT3A's role as "DNA methylation maintenance" could be misleading. We would like to point out to the reviewer that we do not wish to imply that DNMT3A exhibits typical DNA maintenance activity attributed to DNMT1 for methylating hemi-methylated CpGs after DNA replication. Therefore, we have revised the text accordingly to clarify that DNMT3A's role is more about safeguarding against hypomethylation rather than traditional “catalytic maintenance” methylation.

As shown in Supplementary Figure 4B and 4C, we plotted the differential DNA methylation levels for hypomethylated tiles from the single *Dnmt3a* mutant (*3aKO*) versus *Ctrl* against those in single *Dnmt3b* mutant (*3bKO*) versus *Ctrl* in SgU and SgD samples.

We defined *Dnmt3a*-specific tiles that lose DNA methylation for more than 20% with FDR <0.05 in single *3aKO* versus *Ctrl*, but not in single *3bKO* versus *Ctrl*.

Similarly, we defined *Dnmt3b*-specific tiles that lose DNA methylation for more than 20% with FDR <0.05 in single *3bKO* versus *Ctrl*, but not in single *3aKO* versus *Ctrl*.

Additionally, we defined, genetically, *DKO*-specific tiles that lose DNA methylation more than 20% with FDR <0.05 in *DKO* versus *Ctrl* animals, but neither in single *3aKO* versus *Ctrl* nor in single *3bKO* versus *Ctrl*. At these tiles the DNMT3A and DNMT3B activities are redundant. The Rest of the tiles include tiles that are not passing the minimum of 20% DNA methylation loss in any mutant or do not show any specificity to a given type of mutation.

The number of *Dnmt3a*-, *Dnmt3b*- or *DKO*- specific regions are shown in Supplementary Figure 4D.

Since we could not retrieve published DNA motifs for TET1/TET2 enzymes in JASPAR (the database of transcription factor binding motifs), we performed both known transcriptional motif and kmer (k=6) enrichment analysis for the 500nt genomic regions that displayed methylation loss (>20%) specifically in *Dnmt3a*, *Dnmt3b* or *DKO* context in the different stages. In order to assess differential enrichment in these regions we defined a set of control regions encompassing the set 500nt genomic windows that display methylation loss in any assessed context and stage. Selection of these regions aimed to control for sequence biases associated with all regions that can undergo methylation loss in a manner that is not specific to the genetic context. Both motif and kmer enrichment analysis were carried out using the bioconductor monaLisa package using the calcBinnedMotifEnrR and calcBinnedKmerEnr functions respectively. For the known transcriptional motif analysis, we used the Jaspar 2020 database collection of vertebrate transcription factor position weight matrices. Adjusted enrichment p-values for both motif and kmer analyses were calculated using the Benjamini & Hochberg correction method.

We included the result of this analysis in Supplementary Figure 4G and we have revised the Result section page 7 line 236:

“Additionally, *Dnmt3b*-specific tiles were more enriched in intragenic regions, particularly at exons (Supplementary Figure 4F), and did not display any sequence motif preference, in line with a transcription-coupled recruitment of the enzyme (data not shown) (51). For *Dnmt3a*-specific tiles, we detected slight overrepresentation of ELK3 and ELF2/4/5 transcription factor motifs, along with hexamers containing the “CCGG” tetramer, possibly reflecting the MspI biased restriction enzyme sites detected in the RRBS experiment, suggesting that also DNMT3A does not exhibit any genomic feature or sequence motif specificity.”

3. While the authors claimed that H3K36me3 is unlikely to specify DNAm gain during spermatogonial differentiation, H3K36me3 ChIP-seq data from SgU DKO r1 looks quite different from other samples. It remains unclear whether the authors included this data in their analysis to conclude. If this data was included, substituting with another replicate data would help if their claim can be supported.

Thank you for your comment. We wish to provide further information and technical quality control analysis for the ChIP experiment. For each ChIP experiment we have performed 2 biological replicates. We haven't excluded any datasets from any of our experiments. We used the replicates to perform statistical analysis and we merged the replicates for illustration in the

figures. Indeed, the replicate SgU *DKO* r1 shows consistently lower PCC to the other replicates, but still positive correlation coefficients. To reduce technical variability during the sequencing we pooled all the libraries for H3K36me3 together, aiming to a total number of 20 to 30 million sequenced reads per sample replicate. However, for unknown reason(s), we observed approximately a 5-fold lower mapping rate for the SgU *DKO* r1 reads to the mm10 genome compared to the other samples (**Revision Table 1**).

Sample	Mapped Reads	Unmapped Reads
K36m3_SgU_D3AB_WT_r1	23131575	3465331
K36m3_SgU_D3AB_WT_r2	21226069	2737622
K36m3_SgU_D3AB_DKO_r1	4245081	17979567
K36m3_SgU_D3AB_DKO_r2	26754097	3991575
K36m3_SgD_D3AB_WT_r1	21705026	3286766
K36m3_SgD_D3AB_WT_r2	22836561	2544556
K36m3_SgD_D3AB_DKO_r1	25187366	3016112
K36m3_SgD_D3AB_DKO_r2	17682996	3629661

Revision Table 1. Table showing the number of sequencing reads that were mapped or not to the mm10 reference genome from the a-H3K36me3 ChIP-seq replicates.

Principal component analysis (PCA) of the a-H3K36me3 ChIP-seq replicates showed that the a-H3K36me3 SgU *DKO* r1 is positioned further away in the second PC from the other a-H3K36me3 samples. However, we observed a striking separation in the first and second PCs of the a-H3K36me3 ChIP-seq samples from the a-H3K4me3 or the H3K27me3 ChIP-seq. This argues that all the H3K36me3 ChIP samples capture different genomic regions compared to the other histone marks, even the low mapped a-H3K36me3 SgU *DKO* r1 resides closer to the other H3K36me3 ChIPs rather than the other histone modifications (**Revision Figure 2**).

Revision Figure 2. Principal component analysis (PCA) of H3K4me3, H3K27me3 and H3K26me3 log₂RPKM values of 500 bp genomic tiles from Ctrl and *DKO* SgU and SgD cells.

As the majority of H3K36me3 is mediated in a co-transcriptional manner by Setd2 in prospermatogonia (Shirane *et al.* (2020) and Krogan *et al.* (2003)), we calculated the 100%, 75%, 50%, 25% and 0% read count quartiles based on their gene expression in *Ctrl* SgU and

SgD samples and stratified the genes in 4 categories 1) low expressed (0%-25% quartile), 2) mid-low expressed (25%-50% quartile), 3) mid-high expressed (50%-75% quartile) and 4) high expressed (75%-100% quartile). Then we plotted the log₂ RPKM of H3K36me₃ at intragenic 500bp tiles overlapping with each gene category (**Revision Figure 3**). We consistently observed in all replicates, including the SgU *DKO* r1, that the H3K36me₃ signal increases gradually as the gene expression increases. While the median H3K36me₃ enrichment of the intragenic tiles of highly expressed genes in most replicates is above 0.75 log₂RPKM, this enrichment in SgU *DKO* r1 is slightly below 0.75 log₂RPKM. This argues that H3K36me₃ ChIP in SgU *DKO* r1 captured the bona fide regions of H3K36me₃ albeit with lower enrichment values. Importantly, since we didn't observe global effects of *Dnmt3a/Dnmt3b* depletion on the levels of H3K36me₃ in SgD *DKO* replicates, we attribute this effect to the lower sequencing depth of this particular SgU *DKO* r1.

Revision Figure 3. Boxplots show the a-H3K36me₃ log₂ RPKM values on 500 bp tiles overlapping with gene bodies. Genes were binned into 4 categories based on their expression.

Despite the lower mapping rate of a-H3K36me₃ SgU *DKO* r1, we observed similar genomic distribution of a-H3K36me₃ among all replicates with higher enrichment at the gene bodies of the highly expressed genes compared to the intragenic regions as expected (**Revision Figure 4**).

Revision Figure 4. UCSC genome browser snapshot of chromosome 6qD1 partial segment, showing tracks for replicates of a-H3K36me3 ChIP seq from *Ctrl* and *DKO* SgU and tracks of merged replicates of RNA-seq from *Ctrl* and *DKO* SgU. The promoter CGIs and the hypoDMR are highlighted in green and red respectively. Genes showing high RNA expression are highlighted in green.

We kindly ask the reviewer to consider all these quality control analyses and the fact that SgU *DKO* r1 does not satisfy any exclusion criteria. Based on this, we do not think it is necessary to substitute this replicate in our analysis.

Minor points:

1. Lines 141-144: While *Dnmt3a* cKO, *Dnmt3b* cKO, and *Dnmt3a/3b* cDKO males are fertile but have smaller testes compared to their WT counterparts. What would be the plausible reasons that these mutant males have smaller testes? Wouldn't it be also possible that early or efficient deletion of *Dnmt3a/3b* genes in a subset of germ cells by *Stra8-iCre* could lead to the early loss of these cells and the survivors are selected populations that show mild or modest impacts by the deletion of *Dnmt3a/Dnmt3b*.

Thank you for your comment. A similar concern was raised by Reviewer 1. According to Dura *et al.* (2022), failure to establish genome wide DNA methylation in prospermatogonia leads to defects in homeostasis, plasticity and differentiation potential of SSCs. Using publicly available datasets, we have reviewed the timing of completion of global *de novo* methylation and the starting of *Stra8-iCre* expression and we concluded that they are temporally separated by 1.5 days. Given such a short developmental window we cannot exclude the possibility that precocious early activation of *Stra8-iCre* in a subset of prospermatogonia may have impaired their global levels of DNA methylation thereby impeding their conversion into spermatogonial stem cells and subsequent clonal expansion of male germ cells. Such a hypothesis predicts the presence of regions within seminiferous tubules lacking male germ cells, as observed in histological testis sections. We have examined this option by generating, analyzing and quantifying existing and newly generated histological data sets (Supplementary Figure 1E, 1F, 1G).

The obtained results are added to the Result section Page 4 Line 145:

“However, we noticed increased occurrence of seminiferous tubules lacking germ cells in single and double mutants compared to control mice (Supplementary Figure 1F), suggestive of a partial failure to establish a functional spermatogonial stem cell compartment. Accordingly, we observed ~15%-25% reduction in testicular weight upon removal of either DNMT3B or DNMT3A enzyme respectively which was further aggravated to ~30% upon depletion of both enzymes (Supplementary Figure 1G). “

We also discuss this more extensively in the Discussion section Page 16 Line 571:

“Based on previous studies (6-8,88), global *de novo* DNAm in male germ cells appears to be largely completed by postnatal day 0.5 (P0.5). Conditional depletion of DNMT3A using Cre drivers active during PGC development (e.g. *Prdm1-Cre* or *Tnap-Cre*) resulted in differentiation and genomic imprinting defects of spermatogonia after birth. In these studies, *Dnmt3a* deficiency resulted in a greatly hypomethylated genome of spermatogonia (10,13). Contrasting PGC-expressed Cre-drivers, expression of the *Stra8-iCre* transgene was detected by postnatal day 3 (P3) but not at E18.5 (45). Hence, we used *Stra8-iCre* to conditionally delete *Dnmt3a* and

Dnmt3b during postnatal spermatogenesis, a developmental period after which global DNAm patterns had already been established. Given that global DNAm levels were not affected in our model, we interpret that DNAm was globally normally established in prospermatogonia and subsequently properly maintained by DNMT1 in spermatogonial stem cells (43). Nonetheless, we cannot entirely exclude the possibility of precocious expression of the CRE recombinase in some perinatal prospermatogonia, impairing the completion of most *de novo* DNA methylation. In turn, conversion of some prospermatogonia into spermatogonial stem cells may be impaired in our model (10), potentially explaining the increased presence of seminiferous tubules devoid of germ cells and reduced testis weights that we had observed for several *3aKO*, *3bKO* and many *DKO* males (Supplementary Figures 1E, 1F, 1G)."

The remaining germ cells that we have analyzed in our study have efficiently excised the *Dnmt3a/Dnmt3b* alleles (Supplementary Figure 1B, 1C, 1D), in a temporal manner that took place after the completion of the global DNA methylation establishment as supported by the fact that the global DNA methylation levels show only minor difference from 81.7% in *Ctrl* sperm to 76.5% in *DKO* sperm (Figure 1B). We have discussed this scenario considering our new analysis in the revised version of our manuscript. For further details, we encourage the reviewer to see also our reply to Reviewer 1 Major Comment 2 above.

2. Line 157: They claimed that DNAm at paternally methylated imprinting control regions (ICRs) was, however, unaffected. DNAm at the *Rasgrf1* DMR looks affected as highlighted by the red rectangle. This sentence can be reworded.

Thank you for pointing out this omission which we have corrected in the main text as follows.

Page 5 line 163:

"DNA methylation at the paternally methylated imprinting control regions (ICRs) of *H19* and *Meg3* were unaffected, while the ICR for *Rasgrf1* exhibited a minor loss of DNA methylation (Supplementary Figure 2D)."

3. Line 145-146: They mentioned that male mice derived from *DKO* males do not show any evident phenotypic abnormalities or health issues until adulthood. Are these *DKO*-derived males fertile? I'm curious about whether the aberrant H3K4me3 in *DKO* spermatozoa and *DKO*-derived pre-implantation embryos would impact the fertility of the next generation. To be clear, I'm not asking for additional experiments to answer this question.

We appreciate the reviewer's interest in the fertility of male mice derived from *DKO* males and their potential impact on the next generation. We acknowledge that the absence of phenotypic abnormalities or gross developmental/health issues in *DKO*-derived F1 males until adulthood has discouraged us from further investigating potential transgenerational effects.

We apologize for not conducting breeding tests with the *DKO* F1 progeny to assess their fertility. We are grateful for the reviewer's understanding in not requiring conducting such a study which would involve crossing *DKO* and control animals, waiting for the F1 generation to reach reproductive maturity, and then performing fertility tests, which would take approximately 6 to 8 months.

4. Error Line 129: which is active in postnatal undifferentiated and differentiating. spermatogonia may be missing from the phrase.

We thank the reviewer for observing the omission which we have corrected in the manuscript.

Reviewer #3 (Remarks to the Author):

In this manuscript, Fanourgakis et al. investigated the role of DNA methylation in shaping chromatin configuration in sperm and its impact on early embryonic development. The authors employed conditional deletion models targeting DNMT3A and DNMT3B to elucidate their roles in adult spermatogenesis and DNA methylation dynamics. They demonstrate that DNMT3A and DNMT3B enzymes are dispensable in postnatal germ cell development and production of competent gametes. However, they identified novel functions for these enzymes in de novo DNA methylation and H3K4me3 deposition. The authors delineate the temporal dynamics of DNA methylation establishment, suggesting that the majority of de novo DNA methylation occurs during spermatogonial differentiation before entry into meiosis. They further characterize the roles of DNMT3A and DNMT3B in regulating DNA methylation at specific genomic elements, with DNMT3A predominantly involved in establishing basal DNA methylation levels in SgU, while DNMT3B mainly catalyzing de novo DNA methylation in response to differentiation cues in SgD. Additionally, the study sheds light on the relationship between DNA methylation and chromatin organization during spermiogenesis. Their data support a model wherein unmethylated CpGs within GC-rich sequences facilitate nucleosome retention, thereby influencing chromatin architecture in sperm. Lastly, the authors demonstrate that low DNA methylation levels in sperm render paternal alleles permissive for H3K4me3 deposition that can be detected early embryos. However, there is no apparent transcriptional effect associated with the observed changes in methylation and H3K4me3.

Overall, the findings presented in this manuscript advance our understanding of DNA methylation dynamics and the intricate interplay between DNA methylation and chromatin organization in adult spermatogenesis. However, further clarifications are necessary for the completion of the manuscript.

Thank you for your thorough and insightful review of our manuscript. We appreciate your positive assessment of our work and the valuable feedback provided. We are pleased to hear that you find our findings to advance the understanding of DNA methylation dynamics and their relationship with chromatin organization in spermatogenesis and early embryonic development. We have been committed to addressing your concerns to improve the clarity and completeness of our manuscript.

Comments:

1: The authors mention that the reduction in DNA methylation in the here-generated DKO is lower compared to previous studies. In the discussion they mention that this could be attributed to the timepoint of the conditional deletions used in the different studies. I think it is important to emphasize this more in the text and further discuss why Stra8 was used over a different Cre

driver. Also, what are the durations between DKO generation and material collection in this study? Could the difference to previous studies also stem from a shorter interval between cKO and measurement?

Thank you for your feedback to increase the clarity regarding the selection of the Cre driver and the sample collection timing. Following your comment and the Reviewer 1 Major Comment 2, we revised the Discussion section page 16 line 572:

“Conditional depletion of DNMT3A using Cre drivers active during PGC development (e.g. *Prdm1*-Cre or *Tnap*-Cre) resulted in differentiation and genomic imprinting defects of spermatogonia after birth. In these studies, *Dnmt3a* deficiency resulted in a greatly hypomethylated genome of spermatogonia (10,13). Contrasting PGC-expressed Cre-drivers, expression of the *Stra8*-iCre transgene was detected by postnatal day 3 (P3) but not at E18.5 (45). Hence, we used *Stra8*-iCre to conditionally delete *Dnmt3a* and *Dnmt3b* during postnatal spermatogenesis, a developmental period after which global DNAm patterns had already been established.”

In addition, regarding the material collection we revised the Material & Methods section page 20 line 688:

“Testis samples were collected from 3- to 12-month-old animals.”

Importantly, the difference in DNA methylation levels compared to the previous studies arise from the timing of Cre driver activity rather than the timing of collection. For example, Dura *et al.* (2022) shows the presence of spermatogonia stem cells in the testes of *Prdm1*-Cre driven conditional *Dnmt3a* mutant at 6 months of age. Although the authors haven't formally analyzed the DNA methylation levels of these germ cells at this age, based on other lines of evidence, they imply that their global DNA methylation level is low.

2: Could the authors explain the reduced testicular weight in absence of observed transcriptional changes?

Thank you for your question. A hypothesis is that precocious expression of *Stra8*-iCre in perinatal prospermatogonia and precocious deletion of *Dnmt3a/Dnmt3b* genes in a subset of these cells could lead to impaired global DNA methylation establishment, early loss of these cells and/or inability to define a plasticity/differentiation competent spermatogonia stem cell compartment. The prospermatogonia that have not precociously activated the expression of *Stra8*-iCre and have not yet deleted the *Dnmt3a/Dnmt3b* genes may have been selected to establish the SSC compartment, albeit with fewer initial cell founders compared to the *Ctrl* thus showing this mild or modest impacts on the testis weight. For further details on this hypothesis, how we addressed it analytically and in our revised text we kindly refer the reviewer to our reply to the Major comment 2 of Reviewer 1 above.

3: Lines 258-261: The authors suggest that DNMT3A and DNMT3B have partially different functions in different developmental contexts across adult spermatogenesis. Are there genomic loci specifically de novo methylated by DNMT3A and DNMT3B within the same developmental context?

Thank you for your question. As shown now in the revised Supplementary Figure 4B and 4C, we plotted the differential DNA methylation levels for hypomethylated tiles from the single *Dnmt3a* mutant (*3aKO*) versus *Ctrl* against those in single *Dnmt3b* mutant (*3bKO*) versus *Ctrl* in SgU and SgD samples.

We defined *Dnmt3a*-specific tiles that lose DNA methylation more than 20% with FDR <0.05 in single *Dnmt3a* mutant (*3aKO*) versus *Ctrl*, but not in single *Dnmt3b* mutant versus *Ctrl*.

Similarly, we defined *Dnmt3b*-specific tiles that lose DNA methylation more than 20% with FDR <0.05 in single *Dnmt3b* mutant (*3bKO*) versus *Ctrl*, but not in single *Dnmt3a* (*3aKO*) mutant versus *Ctrl*. Note that our classification for the *Dnmt3a*-specific or *Dnmt3b*-specific tiles allows them to be hypomethylated in the *DKO* as well.

Additionally, we defined *DKO*-specific tiles that lose DNA methylation more than 20% with FDR <0.05 in *DKO* versus *Ctrl*, but neither in single *Dnmt3a* mutant (*3aKO*) versus *Ctrl* nor in single *Dnmt3b* mutant (*3bKO*) versus *Ctrl*. At these tiles the DNMT3A and DNMT3B activities are redundant.

The Rest of the tiles include tiles that are not passing the minimum of 20% DNA methylation loss in any mutant or do not show any specificity to a given type of mutation.

The number of *Dnmt3a*-, *Dnmt3b*- or *DKO*-specific regions are shown in Supplementary Figure 4D.

Accordingly, we have revised the Result section page 7 line 232:

“We identified 4961 (in SgU cells) and 2485 (in SgD cells) *Dnmt3a*-specific tiles that significantly lost DNA methylation in *3aKO* versus *Ctrl*, but not in *3bKO* versus *Ctrl*. Similarly, we identified 16 (in SgU cells) and 830 (in SgD cells) *Dnmt3b*-specific tiles that significantly lost DNA methylation in *3bKO* versus *Ctrl*, but not in *3aKO* versus *Ctrl* (Supplementary Figure 4B, 4C, 4D, 4E).”

For additional details regarding the changes in DNMT3 enzyme specificity of the hypomethylated tiles along germ cell development, the genomic distribution of *Dnmt3b*-specific tiles, the chromatin mark enrichment on the *Dnmt3a*- and *Dnmt3b*- specific tiles along with sequence motif enrichment of the *Dnmt3a*- and *Dnmt3b*- specific tiles we refer this reviewer to our responses to Reviewer’s 2 Major Comment 2 and to Reviewer’s 5 Comment 3.

4: Line 352: The title of the section can be misleading. While some regions with increased nucleosome signal display significantly increased H3K4me3, most H3K4me3 enriched regions displayed no change in nucleosome occupancy in *DKO* sperm.

Apologies for the confusion, we have changed the title of this section at page 11 line 380 to:

“Increased H3K4me3 deposition at hypoDMRs in *DKO* sperm.”

5: Figure 1e: This is a very complex figure with groups that are used further in the manuscript. I recommend simplifying the description in the text to make it clearer. I had to read it three times until I could follow what is happening in Fig1e, and I am still not sure I could completely understand what the authors did there.

Thank you for your comment. To increase clarity and be more comprehensive, we restructured the text describing the aim of this Figure (beginning of the text), added separate paragraphs for

each group and incorporated additional annotations to Figure 1C (which is connected to Figure 1E):

The revised text at Page 5, line 180:

“To investigate the variable hypomethylation observed in *DKO* sperm, we compared the sperm methylomes to those of wild-type prospermatogonia (pSg) at postnatal day 1 (PBAT-seq data from (47)), representing the starting stage of postnatal germ cell development. We classified the hypoDMRs into 6 groups, based on a) their sequence composition (GC percentage > 50% for Groups 1,3,5 and < 50% for Groups 2,4,6) and b) their DNAm status in *Ctrl* and *DKO* samples (Figure 1E).

Group 1 & 2 hypoDMRs displayed low-to-mid levels (~10%-50%) of DNAm in pSg which increased to high levels (>50%) in *Ctrl* sperm. In *DKO* sperm, DNAm levels were reduced below levels present in pSg (<20%), as exemplified for the *Bend3* and *Srsf4* loci, suggesting that these are bona fide *de novo* methylated regions during adult spermatogenesis (Figures 1C, 1E, 1F, 1G).

HypoDMRs in groups 3 & 4 displayed only low DNAm (<20%) in pSg, mid-levels of DNAm (25%-50%) in *Ctrl* sperm and extremely low DNAm (<5%) levels in *DKO* sperm, suggesting that DNMT3A/DNMT3B enzymes are also required to sustain DNAm and/or counteract demethylation activities at these regions (Figures 1C, 1E, 1F).

Finally, groups 5 & 6 display mid-to-high DNAm in pSg (~30%-80%) and *Ctrl* sperm (75%-100%). However, despite a significant decrease in DNAm in *DKO* sperm, these regions maintained mid-levels of DNAm (25%-75%), suggesting that DNMT3A/DNMT3B confer *de novo* activity, while other DNA methyltransferases (such as *Dnmt1* which is highly expressed in spermatogonia (Supplementary Figure 3D)), may act on these regions as well (Figures 1E, 1F).”

6: Figure 5: The H3K4me3 heatmaps (5A) do not match the line plots (5B). According to the figure legends, the H3K4me3 signal is assessed by ChIP-seq in Figure 5A and by ULI-NchIP-seq in Figure 5B. According to the Methods section regarding ChIP, histone modifications in spermatogonial cells were profiled using ULI-NchIP-seq (L710-712), while in spermatozoa using native ChIP-seq (L737-L738). The authors should address these discrepancies.

Thank you for pointing out these discrepancies in the legend of Figure 5. The reviewer is correct in noting that we performed ULI-NChIP-seq for histone modifications in spermatogonial cells and native ChIP-seq for spermatozoa. We have corrected the figure legends to accurately reflect this distinction. Additionally, we have thoroughly reviewed all other figure legends to ensure consistency and avoid similar discrepancies. We appreciate the reviewer's attention to detail and apologize for any confusion this may have caused.

7: The authors state that Groups 1,3,5 display increased H3K4me3 enrichment in *DKO* sperm. While Group 1 and 3 show decreased DNA methylation levels in *DKO* sperm compared to the *Ctrl*, Group 5 retains substantial levels of DNA methylation. Can the authors explain how the levels of DNA methylation of Group 5 makes these loci permissive to H3K4me3?

In our analysis of DNA methylation, especially for the regions that display mid DNA methylation we are agnostic to 2 parameters A) the heterogeneity of DNA methylation in the given locus and B) the heterogeneity of the DNA methylation in a given locus among different alleles. For example, to mention extreme options, a 50% degree of DNA methylation at a locus could be the result of (A) half of the CpGs are methylated and half are not methylated in a given locus or (B) in half of the alleles all the CpGs are methylated and in the other half all the CpGs are not methylated as shown below:

Revision Figure 5. Schematic model depicting a DNA segment with unmethylated CpGs shown as white lollipops and methylated CpGs shown as black lollipops.

In scenario A, the regions in which methylated CpGs are located in one part of the region (bottom example) may be permissive for H3K4me3 in the other part. In scenario B, cells that exhibit fully methylated regions may not be permissive for H3K4me3, while cells in which the region is fully unmethylated they may be permissive for H3K4me3. We are aware of this limitation in the experimental design of this study. To distinguish between the scenarios mentioned above we will have to employ different experimental and analytical protocols.

We have acknowledged such limitation in our discussion section page 19 line 658:

“The ChIP-seq methods employed in this study examine relative nucleosome enrichments in large bulk populations of sperm limiting our understanding of the heterogeneity among spermatozoa that would contain nucleosomes at a given genomic region. Future studies examining sperm chromatin variability at a single cell resolution will be required to provide more quantitative insights to the extent by which the presence or absence of DNAm would modulate nucleosome eviction during spermiogenesis.”

8: L398-402: How can the authors explain the differential enrichment between Ctrl and DKO sperm-derived embryos at maternal alleles?

After assessing the differential enrichment of H3K4me3 by ATATA-seq in early 2C embryos we detected 5.31% of tiles without allelic assignment, 0.27% of tiles with maternal assignment and 4.02% of tiles with paternal assignment to be differentially enriched (Supplementary Figure 13B – upper panels). In addition to the minimal differential enrichment observed at maternal alleles compared to paternal alleles, the defined hypoDMRs are not significantly overrepresented among the differentially enriched tiles in the maternal alleles.

Possible explanations for this differential enrichment at maternal alleles are the fact that we used oocytes derived from various JF1 females. Although we did our best to match the ages of the oocyte donors, along with timing of hormone stimulation (we didn't match the estrous cycle before hormone administration) and oocyte extraction, we cannot exclude that such technical differences could impact the differential enrichment analysis.

9: Are the hypoDMRs that gain H3K4me3 in DKO sperm overlapping with those that gain H3K4me3 signal in DKO sperm-derived early embryos?

Thank you for your question regarding the overlap of hypoDMRs that gain H3K4me3 in *DKO* sperm and those that gain H3K4me3 signal in *DKO* sperm-derived early embryos. We have addressed this in Figure 7 of our manuscript. We classified all hypoDMRs based on the changes in H3K4me3 occupancy in *DKO* versus *Ctrl* sperm and their GC richness and then assessed the corresponding H3K4me3 levels in embryos. The results showed that, irrespective of the H3K4me3 occupancy changes in sperm, the *DKO* sperm-derived embryos (non-allelic and at paternal alleles) exhibited similarly increased H3K4me3 levels at GC rich hypoDMRs compared to controls. Then we examined the H3K4me3 changes at single GC rich hypoDMRs using scatterplots.

We have revised the Result section page 14 line 470:

“At single GC-rich hypoDMRs level analysis we observed that only a minor fraction of hypoDMRs show gain of H3K4me3 in both *DKO* sperm and *DKO* sperm-derived embryos (Figure 7C). This indicates that the gain of H3K4me3 in *DKO* sperm does not directly specify gain of H3K4me3 deposition in embryos. Instead, our data suggest that the presence of DNA methylation in sperm mostly restricts H3K4me3 deposition at GC-rich regions in the paternal chromatin of embryos, rather than H3K4me3 in sperm promoting its deposition in embryos.”

10: How do the authors explain changes in H3K36me3 without apparent changes in transcription (Figs S3H vs S7B), assuming K36me3 deposition is coupled to elongating Pol2?

Thank you for your question. First, we assessed the genomic location of the H3K36me3 differentially enriched tiles in SgU *DKO* versus SgU *Ctrl* or in SgD *DKO* versus SgD *Ctrl* as defined in Supplementary Figure 7B. We found that approximately 50% of H3K36me3 differentially enriched tiles in either SgU or SgD contrasts localize at intergenic regions. Despite that the RNA sequencing experiment didn't show a global statistically significant transcriptional mis regulation in SgU *DKO* versus SgU *Ctrl* or in SgD *DKO* versus SgD *Ctrl* (Supplementary Figure 3H), we defined downregulated, not altered and upregulated genes based on the log2 FC in *DKO* versus *Ctrl* contrasts. Downregulated genes exhibit $\log_2 \text{FC} < -1$, not altered genes exhibit $-1 < \log_2 \text{FC} < 1$ and upregulated genes exhibit $\log_2 \text{FC} > 1$. We identified tiles overlapping with each gene category and plotted their H3K36me3 log2 FC in the corresponding SgU *DKO* versus SgU *Ctrl* or in SgD *DKO* versus SgD *Ctrl* contrasts. We found that the median log2 FC around 0 suggesting that H3K36me3 does not change according to the differential gene expression (**Revision Figure 6**).

Revision Figure 6. Upper plots: Bar plots showing the percentage of H3K36me3 differentially enriched tiles in (left) SgU DKO vs Ctrl and (right) SgD DKO vs Ctrl overlapping with gene related features. As a reference, the percentage of non-differentially enriched tiles overlapping with gene related features is shown in the middle bars.

Lower plots: Genes were stratified by their log₂ fold change in (left) SgU DKO vs Ctrl and (right) SgD DKO vs Ctrl contrasts in downregulated (IFC < -1), not changed (-1 < IFC < 1) and upregulated (IFC > 1). Box plots showing the log₂ fold change of H3K36me3 tiles overlapping with the respective gene group.

11: Figure 6e-f. In the text the authors mention “increased H3K4me3 signals at hypoDMRs in GC-rich groups 1,3,5 and moderate increases at group 2,4,6 hypoDMRs, compared to those from Ctrl sperm-derived embryos”. However, the changes observed are very low and I wonder how representative these measurements are.

It is important to acknowledge the limitations of small input chromatin profiling techniques. Despite minimizing steps to ensure minimal sample loss, using a very limited amount of starting material (in our case, 50 2C embryos = 100 cells) results in a significant reduction in enriched

library complexity. Even when sequencing our ATATA-seq libraries to saturation (>50% duplication rate), the data still appear scarce, as observed in Figure 6H and Supplementary Figure 13.

Another limitation is the low signal-to-noise ratio observed with this and similar protocols like Cut&Tag when profiling fewer than 100 cells. Although we capture the majority of H3K4me3 peaks associated with promoters as expected (data not shown), the signal-to-noise ratio is inferior to those published with classic ChIP assays in the ENCODE project.

Notably, individual replicates of H3K4me3 ATATA-seq of 2C embryos show higher correlation to the replicates of published H3K4me3 ChIP of 2C embryos rather than H3K4me3 ChIP from blastocysts, suggesting that the biological context can be captured by this technique.

In addition, the differences in H3K4me3 detected by ATATA-seq between the groups in the Ctrl animals (Figure 6A, 6B) [Group 3 shows the highest signal, while Group 6 shows the lowest signal for H3K4me3] are also recapitulated by a-H3K4me3 ChIP-seq, as shown by Liu et al. in Supplementary Figure 15D. The observed trends, with Group 3 showing the highest signal and Group 6 the lowest for H3K4me3, are consistent between the ATATA and ChIP experiments. Extrapolating the consistency of these trends between ATATA-seq and ChIP-seq experiments suggests that the differences observed in H3K4me3 in Ctrl versus DKO contrasts at hypoDMRs likely reflect genuine biological effects.

Finally, the presence of informative SNPs between the BL6 and JF1 strains poses a limitation for assigning the allelic origin of the mapped reads, leading to further data scarcity. Although the differences in H3K4me3 ATATA-seq signal between control and *DKO* sperm-derived embryos are small (most likely due to the technical limitations mentioned above), we believe they capture the biological effect of the absence of sperm-borne DNA methylation due to the following reasons: a) the extent of differential enrichment of H3K4me3 was more prominent (20 times more regions being differentially enriched) at the paternal allele compared to maternal alleles, and b) HypoDMRs were enriched in the H3K4me3 differentially enriched tiles.

Reviewer #5 (Remarks to the Author):

In the mouse germline, there is extensive DNA methylation reprogramming during embryogenesis. The reestablishment of DNA methylation during spermatogenesis is absolutely crucial for i) keeping transposable elements at bay and ii) maintaining spermatogonial stem cell plasticity (PMID: 35410378). The DNA methylation gains between primordial germ cells and spermatogonia is dramatic. In this study, Fanourgakis and colleagues attempted to dissect the more minor DNA methylation dynamics between mitotic, undifferentiated spermatogonia and post-meiotic, mature sperm. To do so, they utilized a conditional knockout approach to remove either or both of the de novo DNA methyltransferases, DNMT3A and 3B, in undifferentiated spermatogonia.

There are a number of aspects of this study that should be commended. Firstly, the techniques

undertaken in the study are challenging, and were carefully executed. This includes sorting pure cell types from heterogenous tissue, and performing low input chromatin analysis methods. Secondly, there were some novel findings that will be of note to the mammalian developmental epigenetics community. For example, despite the fact it is clearly expressed, a role for DNMT3B in spermatogenesis has been completely lacking, as far as I understand. Here, the authors demonstrate that DNMT3B is responsible for the de novo DNA methylation that occurs between spermatogonia and sperm; DNMT3A, has long been known to be required for fertility, plays more of a maintenance role during these stages. Secondly, the others showed that the DNA methylation program in this window helps shape the nucleosome/H3K4me3 patterns in mature sperm. In theory, this could have implications for intergenerational epigenetic inheritance.

On the less positive side, in my assessment our knowledge advancement is incremental. Firstly, there is a relatively tiny amount of genuine de novo methylation that occurs between undifferentiated and differentiated spermatogonia. Moreover, it is not clear what the biological importance of this DNA methylation is. This is reflected by virtually no transcriptional effects—let alone a developmental or fertility effects—in the conditional mutants. The authors used screen shots of only one single gene (Bend3, which coincidentally or not is a chromatin regulator), and the region of interest does not have an obvious regulatory role. I think the study is fully formed, and I would not recommend heavy, major experiments to somehow unravel a more important role; I tend to think it is what it is.

Thank you for your comprehensive and balanced review of our manuscript. We appreciate your recognition of the challenging techniques we employed and the novel findings our study contributes to the field of mammalian developmental epigenetics. We understand your concern that the advancement in knowledge might seem incremental, especially given the lack of significant transcriptional, developmental, or fertility effects in the conditional mutants. We agree that these findings highlight the complexity of epigenetic regulation and its potential subtle regulatory roles that may not be immediately apparent through transcriptional changes alone in germ cell and embryo development. Thank you for providing your specific comments which helped us to enhance the overall quality and impact of our study.

Below I have added some specific comments:

1. Overall, I found the data presentation, while thorough, difficult to digest and also repetitive. I wonder if, for example, the key results for the nucleosome and H3K4me3 retention could be presented on one figure for the effected groups without all the heat maps/line plots.

We appreciate the reviewer's feedback regarding the presentation of our data. We acknowledge that our data presentation may appear repetitive at first sight. The repetitive nature of the figures may arise from the fact that we examined specific genomic regions, particularly the Hypo-DMRs, throughout this manuscript. However, we have structured the figures in this manner to provide a comprehensive and detailed view of DNA methylation and chromatin dynamic landscape across multiple developmental stages of spermatogenesis and early embryo development at these regions. All main figures (except Figure 7) present new experimental results. By presenting

these experiments separately, we aim to highlight the unique findings and ensure clarity in interpreting the results. While line plots provide an averaged view of nucleosome and H3K4me3 signals across regions in a specific group, we believe that including heatmaps is essential. We strive to present our data in a way that is both honest and informative. Heatmaps illustrate the biological variability inherent in our data, allowing readers to appreciate the differences among individual regions. This level of detail is crucial for understanding the complexity and variability of chromatin dynamics. While we attempted to consolidate key results for nucleosome and H3K4me3 retention into a single figure as per reviewer's suggestion, we find the resulting figure quite cluttered and convoluted. Hence by keeping the figures separate we also maintain the logical flow of the manuscript, ensuring that each experiment is discussed thoroughly and coherently.

2. Fig2A: I found Supp 5b a more useful representation of this data, and would recommend swapping these (ie, put Supp 5b\B in primary, and Fig 2A in supp). If you want to emphasize the specific de novo role form Dnmt3B, perhaps the PCA plot in supp 5A could aslo be put in the primary. This is clearer than the violin plots in any case, in my opinion.

We appreciate the reviewer suggestions to increase the clarity of our data presentation. Indeed, we moved the former panel of Supplementary Figure 5B to the Main Figure 2, as Main Figure 2B to 2D. The panels of Main Figure 2A to 2D are moved to Supplementary Figure 5B to 5E. Additionally, we moved the PCA plot from Supplementary Figure 5A to the Main Figures 2 as panel A. The associated Figure legends have been revised accordingly.

3. For the H3K36me3 analysis, I think it would be worthwhile to interrogate the role the histone mark plays for recruiting the individual Dnmt3 enzymes. Mouse data in other systems points to a general relationship between H3K36me3 and DNMT3B (eg, PMID: 25607372), with H3K36me2 more linked with DNMT3A (eg, PMID: 31485078). It should be possible to overlay H3K36me3 data with the RRBS for the single KOs, no? This would also help distinguish this study from previous work showing H3K36me2 more important from the transition from PGC to prospermatogonia (PMID: 32929285).

Thank you for your insightful suggestion regarding the role of H3K36me3 in recruiting DNMT3 enzymes. We agree that exploring the relationship between H3K36me3 and the individual DNMT3 enzymes could provide valuable insights and further distinguish our study. To address your suggestion, we performed the following analysis:

As shown in Supplementary Figure 4B and 4C, we plotted the differential DNA methylation levels for hypomethylated tiles from the single *Dnmt3a* mutant (*3aKO*) versus *Ctrl* against those in single *Dnmt3b* mutant (*3bKO*) versus *Ctrl* in SgU and SgD samples.

We defined *Dnmt3a*-specific tiles that lose DNA methylation more than 20% with FDR <0.05 in single *3aKO* versus *Ctrl*, but not in single *3bKO* versus *Ctrl*.

Similarly, we defined *Dnmt3b*-specific tiles that lose DNA methylation more than 20% with FDR <0.05 in single *3bKO* versus *Ctrl*, but not in single *3aKO* versus *Ctrl*. Note that our classification

for the *Dnmt3a*-specific or *Dnmt3b*-specific tiles allows them to be hypomethylated in the *DKO* as well.

Additionally, we defined *DKO*-specific tiles that lose DNA methylation more than 20% with FDR <0.05 in *DKO* versus *Ctrl*, but neither in single *3aKO* versus *Ctrl* nor in single *3bKO* versus *Ctrl*. At these tiles the DNMT3A and DNMT3B activities are redundant.

The Rest of the tiles include tiles that are not passing the minimum of 20% DNA methylation loss in any mutant or do not show any specificity to a given type of mutation.

The number of *Dnmt3a*-, *Dnmt3b*- or *DKO*-specific regions are shown in Supplementary Figure 4D.

Dnmt3b-specific tiles were underrepresented in SgU while *Dnmt3a*-specific tiles were the most abundant, consistent with a primary role of DNMT3A safeguarding the DNA methylation at this stage. Additionally, the *Dnmt3b* specific tiles were more enriched in intragenic regions, particularly in exons as shown in the Supplementary Figures 4E and 4F.

Subsequently for the DNMT3-enzyme specific tiles defined in SgU and SgD cells, we plotted the log2 RPKM values of H3K36me3 in SgU and SgD *Ctrl* samples in Supplementary Figures 7D and 7E. We noticed higher H3K36me3 signals at *Dnmt3b*-specific tiles compared to *Dnmt3a*-specific tiles arguing that DNMT3B may indeed be targeted by the PWWP domain interaction with H3K36me3 at gene bodies.

Accordingly, we have revised the Result section page 7 line 232 as follows:

“We identified 4961 (in SgU cells) and 2485 (in SgD cells) *Dnmt3a*-specific tiles that significantly lost DNA methylation in *3aKO* versus *Ctrl*, but not in *3bKO* versus *Ctrl*. Similarly, we identified 16 (in SgU cells) and 830 (in SgD cells) *Dnmt3b*-specific tiles that significantly lost DNA methylation in *3bKO* versus *Ctrl*, but not in *3aKO* versus *Ctrl* (Supplementary Figure 4B, 4C, 4D, 4E). Additionally, *Dnmt3b*-specific tiles were more enriched in intragenic regions, particularly at exons (Supplementary Figure 4F), and did not display any sequence motif preference, in line with a transcription-coupled recruitment of the enzyme (data not shown) (51).”

And page 9 line 302:

“Globally, we noticed an increased H3K36me3 signal at the *Dnmt3b*-specific tiles compared to the *Dnmt3a*-specific tiles arguing that primarily DNMT3B may be targeted to gene bodies via an interaction of its PWWP domain with H3K36me3 (Supplementary Figures 7D, 7E).”

And Discussion section page 17 line 596:

“Previously it was shown that DNMT3B preferentially located at gene bodies of transcribed genes in ESCs (51), possibly through interactions between its PWWP domain and the H3K36me3 modification, deposited by the RNA polymerase II interacting histone methyltransferase SETD2. Like in ESCs, it is plausible that in male germ cells, genic DNA methylation is also deposited by DNMT3B as we found that the *Dnmt3b*-specific tiles were more enriched in H3K36me3 signal compared to the *Dnmt3a*-specific tiles (Supplementary Figure 7D and 7E).”

4. I am not comfortable with calling ATATA-seq an entirely new method. To me it is a modified/optimized CUT&Tag, which has already been an established technique for several years. There are dozens (or more) variations of ChIP, but they are not given completely different names (to wit, ULI-NChIP used in this paper). I found the abstract a bit misleading when I actually read the description of ATATA-seq in the results section.

We appreciate the reviewer's perspective regarding the classification of ATATA-seq as an entirely new method. We understand and acknowledge in our manuscript that it shares similarities with established techniques like CUT&Tag. However, we believe that ATATA-seq incorporates significant modifications that warrant its distinction as a separate method.

Key differences include:

Chimeric Protein Used: We generated a novel Tn5 chimeric protein consisting of two Z domains of staphylococcal protein A fused to the Tn5 transposase (ZZ-Tn5). This differs from the conventional full protein A-Tn5 fusion used in CUT&Tag.

Buffer Conditions: ATATA-seq uses high salt buffer conditions during the targeting phase to restrict the inherent affinity of the transposase to accessible DNA, which is a significant modification aimed at increasing specificity to the antibody targeted regions.

Preloading Transposomes: Unlike CUT&Tag, where antibodies and transposomes are sequentially incubated with the sample, in ATATA-seq method we preload the ZZ-Tn5 transposomes with antibodies before the application to the samples. This preloading step simplifies the procedure and enhances the targeting efficiency.

Protocol Steps: The absence of washing steps in ATATA-seq reduces potential sample loss and contamination, making the technique more streamlined and sensitive.

Furthermore, similar to other techniques that have derived from CUT&Tag but have been given distinct names (e.g., ACT-seq, CoBATCH, etc.), we have chosen to name our method ATATA-seq to reflect its unique components and methodological improvements.

Examples of other techniques named distinctly despite extensive similarities to CUT&Tag (all using the full protein A-Tn5 fusion):

ACT-seq: Mapping histone modifications in low cell number and single cells using antibody-guided chromatin tagmentation (Carter *et al.* (2009), Nature Communications) .

CoBATCH: CoBATCH for High-Throughput Single-Cell Epigenomic Profiling (Wang *et al.* (2019), Molecular Cell).

Stacc-seq: The landscape of RNA Pol II binding reveals a stepwise transition during ZGA (Liu *et al.* (2020), Nature).

These examples show that even minor modifications can justify distinct naming to indicate specific advancements and applications of the technique.

To avoid any mislead we revised the “Abstract” section page 1 line 29 to:

“To assess the impact of altered sperm chromatin in the formation of embryonic chromatin, we measured H3K4me3 occupancy at paternal and maternal alleles in 2-cell embryos using a transposon-based tagging assay for modified chromatin.”

5. Is there a potential role for DNMT3C? This was not mentioned.

Thank you for raising the question about the potential role of DNMT3C. We focused our study on DNMT3A and DNMT3B, however, we appreciate the opportunity to discuss DNMT3C and provide evidence regarding its involvement.

While DNMT3C functions in male germ cells, several lines of evidence, according to Barau *et al.* 2016, suggest it does not play a major role in the processes we studied:

Temporal Expression Patterns: Changes in DNA methylation at hypoDMRs are observed postnatally during the differentiation of spermatogonia in the adult testis. However, the long DNMT3C isoform, which possesses a 709-codon ORF, shows a sharp peak in expression around embryonic day 16.5 (E16.5), coinciding with male germline *de novo* DNA methylation. Only the short, noncoding isoform of DNMT3C is expressed in postnatal testes. This timing does not align with the postnatal changes we observed. Additionally, our transcriptomic data indicates that *Dnmt3c* expression is very low in both undifferentiated spermatogonia (SgU) and differentiating spermatogonia (SgD).

Targeting Mechanism: We observed that hypoDMRs are enriched in the gene bodies of actively transcribed genes, suggesting that H3K36me3 methylation likely directs DNMT3A and DNMT3B to these regions. DNMT3C lacks the Pro-Trp-Trp-Pro (PWWP) domain, which is essential for targeting DNMT3 proteins to gene bodies through recognition of H3K36 trimethylation (H3K36me3). This absence indicates that DNMT3C might not be efficiently targeted to these regions.

Target Specificity: DNMT3C has been suggested to methylate young retrotransposon elements. However, we did not detect any overrepresentation of hypoDMRs at repetitive elements, further arguing against a major role for DNMT3C in the methylation dynamics we studied.

Given these points, we believe that DNMT3A and DNMT3B are the primary contributors to the DNA methylation changes observed in our study.

We included a brief discussion on the potential involvement of DNMT3C in our revised manuscript, acknowledging its functions and the reasons we consider its role to be less significant in the context of our finding as follows:

In Discussion section page 17 line 611:

“Given that *Dnmt3c* is rather low expressed in undifferentiated or differentiated spermatogonia (Supplementary Figure 3D) compared to prospermatogonia around E16.5 (12), and that DNMT3C lacks a PWWP domain and has been implicated in methylating young retrotransposon elements (12), its role in postnatal DNA methylation of gene body associated and H3K36me3 embedded hypoDMRs may be limited.”

6. A model for male germline DNA methylation dynamics would be useful, even for specialists in DNA methylation that are not focused on spermatogenesis (eg, Fig 1a in PMID: 35410378).

Thank you for the suggestion. We added a model in Figure 7D, summarizing the dynamic of global male germline DNA methylation across spermatogenesis and early embryonic development as well as the DNA methylation and chromatin dynamics at the Group 1 GC-rich HypoDMRs in *Ctrl* and *DKO* samples, thereby highlighting our findings. This panel is referenced extensively in our discussion to increase the comprehension of the DNA methylation dynamics during *Ctrl* and *DKO* male germ cell development by our readers.

7. Fig 1A: what does the remaining signal in 3A and 3B KO represent? It appears cytoplasmic? It's background?

Thank you for your observation regarding the remaining signal in the *Dnmt3a* and *Dnmt3b* single KO samples in Figure 1A. The signal observed in the cytoplasmic regions of the *Dnmt3a* and *Dnmt3b* single KO differentiated spermatogonia likely represents background staining. Please note that the remaining signal overlaps closely with the cKIT staining marking the membrane rather than the nucleus - where the DNMT3A/DNMT3B enzymes are expected to localize and function. In addition, the remaining signal in mutant samples appears mostly as spotty foci. These spotty foci can also be observed in the control samples again associated with the cKIT, however in the control samples, we can also detect the fuzzy staining observed inside the nucleus, which is perceived as the real signal of DNMT3A/DNMT3B enzymes. The nuclear fuzzy staining is dramatically reduced in the mutant samples. Despite our efforts to minimize background signals, some nonspecific binding can occur, especially in knockout samples where the target protein is absent.

To ensure clarity to our readers we revised the Figure 1A legend:

“A) Whole mount immunofluorescence staining of (i) DNMT3A in seminiferous tubules of *Ctrl* and *3aKO* testes, and (ii) DNMT3B in seminiferous tubules of *Ctrl* and *3bKO* testes. All samples were co-stained for cKIT and DNA was visualized by DAPI. Maximum projections of multiple confocal z-stacks are shown. Scale bars = 10 μ m. The remaining cytoplasmic spotty signal in the *Dnmt3a* and *Dnmt3b* single KO samples likely represents background staining.”

8. Several IF images are missing in Supp 1A (this was probably some error during the upload process)

Thank you for bringing this to our attention. We apologize for any inconvenience caused by the missing IF images in Supplementary Figure 1A. We carefully rechecked the Supplementary Figures and ensured that all images are included correctly in the PDF file prior to uploading. If there was an error during the upload process, we will correct it and provide the complete set of images in the revised submission.

9. Supp 1B is not properly referenced in text (this plot does not show the floxed exon specifically).

Thank you for making us aware of this mis reference. Page 4 line 134 revised to:

“By RNA sequencing analysis in spermatogonia, despite measuring similar total RNA levels of both *Dnmt3a* and *Dnmt3b* among control (*Ctrl*) and *DKO* samples (Supplementary Figure 1B), we did not detect reads mapping to the floxed exons of both *Dnmt3a* and *Dnmt3b* genes in *DKO* samples (Supplementary Figure 1C), suggesting efficient excision.”

10. Supp 1A: despite what’s written in the text, there appears to be a fair amount of variation between replicates. How can this be explained.

Thank you for your question. Variation between biological replicates in whole-genome methylation sequencing studies can arise due to several factors. In general, we aim to minimize variability through careful experimental design as outlined below:

- a) We used littermate animals housed together after weaning to match as close as possible the environmental exposures.
- b) We performed the isolation and sorting of sperm cells on the same day to avoid calibration bias of the FACS instrument in order to minimize cell population heterogeneity.
- c) We performed the DNA extraction, library preparation and sequencing using the same reagents across replicates to minimize batch effects
- d) We applied the same data analysis pipelines to avoid computational biases.

However, some level of variability arising by stochastic processes is inevitable and accepted in biological studies. The reviewer can appreciate that most of the variability in our biological replicates may be seen in regions with intermediate levels of DNA methylation. This may arise from the fact that our EM-seq experiment was not extremely deeply sequenced. In fact, single biological replicate C coverage is 3X to 3.25X. This is the reason that we haven't performed differential methylation analysis on a single CpG level but rather performed the downstream analysis in 500bp regions that exhibit at least 4 CpGs and at least 25 informative reads on these CpGs in order to examine more enriched data. Still after these stringent criteria DNA molecules that display intermediate DNA methylation levels could show higher variability. Globally, Pearson correlation coefficients showed quite high agreement between the replicates 0.87 for *Ctrl* and 0.89 for *DKO* samples respectively. To showcase that the depth of sequencing may partially explain the residual variability we show scatterplots between biological replicates and Pearson correlation Coefficient for single CpGs that are covered by more than 30 reads [which is considered the gold standard depth in DNA methylation sequencing experiments] (Revision Figure 7). The variability of the intermediate methylated tiles is greatly reduced, and the Pearson correlation coefficient reach 0.98 and 0.97 for *Ctrl* and *DKO* samples respectively (**Revision Figure 7**).

Revision Figure 7. Scatterplot showing percentage DNAm of single CpG dinucleotides covered by more than 30 reads in the EM-seq experiment (left) replicate 1 *Ctrl* (x-axis) and replicate 2 *Ctrl* (y-axis) sperm and (right) replicate 1 *DKO* (x-axis) and replicate 2 *DKO* (y-axis) sperm. The Pearson correlation coefficient between the rep1 and rep2 for each genotype is presented in the lower right corner.

11. Supp 2D i: DNA methylation data missing

Thank you for bringing this to our attention. We apologize for any inconvenience caused by the missing data. We will carefully recheck the Supplementary Figures and ensure that all images appear correctly. We believe that this is a genuine technical error in the uploading/downloading process. We also refer to Reviewer 2, who commented on the DNA methylation status of the *Rasgrf1* DMR, suggesting that part of this figure is correctly shown.

12. Supp 3D mentioned out of sequence (line 189)

Thank you for bringing this to our attention. We wished to refer the reader to Supplementary Figure 3D to appreciate the expression levels of *Dnmt1*, thus we revised the page 6 line 196 to:

“However, despite a significant decrease in DNAm in *DKO* sperm, these regions maintained mid-levels of DNAm (25%-75%), suggesting that DNMT3A/DNMT3B confer *de novo* activity, while other DNA methyltransferases (such as *Dnmt1* which is highly expressed in spermatogonia (Supplementary Figure 3D)), may act on these regions as well (Figures 1E, 1F).”

13. Supp 5A: On PCA plot, why does *Dnmt3b* KO sperm look closer to SgU than SgD?

Thank you for your question about the PCA plot. The scales of the x and y axis in the former plot were not similar. To avoid such confusion, we have updated the plot with fixed axis scales and moved it to the Main Figure 2A, according to the suggestion of the reviewer (see above). Along the line of argumentation presented in manuscript Results and Discussion sections, DNMT3B plays a critical role in *de novo* DNA methylation during the differentiation of spermatogonia, supported by the fact that *Dnmt3b* *sKO* SgU and *Dnmt3b* *sKO* SgD are separated only by the second PC component [which captures much less variation compared to the first], but not the first PC which captures mostly the developmental progression as seen in the *Control* samples. However, the *Dnmt3b* *sKO* sperm clusters further apart from *Dnmt3b* *sKO* SgU and *Dnmt3b* *sKO* SgD in the first PC component suggesting that the presence of DNMT3A [and potentially DNMT1] during meiosis and haploid differentiation can still affect DNA methylation patterns in a way that makes the sperm methylome is slightly different to that of differentiated (SgD) spermatogonia (Figure 2A).

14. Could other examples of screen shots be used besides *Bend3*? It seemed odd only to use example repeatedly, and the dynamically methylated region was not obviously important for gene regulation.

Thank you for your comment. We used the *Bend3* locus as a representative example to follow DNA methylation and chromatin dynamics throughout development. However, we understand your concern about the repeated use of this example and added also the *Srsf4* locus to the corresponding main Figures 1 to 6 to provide a broader perspective. Additionally, using such example loci, we do not wish to claim that the hypoDMRs are important for the transcriptional regulation of their associated genes during spermatogenesis. In fact, the minimal transcriptional changes observed in our mouse model suggest a more nuanced role for these regions in gene regulation.

REVIEWERS' COMMENTS

We would like to express our sincere appreciation for the reviewers' positive comments on our revised manuscript. We are delighted to hear that the revisions and additional analyses we implemented have satisfactorily addressed the reviewers' initial concerns.

Reviewer #1 (Remarks to the Author):

Dear Authors,

I have carefully reviewed your responses to my comments and the revised manuscript. I am pleased to see that you have thoroughly addressed the points raised, especially the clarifications regarding DNMT3A's role in safeguarding DNA methylation and the additional data provided to support the hypotheses on nucleosome occupancy and epigenetic changes.

I find the revisions to be satisfactory, and the additional analyses strengthen the conceptual framework and conclusions of the study. I have no further major comments or concerns.

Therefore, I recommend that this manuscript be accepted for publication in its current form.

Sincerely,

Reviewer #1

We are pleased that the clarifications on DNMT3A's role and the additional analysis have met your expectations. We deeply appreciate your recommendation for the publication of the manuscript in its current revised form.

Reviewer #2 (Remarks to the Author):

The authors have thoroughly addressed my concerns, and I have no further comments on the revised version of the manuscript.

We are grateful for your time and for approving that our responses and revisions adequately addressed your comments.

Reviewer #3 (Remarks to the Author):

The authors have fully addressed all questions raised in the previous review. The manuscript greatly improved.

Thank you for your recognition of the improvements made to the manuscript. We are glad that the revised manuscript fully addresses your queries and meets your standards.

Reviewer #4 (Remarks to the Author):

Thank you for your time and we appreciate your collaborative approach to co-reviewing.

Reviewer #5 (Remarks to the Author):

I appreciate the thorough response provided to the other reviewers and myself. As I stated in my original review, I think this is a scientifically sound study that will be well-received and cited by epigenetics researchers, especially those focused on DNA methylation and mammalian development. The authors appropriately responded to my comments, and I think the manuscript is improved as a result. I am particularly pleased with the new H3K36me3 analysis. I have no reservations about this paper's publication.

Thank you for your thoughtful feedback and positive assessment of the revised manuscript. We appreciate your endorsement of the paper's contribution to the field and your recommendation for publication.